# FedGPS: Statistical Rectification Against Data Heterogeneity in Federated Learning

**Zhiqin Yang**[1]    **Yonggang Zhang**[3]    **Chenxin Li**[1]
**Yiu-ming Cheung**[2]    **Bo Han**[2]    **Yixuan Yuan**[1*]
[1]The Chinese University of Hong Kong    [2]Hong Kong Baptist University
[3]The Hong Kong University of Science and Technology

## Abstract

Federated Learning (FL) confronts a significant challenge known as data heterogeneity, which impairs model performance and convergence. Existing methods have made notable progress in addressing this issue. However, improving performance in certain heterogeneity scenarios remains an overlooked question: *How robust are these methods to deploy under diverse heterogeneity scenarios?* To answer this, we conduct comprehensive evaluations across varied heterogeneity scenarios, showing that most existing methods exhibit limited robustness. Meanwhile, insights from these experiments highlight that sharing statistical information can mitigate heterogeneity by enabling clients to update with a global perspective. Motivated by this, we propose **FedGPS** (**Fed**erated **G**oal-**P**ath **S**ynergy), a novel framework that seamlessly integrates statistical distribution and gradient information from others. Specifically, FedGPS statically modifies each client's learning objective to implicitly model the global data distribution using surrogate information, while dynamically adjusting local update directions with gradient information from other clients at each round. Extensive experiments show that FedGPS outperforms state-of-the-art methods across diverse heterogeneity scenarios, validating its effectiveness and robustness. The code is available at: https://github.com/CUHK-AIM-Group/FedGPS.

## 1 Introduction

Federated Learning (FL) facilitates collaborative model training across distributed data sources, garnering substantial interest in recent years [1, 2, 3, 4]. Its primary goal is to keep data localized to protect sensitive information while harnessing contributions from other participants to enhance individual models [5, 6] or construct a superior global model [7]. However, this decentralized paradigm encounters data heterogeneity [8, 9] (also known as statistical heterogeneity), due to the variations in client devices, geographic locations, and annotation processes [10, 11]. This departure from the assumption of independent and identically distributed (i.i.d.) data presents a substantial challenge, complicating the training of distributed networks across diverse data distributions to achieve robust generalization on the overall data distribution [8, 12].

To enhance performance in FL, numerous studies have advanced efforts to mitigate the impact of data heterogeneity [13, 14, 15, 16]. FedAvg [7] introduces the paradigm of local training followed by aggregation. Moreover, several studies have refined the learning objective of local training by incorporating constraints to mitigate client drift [11, 12, 17, 18]. Client sampling [19, 20] and global aggregation weight adjustments [21, 22] have also been tailored to adapt to heterogeneity scenarios. Additionally, information-sharing strategies [23, 14, 15] have emerged as a promising approach

---

*Corresponding author: Yixuan Yuan (yxyuan@ee.cuhk.edu.hk).

39th Conference on Neural Information Processing Systems (NeurIPS 2025).

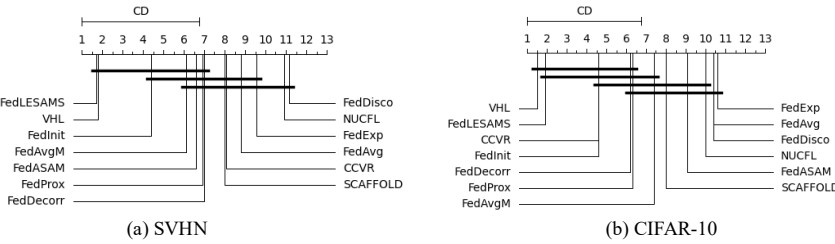

Figure 1: Nemenyi post-hoc test results on the performance under (a) SVHN and (b) CIFAR-10. Black horizontal lines indicate the critical distance (CD).

to mitigate heterogeneity, though they often require increased communication or computational resources and demand careful privacy considerations to protect sensitive data.

Distributed environments are inherently complicated, leading to diverse scenarios involving different clients. As a result, dataset heterogeneity varies across settings. To assess the robustness of algorithms under various data distribution scenarios, we raise a previously underexplored question:

> *To what extent do existing methods maintain robustness across diverse scenarios, and by what mechanisms is this robustness achieved?*

The results presented in Fig. 1, Tabs. 1, 2, 3 and 4 show that most methods exhibit limited robustness, as indicated by the CD intervals that overlap between methods. This overlap highlights the challenges these methods face in adapting to diverse data distributions. Nevertheless, the findings indicate that statistical information from other clients provides valuable insights for refining local updates. Moreover, sharing detailed information risks privacy leakage, which contravenes the core principles of FL, while coarse-grained statistics, such as CCVR [13], sharing the mean and covariance of logits, offer only marginal improvements in adaptability across varied scenarios. Thus, determining which statistical information to use and how to leverage it effectively remains a significant challenge.

First, we revisit the objective of FL, wherein each client trains a model on its local data distribution, and these models are aggregated with the expectation of achieving robust generalization across the global data distribution. However, this process often introduces a distribution gap due to data heterogeneity. **(1) Distribution-Level:** Inspired by [14, 24], we introduce a static modification to the goal of local training, enabling implicit learning of the global data distribution through a *privacy-free* surrogate distribution via a two-stage statistical information alignment process, as depicted in Fig. 3(a). *Stage 1*, the local data distribution is aligned with a local surrogate distribution using the local model. *Stage 2*, the local surrogate distribution is aligned with a global surrogate distribution. Through these stages, the divergence between the local and global distributions is effectively bounded, improving generalization while maintaining privacy.

Furthermore, achieving effective distribution alignment becomes difficult when the distribution shift is substantial or the amount of data available per round is limited (e.g., low client sampling rate). **(2) Gradient-Level:** Drawing inspiration from [25], we propose incorporating gradient information from other clients prior to determining the local update direction. This strategy highlights the importance of utilizing insights from other clients' gradients to dynamically adjust the local optimization path at each step, ensuring a more globally consistent update direction. Moreover, theoretical analysis reveals that careful parameter tuning of this gradient term can further rectify the update direction, resulting in a measurable reduction in the global model's loss function. Building on this two-level alignment strategy, we introduce FedGPS, a synergistic framework that integrates goal and path coordination, designed to ensure robustness in label-distribution-agnostic scenarios. Extensive experiments conducted on three benchmark datasets confirm the effectiveness of FedGPS, showcasing its superior performance across diverse scenarios.

Our contributions are summarized as follows:

- We comprehensively evaluate existing federated learning methods designed to address heterogeneity, showing that most exhibit limited robustness across diverse distribution partitions. Our findings highlight the significant potential of leveraging statistical information from other clients to enhance performance.

- Motivated by these insights, we attempt to propose a new framework to adapt to various heterogeneity scenarios called **FedGPS**, which incorporates statistical information from other clients from two perspectives. At the distribution level, we constrain local models to learn data distribution aligned with the global distribution using surrogate information. Concurrently, we refine the update direction at each step based on other client information at the gradient view, enabling a more holistic optimization process.

- Extensive experiments with our framework across diverse settings and benchmark datasets demonstrate the efficacy of FedGPS. Our results show that FedGPS surpasses existing methods, achieving robust and SOTA performance across various distribution splits.

## 2 Related Work

Federated Learning (FL) enables localized data processing to preserve sensitive information, but this often results in data heterogeneity due to diverse data collection conditions. To mitigate this, several strategies have been developed to align local optimization with global objectives. For instance, FedProx [11] incorporates a proximal term to limit divergence between local and global parameters, ensuring more stable updates. Similarly, SCAFFOLD [12] introduces a control variate to correct local updates, while PAdaMFed [18] enhances convergence by integrating gradients and control terms from consecutive rounds to better estimate the global optimization direction. Another promising approach focuses on achieving a flatter loss landscape to enhance model robustness against heterogeneity. Techniques such as FedSAM [26], MoFedSAM [27], FedGAMMA [28], and FedLESAM [29] perturb local parameters before updates, improving generalization and robustness, as supported by sharpness-aware minimization principles [30]. Sharing information among participants has garnered increasing attention. FedProto [31] shares class prototypes instead of model parameters, preserving privacy while inspiring subsequent approaches such as FedProK [32] and PILORA [33]. Additionally, generative models [34, 35] and local statistical methods [13] have been effectively employed to address heterogeneity challenges, enhancing model robustness across diverse data distributions. However, these methods may raise additional privacy concerns, prompting exploration of privacy-preserving mechanisms [36].

On the server side, optimizing client selection strategies [37, 38, 39] is crucial for minimizing communication overhead by prioritizing clients most relevant to the global model, thereby improving efficiency. Advanced aggregation techniques further address heterogeneity. FedDisco [21] employs discrepancy-aware weights that consider factors beyond mere data size, while other works revisit aggregation protocols for improved performance [40]. Some methods design different aggregation methods on the server side, such as FedMR [41]. Server-side generative approaches, such as data-free knowledge distillation [42, 43], have also been explored to mitigate heterogeneity, offering a complementary perspective to client-side innovations. Besides, FL has also received a lot of attention in many areas, e.g, healthcare [44, 45] and transportation [46].

## 3 Preliminary

**Federated Learning.** In a typical federated learning setup [7, 47], data samples are distributed across a set of $K$ participating clients $\mathcal{S} = \{1, 2, \ldots, K\}$, without being centralized on a server. Each client $k \in \mathcal{S}$ maintains a local model parameterized by $\boldsymbol{\theta}_k$ and collaboratively contributes to training a global model parameterized by $\boldsymbol{\theta}$. For each client $k$, the $i$-th data sample $\boldsymbol{\xi}_{k,i} := (\mathbf{x}_{k,i}, y_{k,i})$, is drawn from its private local distribution $\mathcal{D}_k$. Then, the federated learning process can thus be formulated as the following optimization problem:

$$\boldsymbol{\theta}^* = \underset{\boldsymbol{\theta} \in \mathbb{R}^{|\boldsymbol{\theta}|}}{\arg\min} F(\boldsymbol{\theta}) := \sum_{k=1}^{K} p_k F_k(\boldsymbol{\theta}_k), \tag{1}$$

where $p_k$ represents the weight of client $k$. This equation captures the goal of FL, which seeks to get the optimal global model $\boldsymbol{\theta}^*$ that minimizes the global objective $F(\boldsymbol{\theta})$ by optimizing local objectives $F_k(\boldsymbol{\theta}_k)$ for each client, expressed as:

$$F_k(\boldsymbol{\theta}_k) := \mathbb{E}_{\boldsymbol{\xi}_k \sim \mathcal{D}_k} \left[ \ell(\boldsymbol{\theta}_k; \boldsymbol{\xi}_k) \right], \tag{2}$$

where $\ell(\cdot, \cdot)$ is the loss function, e.g., cross-entropy in a supervised learning task. The local update at $t$-th round usually follows the conventional stochastic gradient descent (SGD) with $\eta_l$ denoting the

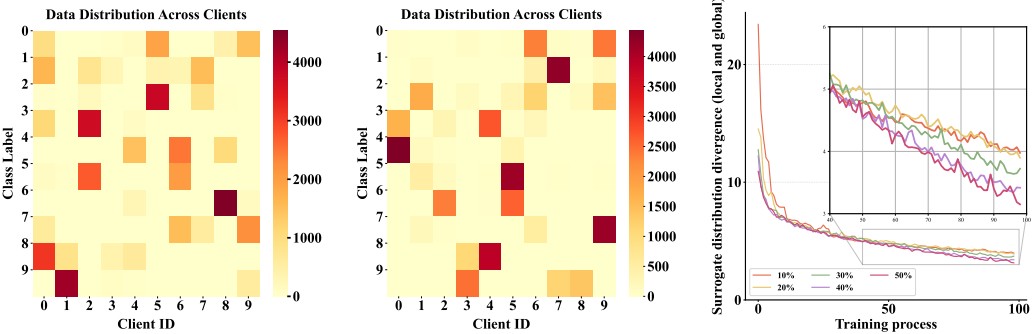

(a) Heterogeneous distribution 1    (b) Heterogeneous distribution 2    (c) Distribution divergence measure

Figure 2: (a) and (b) are examples of data distribution scenarios generated using the Dirichlet partition method under the CIFAR-10 dataset across 10 clients. All scenarios use the same heterogeneity control factor of $\alpha = 0.1$, but vary the random seed to produce different heterogeneous distributions. (c) The divergence between local and global surrogate distributions is computed as the FL training proceeds with different ratios of client sampling, also means the proportion of the data that participates in the global update at each federated training round. The divergence is computed every 5 rounds.

local step size as follows:

$$\boldsymbol{\theta}_k^{t+1} = \boldsymbol{\theta}_k^t - \eta_l \nabla F_k(\boldsymbol{\theta}_k^t; \boldsymbol{\xi}_k). \tag{3}$$

The locally updated models are uploaded to the server, which derives a new global model through an aggregation mechanism AGG($\cdot$) based on the $t$-th round collected local data (e.g., Eq(1)), global model $\boldsymbol{\theta}^t$, and the global step size $\eta_g$:

$$\boldsymbol{\theta}^{t+1} = \text{AGG}(\eta_g; \boldsymbol{\theta}^t; \{\boldsymbol{\theta}_k^{t+1}\}_{k \in \mathcal{S}_t}), \tag{4}$$

where $\mathcal{S}_t$ denotes the set of clients participating in the $t$-th training round.

**Definition 3.1** (Wasserstein Distance). *Consider two probability distributions $\mu$ and $\nu$ over the data space $\mathcal{X} \times \mathcal{Y}$, where $\mathcal{X} \subset \mathbb{R}^d$ is the feature space and $\mathcal{Y}$ is the label space. Given a distance metric $d$ on $\mathcal{X} \times \mathcal{Y}$, the $p$-Wasserstein distance between $mu$ and $nu$, for any $p \geq 1$, is defined as:*

$$W_p(\mu, \nu) := \left( \inf_{\gamma \in \Gamma(\mu, \nu)} \int_{(x,y) \sim \mu, (x',y') \sim \nu} d((x,y), (x',y'))^p \, d\gamma((x,y), (x',y')) \right)^{1/p},$$

*where $\Gamma(\mu, \nu)$ is the set of all joint distributions $\gamma$ with marginals $\mu$ and $\nu$, respectively.*

## 4 Methodology

This section details our proposed "*FedGPS*" framework. We begin by outlining the motivation behind FedGPS(Sec. 4.1). Subsequently, we describe how statistical information is leveraged from two perspectives: the distribution view (Sec. 4.2) and the gradient perspective (Sec. 4.3).

### 4.1 Motivation

Performance degradation in FL stems from the divergence between local and global data distributions. Training on shifted local distributions $\mathcal{D}_k$ while expecting generalization on the global i.i.d. distribution $\mathcal{D}_g$ naturally creates a distribution gap. This can be expressed as:

$$\boldsymbol{\theta}^* = \arg\min_{\boldsymbol{\theta}} \mathbb{E}_{\mathcal{D}_g} \left[ F(\text{AGG}(\boldsymbol{\theta}_k)) \right], \text{where } \boldsymbol{\theta}_k = \arg\min_{\boldsymbol{\theta}_k} \mathbb{E}_{\mathcal{D}_k} \left[ F_k(\boldsymbol{\theta}_k, \boldsymbol{\xi}_k) \right]. \tag{5}$$

Consequently, this divergence results in a performance gap with respect to the global distribution. The distribution shift can be quantified using the $p$-Wasserstein distance based on Definition 3.1.

To address this gap, existing methods often share distribution-related information. For example, FedProto [31] shares class-specific average embeddings as prototypes, while FLGAN [48] uses synthetic data from a Conditional GAN (CGAN) [49]. However, these approaches, which involve sharing raw data-derived information, introduce privacy risks. Additionally, VHL [14] employs

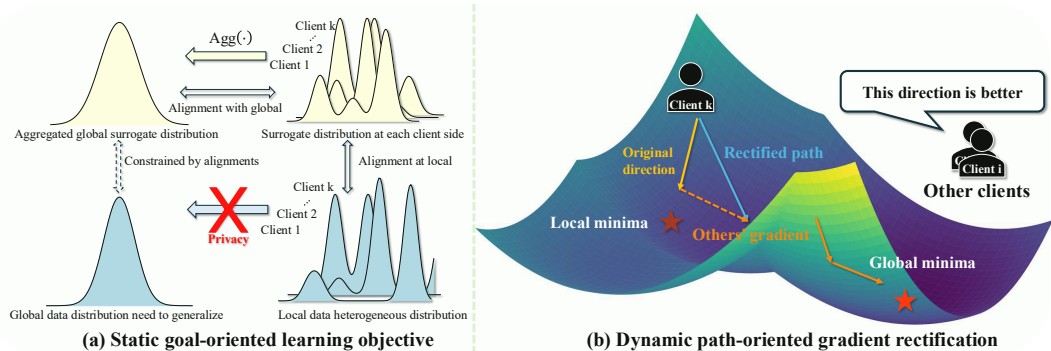

**(a) Static goal-oriented learning objective**      **(b) Dynamic path-oriented gradient rectification**

Figure 3: (a) Static goal-oriented objective. This objective is composed of two stages: local distribution aligns with local surrogate distribution (Alignment at local), and local surrogate distribution aligns with global surrogate distribution (Alignment with global). (b) Dynamic path-oriented rectification corrects the original update direction by the gradient of other clients for a new update path.

domain adaptation [50] with virtual homogeneity data, yet this data still exhibits distribution shifts, as virtual data is trained on shifted models at each client.

Drawing on our evaluations and prior work [14, 24], we introduce a privacy-preserving surrogate distribution (e.g., sampled from distinct Gaussian distributions) to tackle the distribution gap in federated learning (FL). This surrogate improves FL performance by minimizing the upper bound of the global model's generalization error through a two-stage alignment process. However, alignment quality varies with the volume of training data per round. For instance, in low client participation scenarios involving approximately $10\%$ data per round, the surrogate distribution gap between global and local models (red line in Fig. 2(c)) exceeds that observed with $50\%$ data participation (pink line). This insight prompts us to investigate the parameter space, where model updates gain a more global perspective by leveraging statistical information that partially reflects the data distribution. Unlike prior approaches that focus on a single aspect, our method coordinates both distribution and parameter spaces, enhancing robustness across diverse FL heterogeneous scenarios.

## 4.2 Static Goal-oriented Objective

Building on the motivation outlined above, we propose a **static goal-oriented objective function** that changes each client's learning goal to better generalize on the global distribution $\mathcal{D}_g$ by a two-stage alignment (as depicted in Fig. 3(a)), rather than solely optimizing for the local distribution $\mathcal{D}_k$. Firstly, we give the formal definition of the surrogate dataset in Definition 4.1.

**Definition 4.1** (Surrogate Dataset). *FedGPS assigns a distinct Gaussian distribution to each class in the original dataset. The surrogate dataset $\mathcal{D}^s$ is then generated by sampling from these class-specific Gaussian distributions, with each client holding the same surrogate dataset $\mathcal{D}^s$.*

Then, we decompose the model parameters $\theta$ into a classifier $h$ and a feature extractor $\psi$ (where $\theta = h \circ \psi$) cause we perform the alignment in the feature space. $\mathcal{P}_k$ represents the probability distribution of the local data in the feature space, induced by applying the feature extractor $\psi_k$ to samples $\xi_k$, where $\xi_k \sim \mathcal{D}_k$. Following, $\mathcal{P}_k^s$ is the distribution of $k$-th local surrogate distribution with $\mathcal{D}^s$, and the global surrogate distribution $\mathcal{P}_s$ is aggregated at the server side. Specifically, we give the formal definition of local surrogate distribution and global surrogate distribution as follows:

**Definition 4.2** (Local Surrogate Distribution). *For a client $k$ in a federated learning system, the local surrogate distribution $\mathcal{P}_k^s$ is conceptually defined as the set of feature embeddings obtained by passing each data point from the surrogate dataset through the $k$-th local model's feature extractor $\psi_k$ at the client side.*

In the implementation, to ensure privacy and reduce communication overhead, what is transmitted to the server is a compressed, privacy-preserving statistical representation of the surrogate distribution. Typically, these embeddings are then aggregated $\mathcal{E}_{k,c}^s = \frac{1}{|\mathcal{D}_c^s|} \sum_{\xi_c^s \sim \mathcal{D}^s} \psi_k(\xi_s^c)$ to form a set of class-wise prototype vectors (e.g., 512-dimensional), with each prototype representing a specific class $c$. This distribution serves as a compact proxy for the local surrogate distribution.

**Definition 4.3** (Global Surrogate Distribution). *At the server side, the global surrogate distribution $\mathcal{P}^s$ is defined as the set of feature embeddings obtained by passing each data point from the surrogate dataset through the global extractor $\psi$.*

To alleviate the burden of the global surrogate distribution compute, the global surrogate distribution is replaced with the aggregation of selected local surrogate prototypes $\mathcal{E}_c^s = \sum_{k \in \mathcal{S}_t} \mathcal{E}_{k,c}^s$ at round $t$. Furthermore, we introduce the following theorem to formalize the new objective and quantify the alignment between the local and global distributions, which establishes bounds on the Wasserstein-1 distance between the distribution gap we analyzed before.

**Theorem 4.4.** *Given the global feature distribution $\mathcal{P}_g$, the local feature distribution $\mathcal{P}_k$, the surrogate distributions $\mathcal{P}^s$ (global) and $\mathcal{P}_k^s$ (local for the $k$-th client) over their corresponding data space. Suppose there exists $\kappa \geq 0$ such that $W_1(\mathcal{P}_{k,\mathcal{D}}, \mathcal{P}_{g,\mathcal{D}}) \leq \kappa$ under distribution $\mathcal{D}$. If the following conditions hold:*

$$W_1(\mathcal{P}_k^s, \mathcal{P}_k) \leq \epsilon_1, \quad W_1(\mathcal{P}_k^s, \mathcal{P}^s) \leq \epsilon_2,$$

*where $W_1$ is the Wasserstein-1 distance as defined in Definition 3.1. then the Wasserstein-1 distance between $\mathcal{P}_g$ and $\mathcal{P}^s$ is bounded as:*

$$W_1(\mathcal{P}_g, \mathcal{P}^s) \leq \epsilon_1 + \epsilon_2 + \kappa.$$

*Remark* 1. This theorem establishes a key relationship between local and global feature distributions. Specifically, suppose each client's local surrogate feature distribution $\mathcal{P}_k^s$ closely approximates its true local feature distribution $\mathcal{P}_k$ within a tolerance of $\epsilon_1$ (*Stage 1*). Additionally, assume $\mathcal{P}_k^s$ aligns with the global surrogate feature distribution $\mathcal{P}^s$ within a tolerance of $\epsilon_2$ (*Stage 1*). Furthermore, let the local and global feature extractors produce similar outputs for identical data, within a tolerance of $\kappa$. Under these conditions, the global model's feature distribution $\mathcal{P}_g$ will closely resemble $\mathcal{P}^s$ (Detailed proof can be found in the Appendix A).

The bound $\epsilon_1 + \epsilon_2 + \kappa$ ensures that a model trained on surrogate data generalizes effectively to the true global data. In practice, $\epsilon_1$ and $\epsilon_2$ can be optimized using distribution-matching losses, while $\kappa$ can be minimized through parameter regularization. Accordingly, our local optimization goal of each client can be formulated as follows:

$$F_k(\boldsymbol{\theta}_k) := \mathbb{E}_{\boldsymbol{\xi}_k \sim \mathcal{D}_k} \ell(\boldsymbol{\theta}_k; \boldsymbol{\xi}_k) + \mathbb{E}_{\boldsymbol{\xi}_s \sim \mathcal{D}^s} \ell(\boldsymbol{\theta}_k; \boldsymbol{\xi}_s) + \lambda_1 d(\mathcal{P}_k, \mathcal{P}_k^s) + \lambda_2 d(\mathcal{P}^s, \mathcal{P}_k^s) + \lambda_3 \|\boldsymbol{\theta}_k\|^2, \quad (6)$$

where the first two terms enhance the generalization of the local model $\boldsymbol{\theta}_k$ on both the local data distribution $\mathcal{D}_k$ and the surrogate data distribution $\mathcal{D}^s$. The function $d(\cdot, \cdot)$ quantifies the distance between distributions, such as the Wasserstein-1 distance. The terms weighted by hyperparameters $\lambda_1$, $\lambda_2$, and $\lambda_3$ control the trade-off between terms, which are tuned to optimize performance.

## 4.3 Dynamic Path-oriented Rectification

To overcome the limitations of scarce data involved every round in achieving distribution alignment (demonstrated by Fig. 2(c)), we develop another technique, **dynamic path-oriented gradient rectification**, to bolster model robustness. Our motivation draws a high-level concept from the model replacement strategy in the backdoor of federated models [25]. In this scenario, the malicious client exploits a deep understanding of the aggregation mechanism and the collective dynamics of benign clients. By precisely predicting the contributions of other clients' updates to the global model, the attacker meticulously designs and scales their malicious update. The key insight is that awareness of the aggregated influence from other clients confers substantial leverage in shaping the global model.

We define the gradient statistical information from other clients in Definition 4.5. Then we elaborate on how to utilize this information to improve the local update from a more global perspective at the gradient level (as depicted in Fig. 3(b)).

**Definition 4.5** (Non-Self Gradient at Round $t$ of client $i$, $\delta_{\boldsymbol{\theta}_i}^t$). *In a FL framework with a client set $\mathcal{K}$, let $\boldsymbol{\theta}^{t-1}$ denote the global model parameters at the end of round $t-1$, and $\mathcal{S}_{t-1} \subseteq \mathcal{K}$ the subset of clients selected for round $t-1$ and $|\mathcal{S}_{t-1}| \geq 2$. For each client $k \in \mathcal{K}$, $\Delta_{\boldsymbol{\theta}_k}^{t-1}$ be the updated information of client $k$ at round $t-1$, where $\Delta_{\boldsymbol{\theta}_k}^{t-1} = \boldsymbol{\theta}_k^t - \boldsymbol{\theta}_k^{t-1}$. Let $\eta_g$ and $\eta_l$ denote the global and local update steps, respectively.*

*For a client $i \in \mathcal{K}$, the Non-Self Gradient at round $t$ of client $i$, denoted $\delta_{\boldsymbol{\theta}_i}^t$, is defined as:*

$$\delta_{\boldsymbol{\theta}_i}^t = -\eta_g \eta_l \frac{1}{|\mathcal{S}_{t-1} \setminus \{i\}|} \sum_{k \in \mathcal{S}_{t-1} \setminus \{i\}} \Delta_{\boldsymbol{\theta}_k}^{t-1},$$

Table 1: Top-1 accuracy of baselines and our method FedGPS with 5 different heterogeneous scenarios on CIFAR-10, heterogeneity degree $\alpha = 0.1$, local epochs $E = 1$ and total client number $K = 10$.

| Dataset: CIFAR-10 Heterogeneity Level:$\alpha = 0.1$ Client Number:$K = 10$, Client Sampling Rate: 50% Total Communication Round:$T = 500$ Local Epochs:$E = 1$ | | | | | | | | | | | | | | | | |
|---|---|---|---|---|---|---|---|---|---|---|---|---|---|---|---|---|
| Diff Scenario | Heterogeneous scenario 1 | | | Heterogeneous scenario 2 | | | Heterogeneous scenario 3 | | | Heterogeneous scenario 4 | | | Heterogeneous scenario 5 | | | |
| Centralized Training Acc=xxx% | | | | | | | | | | | | | | | | |
| | ACC↑ | ROUND↓ | SpeedUp↑ | ACC↑ | ROUND↓ | SpeedUp↑ | ACC↑ | ROUND↓ | SpeedUp↑ | ACC↑ | ROUND↓ | SpeedUp↑ | ACC↑ | ROUND↓ | SpeedUp↑ | |
| Methods | Target Acc=84% | | | Target Acc=79% | | | Target Acc=80% | | | Target Acc=68% | | | Target Acc=65% | | | Mean Acc± Std |
| FedAvg | 84.21 | 340 | 1.0× | 79.13 | 301 | 1.0× | 80.63 | 416 | 1.0× | 68.62 | 189 | 1.0× | 65.86 | 415 | 1.0× | 75.69 ± 7.99 |
| FedAvgM | 85.74 | 181 | 1.9× | 81.78 | 200 | 1.5× | 81.35 | 310 | 1.3× | 70.15 | 348 | 0.5× | 67.51 | 233 | 1.8× | 77.31 ± 7.98 |
| FedProx | 86.13 | 181 | 1.9× | 83.12 | 179 | 1.7× | 82.37 | 219 | 1.9× | 76.62 | 175 | 1.1× | 68.81 | 168 | 2.5× | 79.41 ± 6.85 |
| SCAFFOLD | 82.39 | None | None | 80.78 | 412 | 0.7× | 79.08 | None | None | 71.83 | 193 | 1.0× | 68.43 | 175 | 2.4× | 76.50 ± 6.05 |
| CCVR | 84.30 | 391 | 0.9× | 83.28 | 136 | 2.2× | 83.20 | 192 | 2.2× | 76.57 | 53 | 3.6× | 74.72 | **66** | **6.3×** | 80.41 ± 4.42 |
| VHL | 89.07 | **116** | **2.9×** | 87.20 | 131 | 2.3× | 86.83 | 210 | 2.0× | 84.30 | 89 | 2.1× | 81.05 | 160 | 2.6× | 85.69 ± 3.10 |
| FedASAM | 86.49 | 270 | 1.3× | 81.99 | 211 | 1.4× | 80.45 | 310 | 1.3× | 73.11 | 188 | 1.0× | 66.68 | 348 | 1.2× | 77.74 ± 7.84 |
| FedExp | 84.00 | 270 | 1.3× | 79.25 | 211 | 1.4× | 79.60 | None | None | 71.55 | 188 | 1.0× | 66.66 | 315 | 1.3× | 76.21 ± 6.97 |
| FedDecorr | 85.76 | 339 | 1.0× | 84.07 | 244 | 1.2× | 81.38 | 358 | 1.2× | 73.14 | 181 | 1.0× | 73.77 | 212 | 2.0× | 79.62 ± 5.85 |
| FedDisco | 85.69 | 270 | 1.3× | 81.84 | 191 | 1.6× | 80.42 | 364 | 1.1× | 70.37 | 188 | 1.0× | 69.94 | 315 | 1.3× | 77.65 ± 7.11 |
| FedInit | 86.84 | 339 | 1.0× | 83.49 | 244 | 1.2× | 80.48 | 414 | 1.0× | 69.44 | 318 | 0.6× | 68.04 | 175 | 2.4× | 77.66 ± 8.46 |
| FedLESAM | 88.80 | 151 | 2.3× | 85.52 | 120 | 2.5× | 84.24 | 233 | 1.8× | 78.99 | 90 | 2.1× | 74.18 | 119 | 3.5× | 82.35 ± 5.77 |
| NUCFL | 83.76 | None | None | 79.45 | 378 | 0.8× | 79.76 | None | None | 68.78 | 210 | 0.9× | 65.78 | 487 | 0.9× | 75.51 ± 7.77 |
| **FedGPS(Ours)** | **90.31** | 139 | 2.4× | **88.45** | **119** | **2.5×** | **87.78** | **158** | **2.6×** | **85.06** | **89** | **2.1×** | **82.04** | 137 | **3.0×** | **86.73 ± 3.23** |

Table 2: Top-1 accuracy of baselines and our method FedGPS with 5 different heterogeneous scenarios on SVHN, heterogeneity degree $\alpha = 0.1$, local epochs $E = 1$ and total client number $K = 10$.

| Dataset: SVHN Heterogeneity Level:$\alpha = 0.1$ Client Number:$K = 10$, Client Sampling Rate: 50% Total Communication Round:$T = 500$ Local Epochs:$E = 1$ | | | | | | | | | | | | | | | | |
|---|---|---|---|---|---|---|---|---|---|---|---|---|---|---|---|---|
| Diff Scenario | Heterogeneous scenario 1 | | | Heterogeneous scenario 2 | | | Heterogeneous scenario 3 | | | Heterogeneous scenario 4 | | | Heterogeneous scenario 5 | | | |
| Centralized Training Acc=84% | | | | | | | | | | | | | | | | |
| | ACC↑ | ROUND↓ | SpeedUp↑ | ACC↑ | ROUND↓ | SpeedUp↑ | ACC↑ | ROUND↓ | SpeedUp↑ | ACC↑ | ROUND↓ | SpeedUp↑ | ACC↑ | ROUND↓ | SpeedUp↑ | |
| Methods | Target Acc=85% | | | Target Acc=92% | | | Target Acc=92% | | | Target Acc=92% | | | Target Acc=92% | | | Mean Acc± Std |
| FedAvg | 85.61 | 151 | 1.0× | 92.56 | 102 | 1.0× | 92.73 | 100 | 1.0× | 92.08 | 340 | 1.0× | 92.11 | 65 | 1.0× | 91.02 ± 2.72 |
| FedAvgM | 88.64 | 150 | 1.0× | 92.41 | 110 | 0.9× | 92.34 | 99 | 1.0× | 92.30 | 144 | 2.4× | 93.34 | 64 | 1.0× | 91.81 ± 1.82 |
| FedProx | 88.65 | 102 | 1.5× | 93.07 | 107 | 1.0× | 93.13 | 154 | 0.6× | 92.56 | 104 | 3.3× | 92.94 | 64 | 1.0× | 92.07 ± 1.92 |
| SCAFFOLD | 87.88 | 98 | 1.5× | 91.58 | None | None | 92.22 | 75 | 1.3× | 91.86 | None | None | 91.74 | None | None | 91.06 ± 1.79 |
| CCVR | 89.77 | **27** | **5.6×** | 91.41 | None | None | 92.68 | 56 | 1.8× | 92.08 | 214 | 1.0× | 92.65 | 91 | 0.7× | 91.72 ± 1.21 |
| VHL | 93.57 | 43 | 3.5× | 94.89 | 110 | 0.9× | 94.99 | 93 | 1.1× | 94.96 | 85 | 4.0× | 94.90 | 64 | 1.0× | 94.66 ± 0.61 |
| FedASAM | 88.14 | 150 | 1.0× | 92.56 | 107 | 1.0× | 92.82 | 92 | 1.1× | 92.65 | 116 | 2.9× | 93.19 | 64 | 1.0× | 91.87 ± 2.10 |
| FedExp | 86.24 | 150 | 1.0× | 92.11 | 110 | 0.9× | 91.87 | None | None | 92.03 | 339 | 1.0× | 92.83 | 64 | 1.0× | 91.02 ± 2.70 |
| FedDecorr | 89.82 | 80 | 1.9× | 92.99 | 235 | 0.4× | 93.02 | 71 | 1.4× | 93.19 | 182 | 1.9× | 93.11 | 64 | 1.0× | 92.43 ± 1.46 |
| FedDisco | 84.54 | None | None | 92.50 | 100 | 1.0× | 92.50 | 99 | 1.0× | 91.91 | None | None | 92.83 | 64 | 1.0× | 90.92 ± 3.58 |
| FedInit | 86.69 | 368 | 0.4× | 90.50 | None | None | 93.83 | 180 | 0.6× | 93.16 | 134 | 2.5× | 93.61 | 64 | 1.0× | 91.56 ± 3.03 |
| FedLESAM | 89.29 | 165 | 0.9× | 93.62 | 173 | 0.6× | 94.86 | 63 | 1.6× | 93.78 | 134 | 2.5× | 94.71 | 64 | 1.0× | 93.25 ± 2.28 |
| NUCFL | 86.49 | 118 | 1.3× | 90.53 | None | None | 91.93 | None | None | 91.36 | None | None | 91.92 | None | None | 90.45 ± 2.28 |
| **FedGPS(Ours)** | **94.20** | **65** | **2.3×** | **95.20** | **67** | **1.5×** | **95.29** | **49** | **2.0×** | **95.23** | **72** | **4.7×** | **95.08** | **39** | **1.7×** | **95.00 ± 0.45** |

*where $\mathcal{S}_{t-1} \setminus \{i\}$ is the set of clients in $\mathcal{S}_{t-1}$ excluding client $i$, and $|\mathcal{S}_{t-1} \setminus \{i\}|$ is its cardinality.*

By integrating this definition, the local client concurrently considers non-self gradient information before computing the update direction, as this information subtly conveys the underlying data distribution from others, which can be expressed as:

$$\hat{\mathbf{g}}_k^{t+1,e+1} = \nabla F_k(\boldsymbol{\theta}_k^{t+1,e} + \lambda_g \frac{\delta_{\boldsymbol{\theta}_k}^t}{\|\delta_{\boldsymbol{\theta}_k}^t\|}). \tag{7}$$

Here, $e$ represents the $e$-th local update iteration within a total of $E$ local epochs per round. The expression $\frac{\delta_{\boldsymbol{\theta}_k}^t}{\|\delta_{\boldsymbol{\theta}_k}^t\|}$ denotes a unit vector aligned with the direction of $\delta_{\boldsymbol{\theta}_k}^t$. We employ the $\lambda_g \frac{\delta_{\boldsymbol{\theta}_k}^t}{\|\delta_{\boldsymbol{\theta}_k}^t\|}$ instead of $\delta_{\boldsymbol{\theta}_k}^t$ to focus exclusively on the update direction from other clients, with the hyperparameter $\lambda_g$ providing adjustable scaling to optimize performance. Lastly, the local model $\boldsymbol{\theta}_k^{t+1,e+1}$ updated by the new rectified path as follows:

$$\boldsymbol{\theta}_k^{t+1,e+1} = \boldsymbol{\theta}_k^{t+1,e} - \eta_l \hat{\mathbf{g}}_k^{t+1,e+1}. \tag{8}$$

This term is deemed dynamic as the gradient path is adjusted at each update iteration. The local update direction is consistently refined using statistical gradient information from other clients.

## 5 Experiments

We organize this section as follows: (a) Detailed description of all the evaluated methods in our comprehensive evaluation (Sec 5.1); (b) The implementation and experimental settings we followed (Sec 5.2); (c) The main results and observations to demonstrate the efficacy of FedGPS (Sec 5.3); (d) Ablation study on two modules of FedGPS (Sec. 5.4).

Table 3: Top-1 accuracy of baselines and our method FedGPS with 5 different heterogeneous scenarios on CIFAR-100, heterogeneity degree $\alpha = 0.1$, local epochs $E = 1$ and total client number $K = 10$.

| | Dataset: CIFAR-100 Heterogeneity Level:$\alpha = 0.1$ Client Number:$K = 10$, Client Sampling Rate: 50% Total Communication Round:$T = 500$ Local Epochs:$E = 1$ | | | | | | | | | | | | | | | |
|---|---|---|---|---|---|---|---|---|---|---|---|---|---|---|---|---|
| Diff Scenario | Heterogeneous scenario 1 | | | Heterogeneous scenario 2 | | | Heterogeneous scenario 3 | | | Heterogeneous scenario 4 | | | Heterogeneous scenario 5 | | | |
| | Centralized Training Acc=78% | | | | | | | | | | | | | | | |
| | ACC↑ | ROUND↓ | SpeedUp↑ | ACC↑ | ROUND↓ | SpeedUp↑ | ACC↑ | ROUND↓ | SpeedUp↑ | ACC↑ | ROUND↓ | SpeedUp↑ | ACC↑ | ROUND↓ | SpeedUp↑ | |
| Methods | Target Acc=69% | | | Target Acc=69% | | | Target Acc=69% | | | Target Acc=70% | | | Target Acc=66% | | | Mean Acc± Std |
| FedAvg | 69.89 | 500 | 1.0× | 69.08 | 411 | 1.0× | 69.13 | 471 | 1.0× | 70.62 | 429 | 1.0× | 66.54 | 436 | 1.0× | 69.05 ± 1.54 |
| FedAvgM | 70.10 | 350 | 1.4× | 69.44 | 476 | 0.9× | 69.69 | 400 | 1.2× | 70.52 | 434 | 1.0× | 66.85 | 491 | 0.9× | 69.32 ± 1.44 |
| FedProx | 69.36 | 460 | 1.1× | 67.46 | None | None | 68.31 | None | None | 69.45 | None | None | 65.23 | None | None | 67.96 ± 1.73 |
| SCAFFOLD | 63.78 | None | None | 63.13 | None | None | 64.45 | None | None | 65.32 | None | None | 60.34 | None | None | 63.40 ± 1.90 |
| CCVR | - | - | - | - | - | - | - | - | - | - | - | - | - | - | - | - |
| VHL | 70.93 | 324 | 1.5× | 69.99 | 407 | 1.0× | 70.08 | 401 | 1.2× | 71.03 | 405 | 1.1× | 68.77 | 306 | 1.4× | 70.16 ± 0.91 |
| FedASAM | 70.04 | 350 | 1.4× | 68.95 | None | None | 69.32 | 389 | 1.2× | 70.74 | 434 | 1.0× | 66.52 | 428 | 1.0× | 69.11 ± 1.60 |
| FedExp | 69.72 | 413 | 1.2× | 69.00 | 476 | 0.9× | 69.61 | 428 | 1.1× | 70.43 | 433 | 1.0× | 65.31 | None | None | 68.81 ± 2.02 |
| FedDecorr | 68.91 | None | None | 68.88 | None | None | 68.11 | None | None | 70.19 | 458 | 0.9× | 62.93 | None | None | 68.38 ± 2.31 |
| FedDisco | 69.50 | 428 | 1.2× | 68.55 | None | None | 69.13 | 427 | 1.1× | 70.71 | 427 | 1.0× | 65.80 | None | None | 68.71 ± 1.63 |
| FedInit | 67.87 | None | None | 66.92 | None | None | 66.82 | None | None | 69.41 | None | None | 63.55 | None | None | 66.91 ± 2.15 |
| FedLESAM | 68.84 | None | None | 67.31 | None | None | 66.57 | None | None | 67.82 | None | None | 65.61 | None | None | 67.23 ± 1.23 |
| NUCFL | 68.29 | None | None | 67.94 | None | None | 65.47 | None | None | 67.81 | None | None | 64.44 | None | None | 66.79 ± 1.72 |
| **FedGPS(Ours)** | **71.14** | **336** | **1.5×** | **70.58** | **427** | **1.0×** | **70.50** | **374** | **1.3×** | **71.44** | **400** | **1.1×** | **69.79** | **292** | **1.5×** | **70.69 ± 0.64** |

## 5.1 Evaluated Details

**Compared Methods:** We evaluate the FL methods that alleviate data heterogeneity from different perspectives. 1) FedAvg [7] is the fundamental work in FL; 2) FedAvgM [51] accumulate model updates with momentum; 3) FedProx [11] constrain the divergence between local and global models; 4) SCAFFOLD [12] use extra term to correct the local gradients; 5) CCVR [13] share statistical logits to sample rectification data at the server side; 6) VHL [14] use virtual homogeneity data to constrain model by domain adaptation. 7) FedASAM [26] and FedLESAMS [29] use the insight of sharpness aware minimization; 8) FedExp [52] is inspired by Projection Onto Convex Sets (POCS) to select global step size adaptively; 9) FedDecorr [53, 54] constrain the feature covariance matrix due to the dimension collapse; 10) FedDisco [21] adjusts the aggregation weight based on discrepancy between clients; 11) FedInit [55] improves the local consistency by related initialization; 12) NUCFL [56] calibrates local classifier after local training.

**Datasets, Models and Metrics:** Following [3, 14, 57], we evaluate our method on three standard datasets: CIFAR-10, CIFAR-100 [58], and SVHN [59]. In line with prior work [57, 60], we use ResNet-18 for CIFAR-10 and SVHN, and ResNet-50 for CIFAR-100. We report three metrics related to communication efficiency and performance, building on previous work [14, 15]: (1) "ACC": the best accuracy achieved during training, with the target accuracy set as the best performance of FedAvg to provide a lower bound for evaluation; (2) "ROUND": the communication round required to reach the target accuracy; and (3) "SpeedUp": the speedup factor compared to FedAvg.

## 5.2 Implementation Details

**Federated Settings:** To simulate a heterogeneous data distribution across clients, we employ the Dirichlet partitioning method, a common approach in recent FL works [8, 57, 51]. This method draws client data proportions $\mathbf{q}$ from a Dirichlet distribution, $\mathbf{q} \sim \text{Dir}(\alpha\mathbf{p})$, where $\alpha$ is the concentration parameter that controls the degree of heterogeneity. We use $\alpha = 0.1$, but vary the random seed to generate multiple distinct heterogeneous data distributions. Examples of these distributions are shown in Fig. 2(a) and 2(b). We simulate cross-silo scenarios using 10 clients and cross-device scenarios using 100 clients. We set the sampling rate $\lambda_s$ as 50% for cross-silos and 10% for cross-devices scenario. We set local epochs $E = 1$ (results for different local epochs are shown in the Appendix D.3).

**Experimental Details:** To ensure a fair and direct comparison, all methods were evaluated under identical conditions, including the same data partitioning, sampling rate, local epochs, and communication rounds. We use the SGD optimizer with 0.01 learning rate and 0.9 momentum, 1e-5 weight decay (also denoted as $\lambda_3$). Among the hyperparameters, $\lambda_1$ and $\lambda_2$ were both set to 0.1, and $\lambda_g$ is fixed at 0.5 for the main experiments (Details can be seen in the Appendix C).

## 5.3 Main Results

Our evaluation results on CIFAR-10, SVHN, and CIFAR-100 are shown respectively in Tabs. 1, 2, 3 and 4. Additionally, if a method fails to produce valid results, e.g., NaN loss, we denote its

Table 4: Top-1 accuracy of baselines and our method FedGPS with 5 different heterogeneous scenarios on CIFAR-10, heterogeneity degree $\alpha = 0.1$, local epochs $E = 1$ and total client number $K = 100$.

| Dataset: CIFAR-10 Heterogeneity Level:$\alpha = 0.1$ Client Number:$K = 100$, Client Sampling Rate: 10% Total Communication Round:$T = 500$ Local Epochs:$E = 1$ | | | | | | | | | | | | | | | | |
|---|---|---|---|---|---|---|---|---|---|---|---|---|---|---|---|---|
| Diff Scenario | Heterogeneous scenario 1 | | | Heterogeneous scenario 2 | | | Heterogeneous scenario 3 | | | Heterogeneous scenario 4 | | | Heterogeneous scenario 5 | | | |
| Centralized Training Acc=xxx% | | | | | | | | | | | | | | | | |
| | ACC↑ | ROUND↓ | SpeedUp↑ | ACC↑ | ROUND↓ | SpeedUp↑ | ACC↑ | ROUND↓ | SpeedUp↑ | ACC↑ | ROUND↓ | SpeedUp↑ | ACC↑ | ROUND↓ | SpeedUp↑ | |
| Methods | Target Acc=48% | | | Target Acc=48% | | | Target Acc=57% | | | Target Acc=39% | | | Target Acc=39% | | | Mean Acc± Std |
| FedAvg | 48.22 | 449 | 1.0× | 48.23 | 452 | 1.0× | 57.61 | 482 | 1.0× | 39.96 | 479 | 1.0× | 39.41 | 498 | 1.0× | 46.69 ± 7.45 |
| FedAvgM | 58.80 | 303 | 1.5× | 60.07 | 363 | 1.2× | 66.40 | 414 | 1.2× | 46.89 | 291 | 1.6× | 45.21 | 432 | 1.2× | 55.47 ± 9.09 |
| FedProx | 52.84 | 357 | 1.3× | 54.18 | 364 | 1.2× | 63.04 | 481 | 1.0× | 44.11 | 370 | 1.3× | 42.90 | 432 | 1.2× | 51.41 ± 8.23 |
| SCAFFOLD | 60.17 | 202 | 2.2× | 62.34 | 158 | 2.9× | 58.24 | 335 | 1.4× | 60.75 | 44 | 10.9× | 60.90 | 37 | 13.5× | 60.48 ± 1.49 |
| CCVR | 64.06 | 69 | 6.5× | 68.93 | 76 | 5.9× | 62.63 | 291 | 1.7× | 62.82 | 38 | 12.6× | 61.73 | 31 | 16.1× | 64.03 ± 2.86 |
| VHL | 72.70 | 128 | 3.5× | 70.21 | 201 | 2.2× | 76.12 | 235 | 2.1× | 68.18 | 143 | 3.3× | 62.44 | 129 | 3.9× | 69.93 ± 5.13 |
| FedASAM | 46.35 | None | None | 45.32 | None | None | 54.35 | None | None | 41.50 | 478 | 1.0× | 33.62 | None | None | 44.23 ± 7.55 |
| FedExp | 38.26 | None | None | 46.76 | None | None | 56.61 | None | None | 43.33 | 367 | 1.3× | 37.55 | None | None | 44.50 ± 7.75 |
| FedDecorr | 63.69 | 303 | 1.5× | 66.58 | 268 | 1.7× | 69.92 | 337 | 1.4× | - | - | - | - | - | - | 66.73 ± 3.12 |
| FedDisco | - | - | - | - | - | - | - | - | - | - | - | - | - | - | - | - |
| FedInit | 71.01 | 130 | 3.5× | 72.09 | 138 | 3.3× | 75.76 | 336 | 1.4× | 62.96 | 90 | 5.3× | 67.38 | 88 | 5.7× | 69.84 ± 4.87 |
| FedLESAM | 72.64 | 110 | 4.1× | 75.48 | 146 | 3.1× | 75.47 | 234 | 2.1× | 77.56 | 75 | 6.4× | 73.73 | 75 | 6.6× | 74.98 ± 1.88 |
| NUCFL | 53.72 | 323 | 1.4× | 52.85 | 297 | 1.5× | 53.80 | None | None | 49.47 | 231 | 2.1× | 46.17 | 356 | 1.4× | 51.20 ± 3.32 |
| **FedGPS(Ours)** | **78.32** | **102** | **4.4×** | **76.97** | **155** | **2.9×** | **76.27** | **232** | **2.1×** | **78.12** | **94** | **5.1×** | **75.53** | **76** | **6.6×** | **77.04 ± 1.19** |

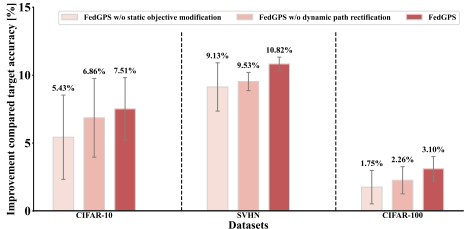

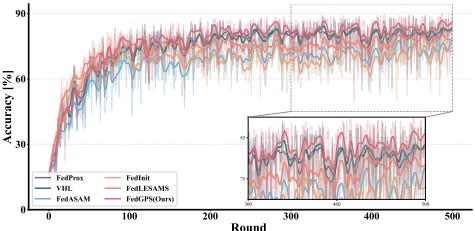

(a) Ablation results on different part of FedGPS.  (b) Test accuracy and convergence rate on different baselines and FedGPS.

Figure 4: The visualization of the ablation study and convergence of FedGPS compared with other baselines. Due to the large volume of baselines, we select the top 5 baselines to plot.

performance as "-" in our results. Based on our experiments, we outline several key observations that highlight the characteristics of the evaluated methods and provide insights for future research:

**Observation 1: Absence of SOTA Methods Across Scenarios.** The results indicate that most methods exhibit limited robustness across various scenarios. For instance, VHL demonstrates superior performance compared to other methods under the setting $\alpha = 0.1, K = 10, \lambda_s = 50\%$. However, on the same dataset with $\alpha = 0.1, K = 100, \lambda_s = 10\%$, its performance degrades significantly. Similar patterns are observed in other methods, suggesting that the performance of a given method can vary substantially across different settings.

**Observation 2: Value of global classifier calibration.** Global classifier calibration proves to be effective in certain contexts. For example, CCVR employs logits-based statistical information sampled from a Gaussian distribution to calibrate the classifier ($\bm{h}$) globally. This approach reduces the number of communication rounds required to achieve the target accuracy on specific datasets, also stabilizes the training curve. As shown in Tab. 4, under certain distributions, CCVR achieves the target accuracy with fewer communication rounds compared to our method, despite lower overall performance. This observation inspires future research to enhance performance using such techniques.

**Observation 3: Performance variability across settings.** The performance of methods varies significantly across different settings, indicating a need for improved adaptability or meticulous hyperparameter tuning. For instance, FedASAM and FedExp outperform vanilla FedAvg under $\alpha = 0.1, K = 10, \lambda_s = 50\%$, but struggle to surpass FedAvg under $\alpha = 0.1, K = 100, \lambda_s = 10\%$. Similarly, many methods achieve performance comparable to or worse than FedAvg on CIFAR-100, underscoring the challenge of generalizing across diverse datasets and configurations.

Our experimental results demonstrate that FedGPS consistently achieves state-of-the-art (SOTA) performance across diverse federated learning settings and datasets. As reported in Tab. 1, FedGPS surpasses the best baseline methods under various heterogeneous scenarios. However, its performance gains on SVHN are modest, as vanilla FedAvg already approximates centralized training perfor-

mance in these scenarios, limiting the potential for improvement in distributed settings. In more challenging environments, such as those detailed in Tab. 3, FedGPS exhibits substantially greater improvements. Crucially, FedGPS prioritizes robustness across heterogeneous data partitions over optimizing for specific scenarios' performance, a design choice that enhances its generalization ability. The convergence rates of different evaluated methods and FedGPS are shown in Fig. 4(b).

### 5.4 Ablation Study on Two Perspectives

To evaluate the individual contributions of the static goal-oriented objective function and the dynamic path-oriented gradient rectification in FedGPS to FL performance, we conduct ablation studies by equipping FedAvg with each module in isolation. As shown in Fig. 4(a), the results report relative performance improvements over the target accuracy of vanilla FedAvg. Specifically, FedGPS without the static objective modification relies exclusively on dynamic path rectification, whereas FedGPS without dynamic path rectification employs only the static objective function. These experiments confirm the distinct effectiveness of each module. Notably, the synergistic integration of both modules yields superior performance across diverse heterogeneous scenarios. Additional ablation studies, exploring varying numbers of clients, datasets, and local epochs, are detailed in the Appendix D. Furthermore, to assess the robustness of FedGPS under different training seeds, we initialize the model with three different random seeds under identical settings and data distribution. Its comprehensive results are provided in the experimental section of the Appendix D.8.

## 6 Conclusion and Further Discussion

In this work, we explore an important and overlooked question: how well do existing notable algorithms perform in multiple heterogeneous scenarios? Extensive experiments show that most of the existing algorithms are limited in robustness, which inspired the **FedGPS** framework. It combines two orthogonal views to achieve label-distribution-agnostic robustness by considering the statistical information of the client from the distribution level and the gradient view, respectively. More analysis about the communication and privacy of FedGPS are listed in the Appendix B. It also catalyzes future research into distribution-agnostic algorithms, paving the way for resilient federated learning in complex, real-world settings.

**Limitations:** Limited computational resources may constrain the performance of FedGPS, as FedGPS relies on additional surrogate data for its distribution alignment process. Future work could investigate more efficient alignment techniques that minimize the need for surrogate data or explore alternative approaches to enhance scalability. Furthermore, FedGPS does not yet address challenges posed by heterogeneous data features, necessitating further research into the robustness of its distribution and gradient collaboration framework across a broader range of heterogeneous FL scenarios, with the goal of achieving distribution-agnostic robustness.

## Broader impacts

Federated learning (FL), defined by its distributed data collection and keeping data locally, inherently navigates complex real-world applications driven by diverse tasks and participants. This complexity has spurred extensive exploration of varied federated settings. Our work tackles a critical challenge in FL: **the pervasive data heterogeneity that undermines the robustness of existing methods across diverse data distributions.** Through over **1100+ groups of experiments**, we investigate mitigation strategies from multiple perspectives, introducing novel insights that significantly enhance robustness. We also provide key observations to guide future research and inform the selection of federated methods for heterogeneous scenarios. **Rather than advocating for a single algorithm tailored to a specific scenario, we emphasize the need for broader, actionable insights to support practical FL deployments, enabling customized solutions for diverse applications.** This paper marks a pivotal step toward distribution-agnostic federated learning, establishing a foundation for robust, scalable, and adaptable FL systems. By bridging experimentation with practical applicability, our contributions aim to catalyze transformative advancements in this rapidly evolving field and future real applications with sensitive data protection.

## Acknowledgements

We thank all the reviewers for their constructive suggestions and dedication to this paper. ZQY, CXL, and YXY were supported by Hong Kong Innovation and Technology Commission Innovation and Technology Fund ITS/229/22. YGZ was funded by Inno HK Generative AI R&D Center. BH was supported by NSFC General Program No. 62376235 and RGC General Research Fund No. 12200725. YMC was supported by the Hong Kong Baptist University (HKBU) under grant RC-FNRA-IG/23-24/SCI/02, and the seed funding for collaborative research grants RC-SFCRG/23-24/R2/SCI/10.

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

# Appendix

# A   Proof Results

**Assumption A.1** (Lipschitz Continuity). *For a local feature extractor $f : \mathcal{X} \to \mathcal{Z}$ parameterized by $\psi$ is L-Lipschitz continuous, that is,*

$$\|f_{\psi_k}(\mathbf{x}) - f_{\psi'_k}(\mathbf{x})\|_{\mathcal{Z}} \le L\|\psi_k - \psi'_k\|_{\psi}, \text{where } k \in \mathcal{S},$$

*for all $\mathbf{x}_k \in \mathcal{X}$, where $\psi, \psi' \in \mathbb{R}^{|\psi|}$. Moreover, $\|\cdot\|_{\mathcal{Z}}$ and $\|\cdot\|_{\psi}$ are norms on the feature and parameter spaces, respectively.*

*Remark* 2. This assumption ensures that small changes in the parameters of the local feature extractor lead to proportionally small changes in the extracted features. Specifically, for any client $k$, if the parameters $\psi$ are slightly modified to $\psi'$, the resulting feature representations remain close in the feature space $\mathcal{Z}$, with the difference bounded by the Lipschitz constant $L$.

**Lemma A.2.** *For random vectors $\mathbf{v}_1, \mathbf{v}_2, \cdots \mathbf{v}_n$, we have*

$$\|\sum_{t=1}^{T} \mathbf{v}_t\|_2 \le \sum_{t=1}^{T} \|\mathbf{v}_t\|_2.$$

## A.1   Proof of Necessary Lemma A.3

**Lemma A.3** (Bounded difference between global and local feature extractor $\psi, \psi_k$). *In FL with $K$ clients, where each round $t$ samples a subset of clients $\mathcal{S}_t \subseteq \mathcal{S}$, and the global feature extractor parameters are updated as $\psi^{t+1} = \frac{1}{|\mathcal{S}_t|} \sum_{k \in \mathcal{S}_t} \psi_k{}^{t+1}$, with local parameters $\psi_k$ updated via bounded optimization, there exists $\Delta_d > 0$ such that:*

$$\|\psi_k - \psi\|_2 \le \Delta_d,$$

*for all $k \in \mathcal{S}$, where $\|\cdot\|_2$ is the Euclidean norm on the parameter space.*

*Proof.* Local parameters $\psi_k$ are updated using optimization (e.g., SGD) with regularization or gradient clipping, ensuring bounded norms. At round $t$, the global parameters are:

$$\psi^{t+1} = \frac{1}{|\mathcal{S}_t|} \sum_{k \in \mathcal{S}_t} \psi_k{}^{t+1}.$$

Consider the parameter difference for any client $k \in \mathcal{S}$, not necessarily in $\mathcal{S}_t$:

$$\|\psi_k{}^{t+1} - \psi^{t+1}\|_2 = \left\|\psi_k{}^{t+1} - \frac{1}{|\mathcal{S}_t|} \sum_{j \in \mathcal{S}_t} \psi_j{}^{t+1}\right\|_2 = \left\|\frac{1}{|\mathcal{S}_t|} \sum_{j \in \mathcal{S}_t} (\psi_k{}^{t+1} - \psi_j{}^{t+1})\right\|_2$$

$$\overset{(a)}{\le} \sum_{j \in \mathcal{S}_t} \|\frac{1}{|\mathcal{S}_t|} \psi_k{}^{t+1} - \psi_j{}^{t+1}\|_2 \overset{(b)}{=} \frac{1}{|\mathcal{S}_t|} \sum_{j \in \mathcal{S}_t} \|\psi_k{}^{t+1} - \psi_j{}^{t+1}\|_2,$$

where $(a)$ is from Lemma A.2 and $(b)$ is because $\|a\mathbf{v}\|_2 = a\|\mathbf{v}\|_2$.

Assume optimization bounds the parameter norm: $\|\psi_k{}^{t+1}\|_2 \le B$, for some $B > 0$, across all rounds and clients (achieved via regularization). Then:

$$\|\psi_k{}^{t+1} - \psi_j{}^{t+1}\|_2 \le \|\psi_k{}^{t+1}\|_2 + \|\psi_j{}^{t+1}\|_2 \le 2B.$$

Thus:

$$\|\psi_k{}^{t+1} - \psi^{t+1}\|_2 \le \frac{1}{|\mathcal{S}_t|} \sum_{j \in \mathcal{S}_t} 2B = 2B.$$

This bound holds for all $k \in \mathcal{S}$, as the maximum difference is independent of whether $k \in \mathcal{S}_t$. Set $\Delta_d = 2B$, so:

$$\|\psi_k - \psi\|_2 \le \Delta_d,$$

which completes the proof of Lemma. A.3. $\qquad\square$

## A.2 Proof of Necessary Lemma A.4

**Lemma A.4.** *In a FL with $K$ clients, let the global feature extractor $\psi$ have parameters $\psi^{t+1} = \frac{1}{|\mathcal{S}_t|} \sum_{k \in \mathcal{S}_t} \psi_k{}^{t+1}$, where $\psi_k$ are parameters of local feature extractors. Let $\mathcal{P}_{k,\mathcal{D}}$ and $\mathcal{P}_{g,\mathcal{D}}$ denote the feature distributions induced by $\psi_k$ and $\psi$ on a certain data distribution $\mathcal{D}$. Given Assumption A.1 and Lemma A.3, there exists $\kappa = L\Delta_d$, such that:*

$$W_1(\mathcal{P}_{k,\mathcal{D}}, \mathcal{P}_{g,\mathcal{D}}) \leq \kappa,$$

*where $W_1$ is the Wasserstein-1 distance in the feature space $\mathcal{Z}$.*

*Proof.* The Wasserstein-1 distance is:

$$W_1(\mathcal{P}_{k,\mathcal{D}}, \mathcal{P}_{g,\mathcal{D}}) = \inf_{\gamma \in \Pi(\mathcal{P}_{k,\mathcal{D}}, \mathcal{P}_{g,\mathcal{D}})} \int \|z - z'\|_{\mathcal{Z}} \, d\gamma(z, z').$$

For $x \sim \mathcal{D}$, by Assumption A.1, with $f : \mathcal{X} \to \mathcal{Z}$ parameterized by local feature extractor $\psi_k$ and global feature extractor $\psi$, respectively:

$$\|\psi_k(x) - \psi(x)\|_{\mathcal{Z}} = \|\psi(x; \psi_k{}^{t+1}) - \psi(x; \psi^{t+1})\|_{\mathcal{Z}} \leq L\|\psi_k{}^{t+1} - \psi^{t+1}\|_2.$$

By Lemma A.3, $\|\psi_k{}^{t+1} - \psi^{t+1}\|_2 \leq \Delta_d$. Thus:

$$\|\psi_k(x) - \psi(x)\|_{\mathcal{Z}} \leq L\Delta_d.$$

Define a coupling $\gamma$ where $z_k = \psi_k(x)$, $z = \psi(x)$, with probability $\mathcal{D}(x)$ and marginals:

- First: $\int_z \gamma(z_k, z) = \mathcal{D}(x : \psi_k(x) = z_k) = \mathcal{P}_{k,\mathcal{D}}$.

- Second: $\int_z \gamma(z, z) = \mathcal{D}(x : \psi(x) = z) = \mathcal{P}_{g,\mathcal{D}}$.

The cost is:

$$\int \|z_k - z\|_{\mathcal{Z}} \, d\gamma(z_k, z) \leq L\Delta_d.$$

Thus:

$$W_1(\mathcal{P}_{k,\mathcal{D}}, \mathcal{P}_{g,\mathcal{D}}) \leq \kappa, \quad \kappa = L\Delta_d,$$

which completes the proof of Lemma A.4. $\qquad\qquad\square$

## A.3 Proof of Theorem 4.1

**Theorem 4.1.** Given the global feature distribution $\mathcal{P}_g$, the local feature distribution $\mathcal{P}_k$, the surrogate distributions $\mathcal{P}^s$ (global) and $\mathcal{P}_k^s$ (local for the $k$-th client) over their corresponding data space. Suppose there exists $\kappa \geq 0$ such that $W_1(\mathcal{P}_{k,\mathcal{D}}, \mathcal{P}_{g,\mathcal{D}}) \leq \kappa$ under distribution $\mathcal{D}$. If the following conditions hold:

$$W_1(\mathcal{P}_k^s, \mathcal{P}_k) \leq \epsilon_1, \quad W_1(\mathcal{P}_k^s, \mathcal{P}^s) \leq \epsilon_2,$$

where $W_1$ is the Wasserstein-1 distance as defined in Definition 3.1. then the Wasserstein-1 distance between $\mathcal{P}_g$ and $\mathcal{P}^s$ is bounded as:

$$W_1(\mathcal{P}_g, \mathcal{P}^s) \leq \epsilon_1 + \epsilon_2 + \kappa.$$

*Proof.* We prove $W_1(\mathcal{P}_g, \mathcal{P}^s) \leq \epsilon_1 + \epsilon_2 + \kappa$ in each round $t$, where a subset of clients $\mathcal{S}_t \subseteq \mathcal{S}$ is sampled.

The Wasserstein-1 distance is:

$$W_1(\mathcal{P}, \mathcal{Q}) = \inf_{\gamma \in \Pi(\mathcal{P}, \mathcal{Q})} \int \|z - z'\|_{\mathcal{Z}} \, d\gamma(z, z').$$

First, we prove the bounded local private feature distribution to the global surrogate distribution distance. For each client $k \in \mathcal{S}$ (not necessarily in the sampled subset $\mathcal{S}_t$), we aim to bound the Wasserstein-1 distance between the local feature distribution $\mathcal{P}_k$ and the global surrogate feature distribution $\mathcal{P}^s$. To do so, we introduce the local surrogate feature distribution $\mathcal{P}_k^s$ as an intermediate

distribution and apply the triangle inequality for the Wasserstein-1 distance. The triangle inequality states that for any three probability distributions, we have:

$$W_1(\mathcal{P}_k, \mathcal{P}^s) \leq W_1(\mathcal{P}_k, \mathcal{P}_k^s) + W_1(\mathcal{P}_k^s, \mathcal{P}^s).$$

This inequality decomposes the distance between $\mathcal{P}_k$ and $\mathcal{P}^s$ into two segments: the distance from the local true feature distribution $\mathcal{P}_k$ to the local surrogate $\mathcal{P}_k^s$, and the distance from the local surrogate $\mathcal{P}_k^s$ to the global surrogate $\mathcal{P}^s$. Now, we have (1) The condition $W_1(\mathcal{P}_k^s, \mathcal{P}_k) \leq \epsilon_1$ implies, by the symmetry of the Wasserstein-1 distance ($W_1(\mathcal{P}, \mathcal{Q}) = W_1(\mathcal{Q}, \mathcal{P})$), that:

$$W_1(\mathcal{P}_k, \mathcal{P}_k^s) = W_1(\mathcal{P}_k^s, \mathcal{P}_k) \leq \epsilon_1.$$

This bound measures the alignment quality between the true local features and the surrogate features for client $k$, reflecting how well the surrogate data approximates the true data in the feature space. (2) The condition $W_1(\mathcal{P}_k^s, \mathcal{P}^s) \leq \epsilon_2$ directly provides:

$$W_1(\mathcal{P}_k^s, \mathcal{P}^s) \leq \epsilon_2.$$

This bound measures the consistency between the local surrogate features and the global surrogate features, reflecting the uniformity of surrogate representations across clients. Substitute these bounds into the triangle inequality:

$$W_1(\mathcal{P}_k, \mathcal{P}^s) \leq W_1(\mathcal{P}_k, \mathcal{P}_k^s) + W_1(\mathcal{P}_k^s, \mathcal{P}^s) \leq \epsilon_1 + \epsilon_2.$$

Thus, we have:

$$W_1(\mathcal{P}_k, \mathcal{P}^s) \leq \epsilon_1 + \epsilon_2. \tag{9}$$

This bound holds for all $k \in \mathcal{S}$, as the given conditions apply to each client, and the global surrogate distribution $\mathcal{P}^s = \frac{1}{K} \sum_{k=1}^{K} \mathcal{P}_k^s$ is defined over all clients.

Additionally, we denote the $\mathcal{P}_{g,k}$ as the feature distribution on the local data distribution $\mathcal{D}_k$ by the global feature extractor $\psi$. Then, we build a connection between $\mathcal{P}_{g,k}$ and $\mathcal{P}^s$. By Lemma A.4 and Eq.(9), we have:

$$W_1(\mathcal{P}_{g,k}, \mathcal{P}^s) \leq W_1(\mathcal{P}_{g,k}, \mathcal{P}_k) + W_1(\mathcal{P}_k, \mathcal{P}^s) \leq \kappa + \epsilon_1 + \epsilon_2. \tag{10}$$

Lastly, since $\mathcal{P}_g$ is the feature distribution of $\psi$ over the global data distribution, equivalent to the average of $\mathcal{P}_{g,k}$, construct $\gamma = \frac{1}{K} \sum_{k=1}^{K} \gamma_k$, where $\gamma_k \in \Pi(\mathcal{P}_{g,k}, \mathcal{P}^s)$. Marginals:

- First: $\int_{z'} \gamma(z, z') = \frac{1}{K} \sum_{k=1}^{K} \mathcal{P}_{g,k} = \mathcal{P}_g$.

- Second: $\int_z \gamma(z, z') = \frac{1}{K} \sum_{k=1}^{K} \mathcal{P}^s = \mathcal{P}^s$.

The cost is:

$$\int \|z - z'\|_Z \, d\gamma(z, z') \leq \frac{1}{K} \sum_{k=1}^{K} W_1(\mathcal{P}_{g,k}, \mathcal{P}^s) \leq \kappa + \epsilon_1 + \epsilon_2.$$

Thus:

$$W_1(\mathcal{P}_g, \mathcal{P}^s) \leq \epsilon_1 + \epsilon_2 + \kappa,$$

which completes the proof of Theorem 4.1. $\qquad\square$

## B   More Facts about FedGPS

In this section, we outline some facts of FedGPS , including the communication overheads (Sec. B.1) and privacy issue of FedGPS(Sec.B.2). Furthermore, we also give a brief introduction about the meaning of Nemenyi post-hoc test (Sec. B.3). We also provide a theoretical justification of dynamic path-oriented rectification (Sec. B.4).

Table 5: Detailed description of the two consecutive upload and download rounds, along with the associated local computational requirements.

| Process | FedAvg | FedGPS |
|---|---|---|
| **Global aggregation at round $t-1$** | Update global model $\boldsymbol{\theta}^t = \sum \boldsymbol{\theta}_{k,E}^{t-1}$ | 1. Update global model $\boldsymbol{\theta}^t = \boldsymbol{\theta}^{t-1} + \eta_g \sum_{k \in \mathcal{S}_{t-1}} \Delta_k^{t-1}$. 2. Update global surrogate prototypes $\mathcal{E}^c = \sum \mathcal{E}_k^c$. |
| **Explanation** | Apart from the direct aggregation parameters, e.g., FedAvg-like. The $\Delta$ of client parameters has also been widely used in many studies. | |
| **Server** | Select subset $\mathcal{S}_t$ to participate Round $t$ | |
| **Round $t$ selected $\mathcal{S}_t$ download from server** | Global model $\boldsymbol{\theta}^t$ (# Comm $M$) | 1. Global model $\boldsymbol{\theta}^t$; 2. Global model update information $\Delta \boldsymbol{\theta}^t = \boldsymbol{\theta}^t - \boldsymbol{\theta}^{t-1} = \eta_g \sum_{k \in \mathcal{S}_{t-1}} \Delta_k^{t-1}$ (# Comm $2M + C*512$) |
| **Explanation** | $\Delta \boldsymbol{\theta}^t$ contains the gradient aggregation information updated by the selected client in the previous round $t-1$. The global surrogate distribution is represented by a prototype for each class. The prototype for each class is a 512-dimensional vector. | |
| **Local operation** | Update local model $\boldsymbol{\theta}_{k,0}^t = \boldsymbol{\theta}^t$ (0 means the model without local epochs training) | 1. Update local model $\boldsymbol{\theta}_{k,0}^t = \boldsymbol{\theta}^t$; 2. Compute Non-Self Gradient based on $\Delta \boldsymbol{\theta}^t$. |
| **Local extra operation explanation** | **A:** $\delta_{\boldsymbol{\theta}}^t = \Delta \boldsymbol{\theta}^t - \Delta_k^{t-1}$ If this client is selected last round which means we should distract its local update $\Delta_k^{t-1}$ of last round (this is kept locally). **B:** $\delta_{\boldsymbol{\theta}}^t = \Delta \boldsymbol{\theta}^t$ If this client is not selected last round which means $\Delta \boldsymbol{\theta}^t$ contains all other client's gradient information. | |
| **Local training** | Traditional SGD uses the corresponding loss function and local data | SGD use Eq. (6) in FedGPS with local data |
| **Explanation** | Incorporating gradient information from other clients only occurs by adding the parameters together before each gradient descent. This additional computation overhead is almost negligible and can be disregarded. There are a total of iteration times of parameter summation operations. (Negligible additional computation expenses) | |
| **Round $t$ selected $\mathcal{S}_t$ upload to server** | New local model parameters $\boldsymbol{\theta}_{k,E}^t$ (# Comm $M$) | 1. Local updated parameters $\Delta_k^t = \boldsymbol{\theta}_{k,E}^t - \boldsymbol{\theta}_{k,0}^t$ 2. Compute local surrogate prototypes $\mathcal{E}_k^c = \frac{1}{\|\mathcal{D}_s^c\|} \sum \psi_k(\xi_s^c)$ (# Comm $M + C*512$) |
| **Global aggregation at round $t$** | $\boldsymbol{\theta}^{t+1} = \sum \boldsymbol{\theta}_{k,E}^t$ | $\boldsymbol{\theta}^{t+1} = \boldsymbol{\theta}^t + \eta_g \sum_{k \in \mathcal{S}_t} \Delta_k^t$ |

We denote the whole model size as $M$ and the total classes of the dataset as $C$, e.g.,$C = 10$ for CIFAR-10.

Table 6: The performance comparison between FedGPS-CF and FedGPS under Heterogeneous scenario 1 with CIFAR-10.

| | $K = 10, \lambda_s = 50\%, R = 500, E = 1$ | $K = 10, \lambda_s = 50\%, R = 200, E = 5$ | $K = 100, \lambda_s = 10\%, R = 500, E = 1$ |
|---|---|---|---|
| FedGPS-CF | 90.01 | 88.13 | 78.07 |
| FedGPS | 90.31 | 88.47 | 78.32 |

## B.1 Communication Analysis of FedGPS

In this subsection, we give a detailed communication overhead analysis regarding FedGPS. In summary, there is one additional model (containing the aggregated gradient) of the same size as the global model during the download stage, while no extra communication overhead during upload from the client to the server side. Thus, FedGPS brings about 1.5 times the communication overhead than vanilla methods, e.g., FedAvg. Specifically, we have detailed the computational costs of the server and client, as well as the communication costs of download and upload between two adjacent rounds, and explained the reasons for these additional costs of FedGPS in Tab. 5. The extra $C*512$ (where $C*512 \ll M$, e.g., 0.05% in CIFAR-10) is the local uploaded local surrogate distribution prototypes and download global surrogate distribution prototype.

We also tried a communication-friendly version of FedGPS, which was denoted as FedGPS-CF. Specifically, when uploading the model, FedGPS-CF, like FedGPS, only has a communication cost of $M + C*512$. When downloading from the server, it still only downloads the global model and global surrogate prototypes, meaning the communication cost remains $M + C*512$. Here, we use the difference between the global model downloaded in two rounds from the server to represent the gradient aggregation information of other clients. Similarly, if a client is selected in both adjacent rounds, its own update information should be removed. Finally, FedGPS-CF achieves comparable performance to FedGPS; the results are listed in Tab. 6. Moreover, it reduces the download overhead of $M$, making FedGPS-CF only have an additional communication cost of $C*512$ compared to FedAvg, which is negligible.

## B.2 Privacy Analysis of FedGPS

The two technologies of FedGPS do not transmit any additional client's information to any other client compared to traditional methods. On the contrary, FedGPS replaces raw data prototypes with surrogate prototypes instead, thereby further protecting privacy compared to previous works [31, 61]. We list the privacy clarification among surrogate distribution and gradient information in the following:

- **Regarding the surrogate distribution privacy issue:** Because the surrogate is sampled from different Gaussian distributions, it does not contain any information related to the local data. Further, we transmit the aggregated class-wise prototypes of surrogate data. After aggregating all the high-dimensional embeddings of each class, the surrogate data information is further protected. Many papers [31, 61] also transmit using the original data prototypes, which is a weaker level of information protection than FedGPS.
- **Regarding the gradient information privacy issue:** FedGPS doesn't transmit the gradient information of a certain client to any other client. Every client only upload its own information to server and download the aggregated information from the server. This process is the same as most of other federated methods in uploading and downloading the aggregated information (Detailed information can be referred at the Tab. 5).

## B.3 Brief introduction of Nemenyi post-hoc test method

The Nemenyi post-hoc test in Fig. 1 is a non-parametric statistical method used for pairwise comparisons of multiple groups [62] (e.g., algorithms or models) after a significant result from a Friedman test (a non-parametric analog to repeated-measures ANOVA). It ranks the performance of each method across multiple independent runs or datasets and computes a "critical distance" (CD) threshold. If the average rank difference between two methods exceeds the CD, their performances are considered statistically significantly different at a given significance level (typically $\alpha$=0.05). The test is conservative and accounts for multiple comparisons to control the family-wise error rate, making it robust for scenarios like ours, where we evaluate algorithm robustness across diverse heterogeneity partitions (e.g., different random seeds for Dirichlet distributions). In Fig. 1, the Nemenyi post-hoc test assesses the robustness of baseline methods across different heterogeneity scenarios. The results show overlapping CD intervals for most baselines, indicating no statistically significant performance differences among them. This finding highlights the need for our proposed approach, as existing methods exhibit limited adaptability to varied data distributions. Furthermore, the Nemenyi test has been widely adopted in holistic evaluations [63, 64, 65] and federated learning scenarios [66, 67].

## B.4 Theoretical Justification of Dynamic Path-oriented Rectification

The insight behind incorporating information from other clients in FedGPS stems from works like SCAFFOLD [12] and other related research [18], which also use other client information as a control variate to adjust update direction. Beyond this intuition, we provide a theoretical justification using a Taylor expansion to demonstrate that integrating other clients' information can indeed further decrease the deviation between local and global update directions.

We denote local loss function as $f_k(\cdot) : \mathbb{R}^d \to \mathbb{R}$ and global loss function $F(\cdot)$. First of all: The original local update use the vanilla gradient descent on the local model $\boldsymbol{\theta}_k$ is denoted as $g_{\text{old}} = \nabla f_k(\boldsymbol{\theta}_k)$. For FedGPS, we incorporate non-self gradient $\delta_{\boldsymbol{\theta}_k}$ to local model , we denote $\frac{\delta_{\boldsymbol{\theta}_k}}{||\delta_{\boldsymbol{\theta}_k}||}$ as $g'_k$:

$$\boldsymbol{\theta}'_k = \boldsymbol{\theta} + \lambda_g g'_k.$$

Then we get a new update direction computed based on the new model parameters $\boldsymbol{\theta}'_k$:

$$g_{\text{new}} = \nabla f_k(\boldsymbol{\theta}'_k).$$

For a continuously twice-differentiable function $f_k(\boldsymbol{\theta})$, the gradient function $\nabla f_k(\boldsymbol{\theta})$ expands around point $\boldsymbol{\theta}$ along direction $g'_k$:

$$\nabla f_k(\boldsymbol{\theta}_k + \lambda_g g'_k) \approx \nabla f_k(\boldsymbol{\theta}_k) + \nabla^2 f_k(\boldsymbol{\theta}_k)(\lambda_g g'_k) + R_3,$$

where $R_3$ represents higher-order terms that can be neglected and $\nabla^2 f_k$ is the Hessian at $\boldsymbol{\theta}_k$. Thus we can get:

$$\nabla f_k(\boldsymbol{\theta}'_k) \approx \nabla f_k(\boldsymbol{\theta}_k) + \lambda_g \nabla^2 f_k(\boldsymbol{\theta}_k) g'_k.$$

Here, we assume that the loss function is convex. So the Hessian $\bar{H} = \nabla^2 f_k$ is positive semi-definite. Ideally, we assume non-self gradients contain all the gradients from other clients; we can denote $\delta_{\boldsymbol{\theta}_k} = \nabla F(\boldsymbol{\theta}) - \nabla f_k(\boldsymbol{\theta}_k)$. Substitute into the expansion:

$$\nabla f_k(\boldsymbol{\theta}_k') \approx \nabla f_k(\boldsymbol{\theta}_k) + \lambda_g \bar{H}(\nabla F(\boldsymbol{\theta}) - \nabla f_i(\boldsymbol{\theta})) = (I - \lambda_g \bar{H})\nabla f_k(\boldsymbol{\theta}_k) + \lambda_g \bar{H}\nabla F(\boldsymbol{\theta}),$$

where $I$ is the identity matrix. As a result: - The original bias between local and global gradient: $d_0 = \|g_{\text{old}} - \nabla F(\boldsymbol{\theta})\|$ - Refined update direction bias between new model parameters $\boldsymbol{\theta}_k'$ and global gradient: $d' = \|g_{\text{new}} - \nabla F(\boldsymbol{\theta})\| \approx \|(I - \lambda_g \bar{H})(\nabla f_k(\boldsymbol{\theta}) - \nabla F(\boldsymbol{\theta}))\| \leq \|I - \lambda_g \bar{H}\| \cdot d_0$. If $\lambda_g$ is tuned to make $\|I - \lambda_g \bar{H} < 1\|$, then $d' < d_0$, which reduces the shift.

In practice, direct access to all client gradient information is often limited due to privacy and communication overhead. Nevertheless, through careful hyperparameter tuning, FedGPS consistently achieves state-of-the-art (SOTA) performance across various heterogeneous scenarios.

## C  More Experimental Details

In this section, we present a comprehensive overview of our experimental implementation and process. First, we visualize various data distributions across diverse heterogeneous scenarios (Sec. C.1). Next, we provide the hyperparameters for both the baselines and FedGPS (Sec. C.2). Additionally, this section includes the process and pseudocode for FedGPS (Sec. C.3).

### C.1  More Data Distribution

We present a detailed visualization of different data distribution across various datasets and client numbers with the same heterogeneity degree $\alpha = 0.1$. A heatmap visualizes the distribution, with darker colors indicating higher quantities for the corresponding label.

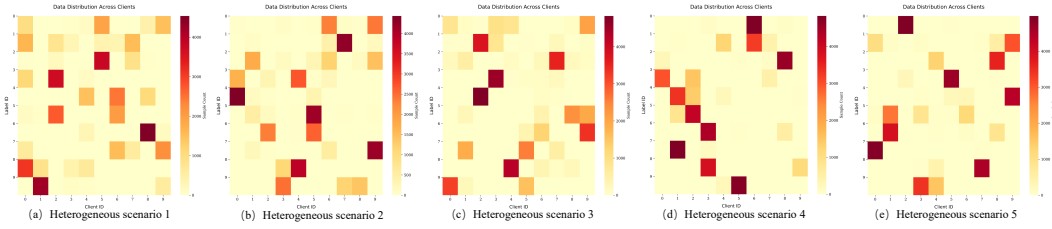

Figure 5: The visualization of the CIFAR-10 dataset distribution across $K = 10$ clients under five different heterogeneous scenarios.

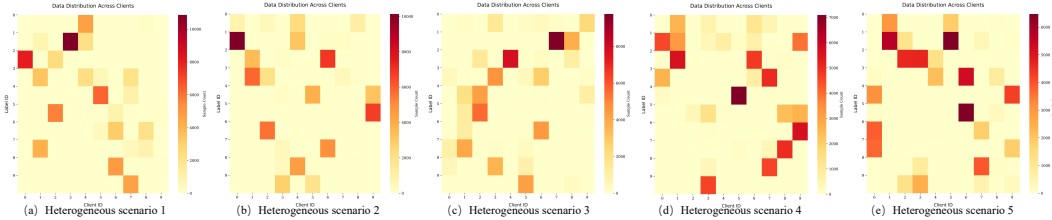

Figure 6: The visualization of the SVHN dataset distribution across $K = 10$ clients under five different heterogeneous scenarios.

The Figs 5, 6, 7, 8, 9 and 10 demonstrate that, despite using the same dataset and degree of heterogeneity, the data distributions across different scenarios vary significantly. This variation can lead to differing performances of the same algorithm. Therefore, the algorithm's robustness across various heterogeneous scenarios is crucial.

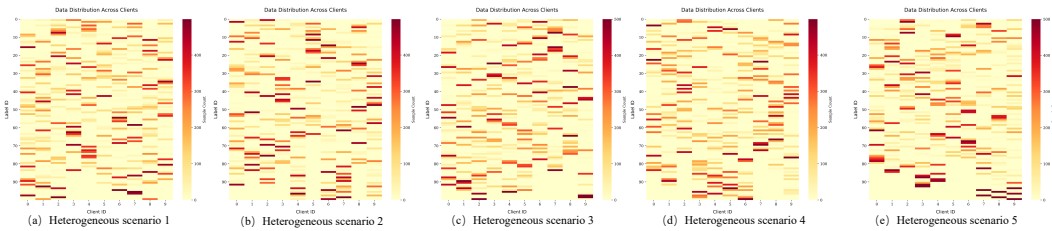

Figure 7: The visualization of the CIFAR-100 dataset distribution across $K = 10$ clients under five different heterogeneous scenarios.

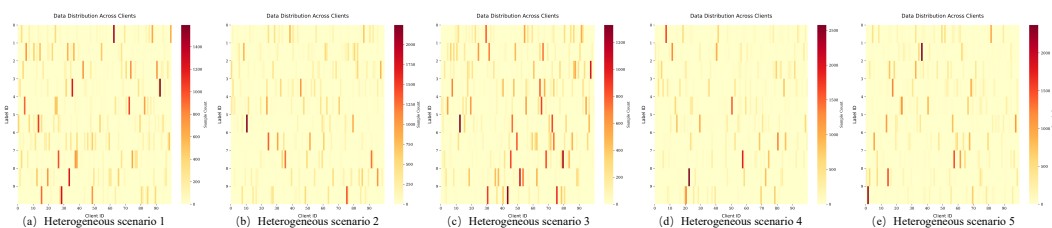

Figure 8: The visualization of the CIFAR-10 dataset distribution across $K = 100$ clients under five different heterogeneous scenarios.

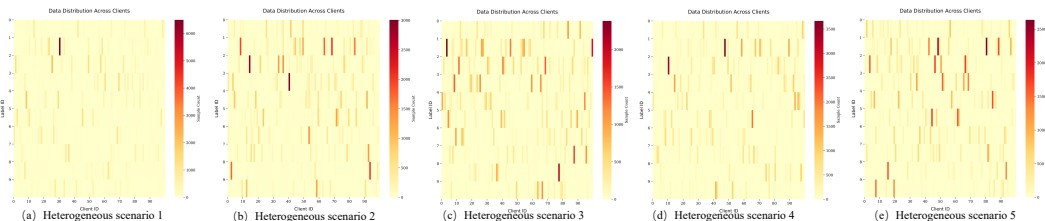

Figure 9: The visualization of the SVHN dataset distribution across $K = 10$ clients under five different heterogeneous scenarios.

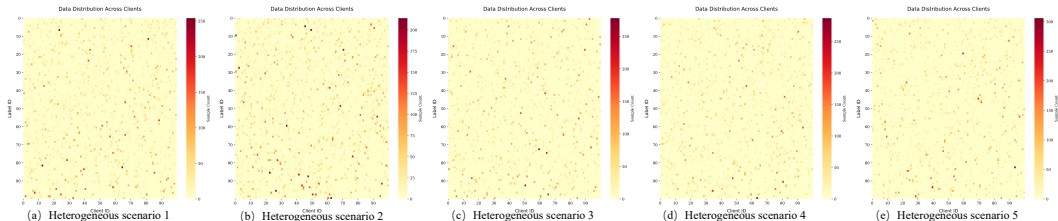

Figure 10: The visualization of the CIFAR-100 dataset distribution across $K = 100$ clients under five different heterogeneous scenarios.

Table 7: All the hyperparameters of the compared baselines, including learning rate, momentum, weight decay, Nesterov, and the remaining hyperparameters of the method itself.

| Method | Learning rate | Momentum | Weight decay | Nesterov | Other hyperparameters |
|---|---|---|---|---|---|
| FedAvg | 0.01 | 0.9 | 0.00001 | False | None |
| FedAvgM | 0.01 | 0.9 | 0.00001 | True | Momentum coefficient: 0.9 |
| FedProx | 0.01 | 0.9 | 0.00001 | False | Proximal coefficient: 0.125 |
| SCAFFOLD | 0.01 | 0.9 | 0.00001 | False | Global step size: 1.0 |
| CCVR | 0.001 | 0.9 | 0.00001 | False | None |
| VHL | 0.01 | 0.9 | 0.00001 | False | VHL alpha:1.0 |
| FedASAM | 0.01 | 0.9 | 0.00001 | False | Rho: 0.1, eta: 0 |
| FedExp | 0.01 | 0.9 | 0.00001 | False | Eps: 1e-3, eta_g: 1.0, lr_weight_decay: 0.998 |
| FedDecorr | 0.01 | 0.9 | 0.00001 | False | Feddecorr term coefficient: 0.1 |
| FedDisco | 0.01 | 0.9 | 0.00001 | False | Metri: 'KL divergence' feddisco a: 0.5, feddisco b: 0.1 |
| FedInit | 0.1 | None | 0.001 | False | Beta: 0.1 |
| FedLESAM | 0.1 | None | 0.001 | False | Rho:0.5, beta:0.1, max_norm:10.0, global step size:1.0 |
| NUCFL | 0.001 | 0.9 | 0.0001 | False | Calibration method: DCA, Non-uniform penalty: CKA |
| **FedGPS (Ours)** | 0.01 | 0.9 | 0.00001 | False | $\lambda_1 : 0.1, \lambda_2 : 0.2, \lambda_g : 0.5, \eta_g : 1.0$ |

---

**Algorithm 1** Pseudo-code of FedGPS

---

**Server input:** communication round $T$, server initialize the model $\boldsymbol{\theta}^0$
**Client $k$'s input:** local epochs $E$, $k$-th local dataset $\mathcal{D}^k$

    **Initialization:** all clients initialize the model $\boldsymbol{\theta}_k^0$ and surrogate data $\mathcal{D}^s$.
    **Server Executes:**
    **for** each round $t = 1, 2, \cdots, T$ **do**
        server random samples a subset of clients $\mathcal{S}_t \subseteq \mathcal{K}$,
        server **communicates** $\boldsymbol{\theta}^t$ to selected clients
        **for** each client $k \in \mathcal{S}_r$ **in parallel do**
            $\Delta_k^{t+1} \leftarrow$ Local_Training $(k, \boldsymbol{\theta}^t)$
        **end for**
        $\boldsymbol{\theta}^{t+1} = \boldsymbol{\theta}^t + \eta_g \frac{1}{|\mathcal{S}_t|} \sum_{k \in \mathcal{S}_t} \Delta_k^{t+1}$
    **end for**

    **Local_Training**$(k, \boldsymbol{\theta}^t)$**:**
    Update local model by global model $\boldsymbol{\theta}_k^t \leftarrow \boldsymbol{\theta}^t$
    Compute $\delta_{\boldsymbol{\theta}_k}^t$ using Definition 4.2
    **for** each iterations $e = 1, 2, \cdots, E$ **do**
        Compute the $\hat{\mathbf{g}}_k^{t+1,e+1}$ new gradient using Eq. 6 and Eq. 7
        Update local model at $e$ iteration: $\boldsymbol{\theta}_k^{t+1,e+1} = \boldsymbol{\theta}_k^{t+1,e} - \eta_l \hat{\mathbf{g}}_k^{t+1,e+1}$
    **end for**
    Compute the update information at this round for client $k$: $\Delta_k^{t+1} = \boldsymbol{\theta}_k^{t+1} - \boldsymbol{\theta}_k^t$
    Storage $\Delta_k^{t+1}$ for Non-self gradient computation
    **Return** $\Delta_k^{t+1}$ to server

---

## C.2 Detailed Hyperparameters

We list all the hyperparameters of the baselines and our framework FedGPS in the Tab. 7.

## C.3 Process and Pseudocode of Algorithm

In the Algorithm section, we elaborate on the pseudocode workflow of FedGPS in Algorithm 1. Consistent with other federated learning (FL) frameworks, we predefine the number of communication

Table 8: Top-1 accuracy of baselines and our method FedGPS with 5 different heterogeneous scenarios on SVHN, heterogeneity degree $\alpha = 0.1$, local epochs $E = 1$ and total client number $K = 100$.

| Dataset: SVHN Heterogeneity Level:$\alpha = 0.1$ Client Number:$K = 100$, Client Sampling Rate: 10% Total Communication Round:$T = 500$ Local Epochs:$E = 1$ | | | | | | | | | | | | | | | | |
|---|---|---|---|---|---|---|---|---|---|---|---|---|---|---|---|---|
| Diff Scenario | Heterogeneous scenario 1 | | | Heterogeneous scenario 2 | | | Heterogeneous scenario 3 | | | Heterogeneous scenario 4 | | | Heterogeneous scenario 5 | | | |
| | Centralized Training Acc=97% | | | | | | | | | | | | | | | |
| | ACC↑ | ROUND↓ | SpeedUp↑ | ACC↑ | ROUND↓ | SpeedUp↑ | ACC↑ | ROUND↓ | SpeedUp↑ | ACC↑ | ROUND↓ | SpeedUp↑ | ACC↑ | ROUND↓ | SpeedUp↑ | |
| Methods | Target Acc=91% | | | Target Acc=91% | | | Target Acc=91% | | | Target Acc=91% | | | Target Acc=91% | | | Mean Acc± Std |
| FedAvg | 91.15 | 473 | 1.0× | 91.06 | 480 | 1.0× | 91.94 | 393 | 1.0× | 91.67 | 437 | 1.0× | 91.17 | 399 | 1.0× | 91.40 ± 0.39 |
| FedAvgM | 92.06 | 368 | 1.3× | 92.55 | 348 | 1.4× | 93.20 | 319 | 1.2× | 92.35 | 497 | 0.9× | 91.38 | 457 | 0.9× | 92.31 ± 0.67 |
| FedProx | 90.92 | None | None | 91.73 | 476 | 1.0× | 91.97 | 496 | 0.8× | 90.97 | None | None | 91.21 | 471 | 0.8× | 91.36 ± 0.47 |
| SCAFFOLD | 93.25 | 300 | 1.6× | 91.86 | 353 | 1.4× | 92.07 | 415 | 0.9× | 92.06 | 434 | 1.0× | 93.00 | 309 | 1.3× | 92.45 ± 0.63 |
| CCVR | 91.74 | **107** | **4.7×** | 92.62 | **102** | **4.0×** | 91.71 | **104** | **4.5×** | 91.61 | **93** | **4.6×** | 90.70 | **155** | **2.8×** | 91.68 ± 0.68 |
| VHL | 93.47 | 314 | 1.5× | 93.75 | 255 | 1.9× | 94.43 | 293 | 1.3× | 94.23 | 265 | 1.6× | 94.05 | 298 | 1.3× | 93.99 ± 0.38 |
| FedASAM | 90.84 | None | None | 91.11 | 476 | 1.0× | 92.55 | 392 | 1.0× | 91.90 | 434 | 1.0× | 92.11 | 479 | 0.8× | 91.70 ± 0.71 |
| FedExp | 90.91 | None | None | 90.73 | None | None | 92.11 | 496 | 0.8× | 91.43 | None | None | 91.40 | 398 | 1.0× | 91.32 ± 0.54 |
| FedDecorr | 91.01 | None | None | 91.15 | 479 | 1.0× | 91.83 | None | None | 90.89 | None | None | 90.47 | None | None | 91.07 ± 0.49 |
| FedDisco | - | - | - | - | - | - | - | - | - | - | - | - | - | - | - | - |
| FedInit | 94.13 | 259 | 1.8× | 93.46 | 291 | 1.6× | 94.25 | 293 | 1.3× | 94.28 | 293 | 1.5× | 94.09 | 320 | 1.2× | 94.04 ± 0.33 |
| FedLESAM | 94.62 | 211 | 2.2× | 94.83 | 160 | 3.0× | 94.65 | 204 | 1.9× | 94.67 | 191 | 2.3× | 94.78 | 187 | 2.1× | 94.71 ± 0.09 |
| NUCFL | 90.61 | None | None | 91.08 | 458 | 1.0× | 91.26 | None | None | 91.29 | None | None | 91.41 | 479 | 0.8× | 91.13 ± 0.31 |
| **FedGPS(Ours)** | **95.03** | 213 | 2.2× | **95.17** | 181 | 2.7× | **95.14** | 239 | 1.6× | **95.01** | 223 | 2.0× | **95.05** | 198 | 2.0× | **95.08 ± 0.07** |

Table 9: Top-1 accuracy of baselines and our method FedGPS with 5 different heterogeneous scenarios on CIFAR-100, heterogeneity degree $\alpha = 0.1$, local epochs $E = 1$ and total client number $K = 100$.

| Dataset: CIFAR-100 Heterogeneity Level:$\alpha = 0.1$ Client Number:$K = 100$, Client Sampling Rate: 10% Total Communication Round:$T = 500$ Local Epochs:$E = 1$ | | | | | | | | | | | | | | | | |
|---|---|---|---|---|---|---|---|---|---|---|---|---|---|---|---|---|
| Diff Scenario | Heterogeneous scenario 1 | | | Heterogeneous scenario 2 | | | Heterogeneous scenario 3 | | | Heterogeneous scenario 4 | | | Heterogeneous scenario 5 | | | |
| | Centralized Training Acc=78% | | | | | | | | | | | | | | | |
| | ACC↑ | ROUND↓ | SpeedUp↑ | ACC↑ | ROUND↓ | SpeedUp↑ | ACC↑ | ROUND↓ | SpeedUp↑ | ACC↑ | ROUND↓ | SpeedUp↑ | ACC↑ | ROUND↓ | SpeedUp↑ | |
| Methods | Target Acc=44% | | | Target Acc=43% | | | Target Acc=41% | | | Target Acc=42% | | | Target Acc=33% | | | Mean Acc± Std |
| FedAvg | 44.72 | 484 | 1.0× | 43.40 | 500 | 1.0× | 41.55 | 483 | 1.0× | 42.43 | 489 | 1.0× | 33.17 | 492 | 1.0× | 41.05 ± 4.56 |
| FedAvgM | 47.22 | 457 | 1.1× | 49.55 | 393 | 1.3× | 49.11 | 372 | 1.3× | 50.69 | 340 | 1.4× | 47.66 | 237 | 2.1× | 48.85 ± 1.42 |
| FedProx | 41.65 | None | None | 37.13 | None | None | 38.13 | None | None | 40.73 | None | None | 28.26 | None | None | 37.18 ± 5.32 |
| SCAFFOLD | 43.04 | None | None | 43.34 | 496 | 1.0× | 41.72 | 483 | 1.0× | 41.36 | None | None | 40.14 | 352 | 1.4× | 41.92 ± 1.30 |
| CCVR | - | - | - | - | - | - | - | - | - | - | - | - | - | - | - | - |
| VHL | 54.51 | 334 | 1.4× | 52.45 | 337 | 1.5× | 53.12 | 304 | 1.6× | 53.83 | 320 | 1.5× | 51.72 | 255 | 1.9× | 53.13 ± 1.10 |
| FedASAM | 43.63 | None | None | 45.69 | 449 | 1.1× | 44.66 | 430 | 1.1× | 46.28 | 432 | 1.1× | 39.93 | 420 | 1.2× | 44.04 ± 2.51 |
| FedExp | - | - | - | - | - | - | - | - | - | - | - | - | - | - | - | - |
| FedDecorr | 47.76 | 427 | 1.1× | 44.06 | 475 | 1.1× | 46.04 | 424 | 1.1× | 47.85 | 391 | 1.3× | 45.03 | 319 | 1.5× | 46.15 ± 1.67 |
| FedDisco | - | - | - | - | - | - | - | - | - | - | - | - | - | - | - | - |
| FedInit | 56.69 | 354 | 1.4× | 57.06 | 338 | 1.5× | 54.59 | 339 | 1.4× | 56.60 | 334 | 1.5× | 55.80 | 257 | 1.9× | 56.15 ± 0.98 |
| FedLESAM | 57.78 | 279 | 1.7× | 57.17 | 269 | 1.9× | 55.07 | 264 | 1.8× | 56.65 | 277 | 1.8× | 56.14 | 211 | 2.3× | 56.56 ± 1.03 |
| NUCFL | - | - | - | - | - | - | - | - | - | - | - | - | - | - | - | - |
| **FedGPS(Ours)** | **58.77** | 264 | 1.8× | **57.84** | 282 | 1.8× | **55.89** | 249 | 1.9× | **57.52** | 260 | 1.9× | **57.93** | 207 | 2.4× | **57.59 ± 1.06** |

rounds and initialize both global and local model parameters. In each round, a subset of $|\mathcal{S}_t|$ clients is sampled for local training and model weight communication. Distinctively, FedGPS first computes the non-self gradient locally. This non-self gradient is then used to derive new weights, corresponding to the dynamic path-oriented rectification process. Subsequently, using these new weights, a new gradient direction is obtained via Eq. 6, which aligns with the static goal-oriented learning objective. The original parameters are then updated based on this new gradient direction. We highlight the key differences from the vanilla FedAvg algorithm to underscore the unique contributions of FedGPS . All related hyperparametrs can be referred to Tab. 7.

# D   Further Experimental Results

In this section, we perform additional ablation studies to demonstrate the effectiveness of FedGPS. We first compare with more baselines (Sec. D.1). Then we explore various settings, including different numbers of clients (Sec. D.2), varying local training epochs (Sec. D.3), various client sampling rate (Sec. D.4), different degrees of heterogeneity (Sec. D.5), another heterogeneity partition method (Sec. D.6), and multiple training seeds to ensure robustness against variations in training initialization and procedures arising from random client sampling (Sec. D.8). Lastly, we give some visualization of the whole training process to verify the performance and convergence (Sec. D.7).

Table 10: The results comparison between FedGPSand more baselines under Heterogeneous scenario 1 with different datasets.

| | $K = 10, \lambda_s = 50\%, R = 500, E = 1$ | $K = 10, \lambda_s = 50\%, R = 200, E = 5$ | $K = 100, \lambda_s = 10\%, R = 500, E = 1$ |
|---|---|---|---|
| CIFAR-10 | | | |
| FedAdam [68] | 85.34 | 85.05 | 67.43 |
| $\Delta$-SGD [69] | 86.96 | 86.66 | 69.49 |
| FedMR [41] | 84.28 | 86.83 | – |
| FedGPS | **90.31** | **88.47** | **78.32** |
| SVHN | | | |
| FedAdam [68] | 89.80 | 91.01 | 93.27 |
| $\Delta$-SGD [69] | 89.78 | 90.21 | 94.08 |
| FedMR [41] | 91.32 | 86.64 | – |
| FedGPS | **94.20** | **93.61** | **95.03** |
| CIFAR-100 | | | |
| FedAdam [68] | 69.89 | 65.33 | 55.43 |
| $\Delta$-SGD [69] | 70.07 | 67.78 | 57.48 |
| FedMR [41] | 69.45 | 67.45 | – |
| FedGPS | **71.14** | **68.90** | **58.77** |

## D.1 More Baselines

Besides, we also compare with other strategies to mitigate the heterogeneity problem in FL. Firstly, we compare with adaptive methods. We select two representative adaptive methods [70], e.g., $\Delta$-SGD [69] and FedAdam [68]. Furthermore, we also include the FedMR [41], which is a new method to modify the aggregation strategy [71]. The results are shown in Tab. 10, FedGPS still outperforms these methods.

## D.2 Ablation Study on Client Number $K$

In this section, the results are listed on Tabs. 8 and 9. We extend our evaluation of FedGPS beyond the CIFAR-10 dataset to include the SVHN and CIFAR-100 datasets, focusing on scenarios with a large number of clients (simulating cross-device settings). Specifically, we set 100 clients, with 10% randomly sampled each round for local training with $E = 1$ local epoch, followed by aggregation. Experimental results reveal that, compared to the 10-client scenario, the 100-client setup with a lower sampling rate leads to reduced model performance within the same number of communication rounds. The SVHN dataset, owing to its relative simplicity, exhibits minimal performance degradation. In contrast, the CIFAR-100 dataset experiences a more pronounced impact. Furthermore, several baseline methods struggle to converge when training larger models, such as ResNet-50, on CIFAR-100 with a low sampling rate, often requiring extensive hyperparameter tuning to address these challenges. Further experiments are needed to investigate these issues thoroughly. We further conduct additional client experiments, e.g., 500 clients across the entire FL system, as shown in Table 11. The results still show the superior performance of FedGPS. Beyond the ResNet-based model, FedGPS also consistently shows improvement on ViT-based models, as Tab. 12 shows.

Table 11: Experimental results of a larger number under Heterogeneous scenario 1 with CIFAR-10 dataset.

| $K = 500, \lambda_s = 10\%$,R=500, E=1 | | | |
|---|---|---|---|
| FedAvg | 73.19 | FedExp | 74.48 |
| FedAvgM | 75.43 | FedDecorr | 74.24 |
| FedProx | 76.72 | FedDisco | - |
| SCAFFOLD | 64.18 | FedInit | 73.05 |
| CCVR | 64.43 | FedLESAM | 80.67 |
| VHL | 80.58 | NUCFL | 73.98 |
| FedASAM | 75.06 | **FedGPS(Ours)** | **82.18** |

Table 12: Experimental results of ViT-based model for 100 clients under Heterogeneous scenario 1 with CIFAR-10 dataset.

| $K = 100, \lambda_s = 10\%$,R=500, E=1 | | | |
|---|---|---|---|
| FedAvg | 30.45 | FedExp | 28.35 |
| FedAvgM | 33.56 | FedDecorr | 39.63 |
| FedProx | 48.46 | FedDisco | - |
| SCAFFOLD | 35.38 | FedInit | 32.34 |
| CCVR | - | FedLESAM | 45.83 |
| VHL | 47.36 | NUCFL | 38.89 |
| FedASAM | 35.43 | **FedGPS(Ours)** | **56.71** |

Table 13: Top-1 accuracy of baselines and our method FedGPS with 5 different heterogeneous scenarios on CIFAR-10, heterogeneity degree $\alpha = 0.1$, local epochs $E = 5$ and total client number $K = 10$.

Dataset: CIFAR-10 Heterogeneity Level:$\alpha = 0.1$ Client Number:$K = 10$, Client Sampling Rate: 50% Total Communication Round:$T = 200$ Local Epochs:$E = 5$

Centralized Training Acc=95%

| Methods | \| Heterogeneous scenario 1 ACC↑ | ROUND↓ | SpeedUp↑ | Heterogeneous scenario 2 ACC↑ | ROUND↓ | SpeedUp↑ | Heterogeneous scenario 3 ACC↑ | ROUND↓ | SpeedUp↑ | Heterogeneous scenario 4 ACC↑ | ROUND↓ | SpeedUp↑ | Heterogeneous scenario 5 ACC↑ | ROUND↓ | SpeedUp↑ | Mean Acc± Std |
|---|---|---|---|---|---|---|---|---|---|---|---|---|---|---|---|---|
| | Target Acc=85% | | | Target Acc=85% | | | Target Acc=84% | | | Target Acc=72% | | | Target Acc=65% | | | |
| FedAvg | 85.89 | 192 | 1.0× | 85.36 | 136 | 1.0× | 84.21 | 147 | 1.0× | 72.71 | 91 | 1.0× | 65.51 | 151 | 1.0× | 78.76 ± 8.17 |
| FedAvgM | 85.60 | 118 | 1.6× | 86.84 | 113 | 1.2× | 84.06 | 140 | 1.1× | 75.56 | 67 | 1.4× | 70.35 | 75 | 2.0× | 80.48 ± 7.18 |
| FedProx | 85.52 | 158 | 1.2× | 84.31 | None | None | 82.87 | None | None | 76.16 | 87 | 1.0× | 74.60 | 62 | 2.4× | 80.69 ± 4.97 |
| SCAFFOLD | 83.75 | None | None | 80.10 | None | None | 82.14 | None | None | 73.32 | 90 | 1.0× | 74.14 | 47 | 3.2× | 78.69 ± 4.72 |
| CCVR | 83.95 | None | None | 83.32 | None | None | | | | 78.54 | 21 | 4.3× | 75.36 | 15 | 10.1× | 81.01 ± 3.88 |
| VHL | 88.10 | 92 | 2.1× | 86.40 | 148 | 0.9× | 84.50 | 146 | 1.0× | 80.91 | 47 | 1.9× | 76.88 | 67 | 2.3× | 83.36 ± 4.50 |
| FedASAM | 85.79 | 118 | 1.6× | 86.12 | 135 | 1.0× | 81.38 | None | None | 74.91 | 67 | 1.4× | 68.01 | 75 | 2.0× | 79.24 ± 7.74 |
| FedExp | 85.05 | 118 | 1.6× | 85.64 | 135 | 1.0× | 82.49 | None | None | 74.15 | 67 | 1.4× | 73.36 | 67 | 2.3× | 80.14 ± 5.95 |
| FedDecorr | 84.53 | None | None | 84.90 | None | None | 83.36 | None | None | 74.75 | 67 | 1.4× | 71.74 | 75 | 2.0× | 79.86 ± 6.15 |
| FedDisco | 85.60 | 191 | 1.0× | 85.59 | 135 | 1.0× | 84.66 | 146 | 1.0× | 70.14 | None | None | 66.79 | 99 | 1.5× | 78.56 ± 9.30 |
| FedInit | 79.23 | None | None | 74.43 | None | None | 75.76 | None | None | 61.05 | None | None | 62.82 | None | None | 70.66 ± 8.18 |
| FedLESAM | 86.36 | 164 | 1.2× | 80.94 | None | None | 81.33 | None | None | 65.53 | None | None | 64.99 | None | None | 75.83 ± 9.88 |
| NUCFL | 82.80 | None | None | 78.48 | None | None | 76.51 | None | None | 66.80 | None | None | 64.57 | None | None | 73.83 ± 7.82 |
| **FedGPS(Ours)** | **88.47** | **68** | **2.8×** | **87.96** | **68** | **2.0×** | **85.79** | **146** | **1.0×** | **84.69** | **47** | **1.9×** | **77.70** | **44** | **3.4×** | **84.92 ± 4.32** |

Table 14: Top-1 accuracy of baselines and our method FedGPS with 5 different heterogeneous scenarios on SVHN, heterogeneity degree $\alpha = 0.1$, local epochs $E = 5$ and total client number $K = 10$.

Dataset: SVHN Heterogeneity Level:$\alpha = 0.1$ Client Number:$K = 10$, Client Sampling Rate: 50% Total Communication Round:$T = 200$ Local Epochs:$E = 5$

Centralized Training Acc=97%

| Methods | \| Heterogeneous scenario 1 ACC↑ | ROUND↓ | SpeedUp↑ | Heterogeneous scenario 2 ACC↑ | ROUND↓ | SpeedUp↑ | Heterogeneous scenario 3 ACC↑ | ROUND↓ | SpeedUp↑ | Heterogeneous scenario 4 ACC↑ | ROUND↓ | SpeedUp↑ | Heterogeneous scenario 5 ACC↑ | ROUND↓ | SpeedUp↑ | Mean Acc± Std |
|---|---|---|---|---|---|---|---|---|---|---|---|---|---|---|---|---|
| | Target Acc=89% | | | Target Acc=91% | | | Target Acc=92% | | | Target Acc=92% | | | Target Acc=92% | | | |
| FedAvg | 89.21 | 167 | 1.0× | 91.99 | 101 | 1.0× | 92.51 | 100 | 1.0× | 92.55 | 133 | 1.0× | 92.57 | 65 | 1.0× | 91.77 ± 1.45 |
| FedAvgM | 86.64 | None | None | 93.06 | 51 | 2.0× | 92.09 | 102 | 1.0× | 91.38 | None | None | 92.71 | 64 | 1.0× | 91.18 ± 2.62 |
| FedProx | 88.81 | None | None | 91.69 | 86 | 1.2× | 93.32 | 82 | 1.2× | 91.78 | None | None | 93.22 | 52 | 1.2× | 91.76 ± 1.82 |
| SCAFFOLD | 83.42 | None | None | 90.93 | None | None | 91.53 | None | None | 92.15 | 103 | 1.3× | 91.61 | None | None | 89.93 ± 3.66 |
| CCVR | 85.06 | None | None | 90.24 | None | None | 91.26 | None | None | 90.48 | None | None | 91.41 | None | None | 89.69 ± 2.64 |
| VHL | 93.10 | 80 | 2.1× | 93.56 | 67 | 1.5× | 94.31 | 64 | 1.6× | 93.62 | 71 | 1.9× | 94.03 | 57 | 1.1× | 93.72 ± 0.46 |
| FedASAM | 86.96 | None | None | 92.94 | 71 | 1.4× | 93.50 | 71 | 1.4× | 92.19 | 144 | 0.9× | 93.30 | 64 | 1.0× | 91.78 ± 2.74 |
| FedExp | 88.31 | None | None | 92.46 | 100 | 1.0× | 92.59 | 71 | 1.4× | 92.08 | 148 | 0.9× | 92.92 | 64 | 1.0× | 91.67 ± 1.90 |
| FedDecorr | 86.97 | None | None | 91.79 | 101 | 1.0× | 93.49 | 47 | 2.1× | 92.44 | 130 | 1.0× | 93.60 | 64 | 1.0× | 91.66 ± 2.73 |
| FedDisco | 88.43 | None | None | 91.72 | 100 | 1.0× | 92.53 | 99 | 1.0× | 92.27 | 97 | 1.4× | 92.86 | 64 | 1.0× | 91.56 ± 1.80 |
| FedInit | 65.26 | None | None | 77.42 | None | None | 90.57 | None | None | 85.57 | None | None | 88.56 | None | None | 81.48 ± 10.36 |
| FedLESAM | 72.39 | None | None | 88.16 | None | None | 91.31 | None | None | 87.07 | None | None | 91.19 | None | None | 86.02 ± 7.84 |
| NUCFL | 86.68 | None | None | 90.20 | None | None | 90.75 | None | None | 91.26 | None | None | 91.58 | None | None | 90.09 ± 1.98 |
| **FedGPS(Ours)** | **93.61** | **80** | **2.1×** | **94.20** | **57** | **1.0×** | **95.08** | **49** | **2.0** | **94.30** | **68** | **2.0** | **94.76** | **50** | **1.3** | **94.39 ± 0.56** |

## D.3 Ablation Study on Local Epoch $E$

The number of local training epochs significantly impacts the performance of federated learning (FL). Excessive local fitting can exacerbate performance degradation. To investigate the effect of local epochs on various existing methods and FedGPS , we adopt a consistent experimental setup with $K = 10$ clients, a heterogeneity degree $\alpha = 0.1$, and 50% of clients randomly sampled per round, varying only the number of local training epochs $E = 5$. Experiments are conducted across the CIFAR-10, CIFAR-100, and SVHN datasets, with results reported in Tabs. 13, 15 and 14. The findings indicate that performance generally declines as local epochs increase for most methods, including FedGPS , VHL, and FedLESAM. For the methods with increasing performance, especially for some local learning objective modification methods, the additional penalty term increases the learning difficulty, and a small local epochs cannot fully train the local model, so a larger local epochs is more suitable for such methods. However, the larger local epochs make the performance of these methods after over-training still needs to be further verified.

**Observation 4: SAM-based method needs to be adapted to different local epochs.** SAM-based methods require distinct gradient perturbation strategies depending on the number of local epochs. Notably, FedLESAM, a SAM-based method, exhibits significant performance degradation with

Table 15: Top-1 accuracy of baselines and our method FedGPS with 5 different heterogeneous scenarios on CIFAR-100, heterogeneity degree $\alpha = 0.1$, local epochs $E = 5$ and total client number $K = 10$.

| | Dataset: CIFAR-100 Heterogeneity Level:$\alpha = 0.1$ Client Number:$K = 10$, Client Sampling Rate: 50% Total Communication Round:$T = 200$ Local Epochs:$E = 5$ | | | | | | | | | | | | | | | |
|---|---|---|---|---|---|---|---|---|---|---|---|---|---|---|---|---|
| Diff Scenario | Heterogeneous scenario 1 | | | Heterogeneous scenario 2 | | | Heterogeneous scenario 3 | | | Heterogeneous scenario 4 | | | Heterogeneous scenario 5 | | | |
| | Centralized Training Acc=78% | | | | | | | | | | | | | | | |
| | ACC↑ | ROUND↓ | SpeedUp↑ | ACC↑ | ROUND↓ | SpeedUp↑ | ACC↑ | ROUND↓ | SpeedUp↑ | ACC↑ | ROUND↓ | SpeedUp↑ | ACC↑ | ROUND↓ | SpeedUp↑ | |
| Methods | Target Acc=67% | | | Target Acc=67% | | | Target Acc=67% | | | Target Acc=69% | | | Target Acc=64% | | | Mean Acc± Std |
| FedAvg | 67.98 | 138 | 1.0× | 67.72 | 178 | 1.0× | 67.50 | 147 | 1.0× | 69.09 | 181 | 1.0× | 64.32 | 170 | 1.0× | 67.32 ± 1.79 |
| FedAvgM | 68.85 | **108** | 1.3× | 68.38 | 159 | 1.1× | 68.08 | 144 | 1.0× | 68.46 | 180 | 1.0× | 63.55 | None | None | 67.46 ± 2.21 |
| FedProx | 66.95 | None | None | 66.38 | None | None | 66.56 | None | None | 67.74 | None | None | 60.39 | None | None | 65.60 ± 2.96 |
| SCAFFOLD | 64.54 | None | None | 63.15 | None | None | 62.84 | None | None | 64.14 | None | None | 61.91 | None | None | 63.32 ± 1.05 |
| CCVR | - | - | - | - | - | - | - | - | - | - | - | - | - | - | - | - |
| VHL | 68.53 | 123 | 1.1× | 67.44 | 185 | 1.0× | 68.19 | 173 | 0.8× | 68.45 | None | None | 66.13 | 132 | 1.3× | 67.75 ± 1.00 |
| FedASAM | 68.73 | 110 | 1.3× | 68.30 | 153 | 1.2× | 68.09 | 146 | 1.0× | **69.13** | **176** | 1.0× | 62.35 | None | None | 67.32 ± 2.81 |
| FedExp | 68.70 | 123 | 1.1× | 67.50 | 170 | 1.0× | 68.07 | 162 | 0.9× | 68.34 | 199 | 0.9× | 57.95 | None | None | 66.11 ± 4.58 |
| FedDecorr | 67.41 | 184 | 0.8× | 66.80 | None | None | 67.15 | **146** | 1.0× | 67.77 | None | None | 61.00 | None | None | 66.03 ± 2.83 |
| FedDisco | 67.59 | 137 | 1.0× | 68.29 | 159 | 1.1× | 68.21 | 163 | 0.9× | 68.23 | 180 | 1.0× | 63.75 | None | None | 67.21 ± 1.96 |
| FedInit | 61.70 | None | None | 61.38 | None | None | 60.25 | None | None | 63.15 | None | None | 57.57 | None | None | 60.81 ± 2.09 |
| FedLESAM | 61.61 | None | None | 60.25 | None | None | 60.14 | None | None | 61.88 | None | None | 58.21 | None | None | 60.42 ± 1.46 |
| NUCFL | 61.75 | None | None | 59.06 | None | None | 59.18 | None | None | 60.77 | None | None | 59.23 | None | None | 60.00 ± 1.20 |
| **FedGPS(Ours)** | **68.90** | 139 | 1.0× | **68.45** | 147 | 1.2× | **68.56** | 180 | 0.8× | 68.76 | None | None | **66.71** | 114 | 1.5× | **68.28 ± 0.89** |

Table 16: Top-1 accuracy of baselines and our method FedGPS with 5 different heterogeneous scenarios on CIFAR-10, heterogeneity degree $\alpha = 0.05$, local epochs $E = 1$ and total client number $K = 10$.

| | Dataset: CIFAR-10 Heterogeneity Level:$\alpha = 0.05$ Client Number:$K = 10$, Client Sampling Rate: 50% Total Communication Round:$T = 500$ Local Epochs:$E = 1$ | | | | | | |
|---|---|---|---|---|---|---|---|
| Diff Scenario | Heterogeneous scenario 1 | | | Heterogeneous scenario 2 | | | |
| | Centralized Training Acc=95% | | | | | | |
| | ACC↑ | ROUND↓ | SpeedUp↑ | ACC↑ | ROUND↓ | SpeedUp↑ | |
| Methods | Target Acc=75% | | | Target Acc=49% | | | Mean Acc± Std |
| FedAvg | 75.28 | 298 | 1.0× | 45.59 | 233 | 1.0× | 60.44 ± 20.99 |
| FedAvgM | 75.81 | 105 | 2.8× | 49.26 | 275 | 0.8× | 62.53 ± 18.77 |
| FedProx | 79.77 | 176 | 1.7× | 52.17 | 237 | 1.0× | 65.97 ± 19.52 |
| SCAFFOLD | 46.67 | None | None | 51.14 | 180 | 1.3× | 48.91 ± 3.16 |
| CCVR | 78.20 | **94** | 3.2× | 60.53 | **40** | **5.8×** | 69.37 ± 12.49 |
| VHL | 85.22 | 143 | 2.1× | 69.09 | 84 | 2.8× | 77.16 ± 11.41 |
| FedASAM | 75.84 | 198 | 1.5× | 50.62 | 275 | 0.8× | 63.23 ± 17.83 |
| FedExp | 75.70 | 198 | 1.5× | 46.86 | None | None | 61.28 ± 20.39 |
| FedDecorr | 81.03 | 105 | 2.8× | 51.69 | 96 | 2.4× | 66.36 ± 20.75 |
| FedDisco | 82.37 | 176 | 1.7 | 51.01 | 110 | 2.1× | 66.69 ± 22.17 |
| FedInit | 77.55 | 474 | 0.6× | 47.99 | None | None | 62.77 ± 20.90 |
| FedLESAM | 79.92 | 198 | 1.5× | 51.90 | 103 | 2.3× | 65.91 ± 19.81 |
| NUCFL | 72.96 | None | None | 46.92 | None | None | 59.94 ± 18.41 |
| **FedGPS(Ours)** | **86.97** | 198 | 1.5× | **70.48** | 94 | 2.5× | **78.72 ± 11.66** |

increasing local epochs across all three datasets (SVHN, CIFAR-10, and CIFAR-100). In contrast, FedASAM, another SAM-based method, shows minimal degradation and, in some cases, slight improvement. This suggests that SAM-based methods necessitate tailored gradient perturbation mechanisms when local epochs are extended.

### D.4   Ablation Study on Client Sampling Rate $\lambda_s$

To investigate the impact of client participation on the performance of FL systems, we conduct an ablation study on the client sampling rate. In FL frameworks like FedAvg [7], the sampling rate

Table 17: Top-1 accuracy of baselines and our method FedGPS with 5 different heterogeneous scenarios on CIFAR-10, heterogeneity degree $\alpha = 0.05$, local epochs $E = 1$ and total client number $K = 100$.

| Dataset: CIFAR-10 Heterogeneity Level:$\alpha = 0.05$ Client Number:$K = 100$, Client Sampling Rate: $10\%$ Total Communication Round:$T = 500$ Local Epochs:$E = 1$ | | | | | | | |
|---|---|---|---|---|---|---|---|
| Diff Scenario | Heterogeneous scenario 1 | | | Heterogeneous scenario 2 | | | |
| Centralized Training Acc=95% | | | | | | | |
| | ACC↑ | ROUND ↓ | SpeedUp↑ | ACC↑ | ROUND ↓ | SpeedUp↑ | |
| Methods | Target Acc=32% | | | Target Acc=34% | | | Mean Acc± Std |
| FedAvg | 32.97 | 464 | 1.0× | 34.36 | 473 | 1.0× | $33.67 \pm 0.98$ |
| FedAvgM | 41.86 | 256 | 1.8× | 33.15 | None | None | $37.50 \pm 6.16$ |
| FedProx | 36.63 | 241 | 1.9× | 34.99 | 486 | 1.0× | $35.81 \pm 1.16$ |
| SCAFFOLD | 48.80 | **27** | **17.2×** | 50.16 | 31 | 15.3× | $49.48 \pm 0.96$ |
| CCVR | 55.28 | 28 | 16.6× | 59.36 | **24** | **19.7×** | $57.32 \pm 2.88$ |
| VHL | 59.79 | 115 | 4.0× | 59.85 | 163 | 2.9× | $59.82 \pm 0.04$ |
| FedASAM | 33.35 | 477 | 1.0× | 30.14 | None | None | $31.75 \pm 2.27$ |
| FedExp | 35.90 | 409 | 1.1× | 29.68 | None | None | $32.79 \pm 4.40$ |
| FedDecorr | - | - | - | 47.85 | 212 | 2.2× | - |
| FedDisco | - | - | - | - | - | - | - |
| FedInit | 52.98 | 57 | 8.1× | 58.51 | 65 | 7.3× | $55.74 \pm 3.91$ |
| FedLESAM | 60.90 | 55 | 8.4× | 64.63 | 69 | 6.9× | $62.77 \pm 2.64$ |
| NUCFL | 40.70 | 264 | 1.8× | 37.91 | 430 | 1.1× | $39.30 \pm 1.97$ |
| **FedGPS(Ours)** | **65.86** | 70 | 6.6× | **70.14** | 72 | 6.6× | **$68.00 \pm 3.03$** |

Table 18: The ablation study of client sampling rate under $\alpha = 0.1$, $K = 100$, $R = 500$, $E = 1$ with CIFAR-10 dataset.

| Sampling rate $\lambda_s$ | 5% | 10% | 20% | 50% | Sampling rate $\lambda_s$ | 5% | 10% | 20% | 50% |
|---|---|---|---|---|---|---|---|---|---|
| FedAvg | 34.59 | 48.22 | 67.38 | 72.45 | FedExp | 34.38 | 38.26 | 58.35 | 64.52 |
| FedAvgM | 35.32 | 58.80 | 68.34 | 73.89 | FedDecorr | – | 63.69 | 77.31 | 80.03 |
| FedProx | 32.89 | 52.84 | 65.25 | 74.24 | FedDisco | – | – | – | – |
| SCAFFOLD | 36.73 | 60.17 | 67.59 | 69.34 | FedInit | 30.56 | 71.01 | 76.89 | 77.59 |
| CCVR | 57.93 | 64.06 | 74.25 | 78.78 | FedLESAM | 68.86 | 72.64 | 76.35 | 76.90 |
| VHL | 57.14 | 72.70 | 76.58 | 80.78 | NUCFL | 38.85 | 53.72 | 68.31 | 71.91 |
| FedASAM | 34.78 | 46.35 | 61.84 | 66.48 | **FedGPS(Ours)** | **70.89** | **78.32** | **79.72** | **81.97** |

determines the fraction of clients randomly selected per training round, balancing computational load, communication overhead, and model convergence. Lower sampling rates reduce bandwidth usage and enable scalability in resource-constrained environments, but may slow convergence due to noisier updates from fewer participants. Conversely, higher rates accelerate learning at the cost of increased synchronization demands. By varying the sampling rate (e.g., from 10% to 50%) while keeping other hyperparameters fixed, this experiment isolates its effects on metrics such as test accuracy. The results are shown in Tab. 18. Under different client sampling rates, FedGPS still performs better than other baselines.

### D.5 Ablation Study on Heterogeneity Degree $\alpha$

To assess the impact of varying degrees of heterogeneity, we conduct experiments under constrained computational resources. Specifically, we evaluated two distinct heterogeneity scenarios on the CIFAR-10 dataset, with setups of 10 and 100 clients, respectively. The results are presented in Tabs. 16 and 17. The results demonstrate that as heterogeneity increases, overall performance declines, and the performance gap across different heterogeneous scenarios widens. Specifically, as shown in Table 16, FedAvg's accuracy drops significantly from 75.28 to 45.59, a decrease of 29.69. Notably, FedGPS exhibits greater performance improvements in more challenging scenarios. For instance, in the $\alpha = 0.1$ scenario, FedGPS outperforms the best baseline method by an average of 1.04 in accuracy, while in the more heterogeneous $\alpha = 0.05$ scenario, this improvement rises to 1.56. The advantage of FedGPS becomes even more pronounced with a larger number of clients and lower sampling rates, achieving an improvement of 2.06 in the $\alpha = 0.1$ scenario and 5.23 in the $\alpha = 0.05$ scenario as shown in Tab. 17. These findings further validate the effectiveness and robustness of FedGPS under diverse and challenging FL conditions.

### D.6 Different Heterogeneity Partition Strategy

Table 19: Experimental results under $C = N$ heterogeneity partition method with CIFAR-10 dataset. Here we mainly select $C = 2$ and $C = 3$ these two scenarios.

|  | $C = 2$ | $C = 3$ |  | $C = 2$ | $C = 3$ |
|---|---|---|---|---|---|
| FedAvg | 51.75 | 69.97 | FedExp | 47.53 | 67.29 |
| FedAvgM | 50.61 | 73.21 | FedDecorr | 69.36 | 79.60 |
| FedProx | 49.64 | 68.96 | FedDisco | – | – |
| SCAFFOLD | 55.08 | 74.34 | FedInit | 54.80 | 71.12 |
| CCVR | 55.83 | 73.69 | FedLESAM | 66.62 | 79.42 |
| VHL | 73.53 | 84.16 | NUCFL | 50.03 | 72.89 |
| FedASAM | 55.33 | 69.37 | **FedGPS(Ours)** | **78.17** | **85.71** |

Besides the Dirichlet distribution-based partitioning, another common approach is label distribution skew with limited classes per client, often denoted as $C = N$ [72]. In this method, each client is restricted to samples from only $N$ distinct classes out of the total available classes in the dataset, creating extreme heterogeneity. This partitioning simulates scenarios where clients have specialized or siloed data, such as different devices capturing only certain categories (e.g., one client with images of cats and dogs only, while another has birds and fish). It emphasizes qualitative skew (absence of entire classes) rather than quantitative skew (imbalanced sample counts per class). To verify that FedGPS is still robust under other heterogeneous partition methods, the experimental results in $C = 2$ and $C = 3$ scenarios are shown in Tab 19, indicating that FedGPS is still robust to different heterogeneous partition methods.

### D.7 More Visualization of Results

In this section, we visualize additional experimental results, which illustrate the dynamic training process while highlighting performance and convergence speed. Due to the large number of baselines, we visualize only the top-5 methods in this setting alongside FedGPS for comparison. The results are shown in Figs. 11, 12, 13, 14 and 15. The results reveal that in the early training rounds, FedGPS does not exhibit a significant speed advantage over other methods and, in some cases, converges more slowly. However, as training progresses, FedGPS demonstrates sustained performance improvements in later rounds, while other methods plateau, with their performance stabilizing. This observation motivates future research into developing more efficient variants of FedGPS .

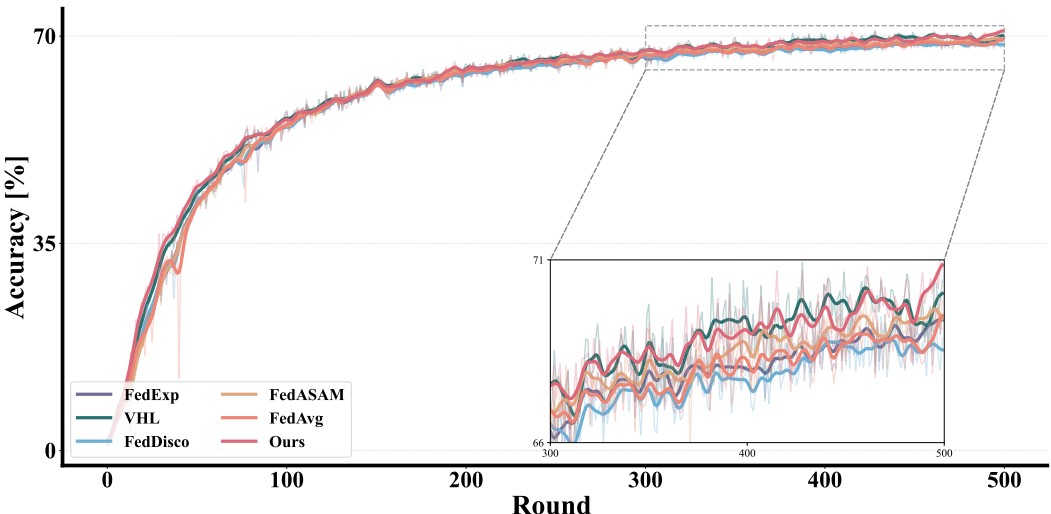

Figure 11: The training process visualization of top-5 baselines and our method FedGPS on CIFAR-100, heterogeneity degree $\alpha = 0.1$, local epochs $E = 1$ and total client number $K = 10$ under heterogeneous scenario 1.

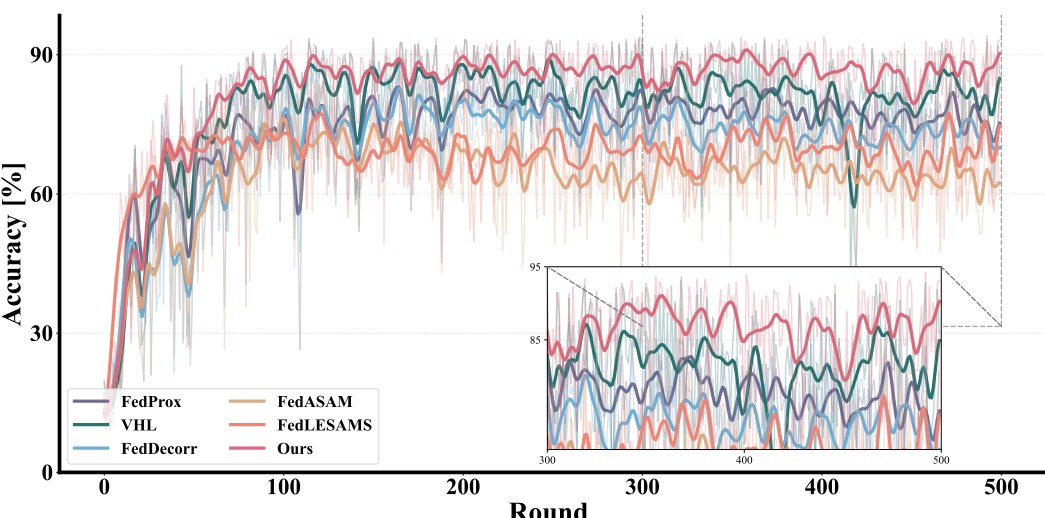

Figure 12: The training process visualization of top-5 baselines and our method FedGPS on SVHN, heterogeneity degree $\alpha = 0.1$, local epochs $E = 1$ and total client number $K = 10$ under heterogeneous scenario 1.

## D.8 Different Random Training Seeds

The choice of random training seeds impacts model initialization and the random client sampling process. To further validate the effectiveness of FedGPSin this context, we conduct experiments across the same heterogeneous scenarios using three distinct random seeds to control for this randomness. The results, presented in Tabs. 20 and 21, reveal that such randomness noticeably affects algorithm performance, with variations in some scenarios reaching up to $\pm 2$ or more. Despite this, FedGPSconsistently mitigates the impact of randomness on performance, as evidenced by lower standard deviations. These findings underscore the robustness of FedGPS, not only in addressing

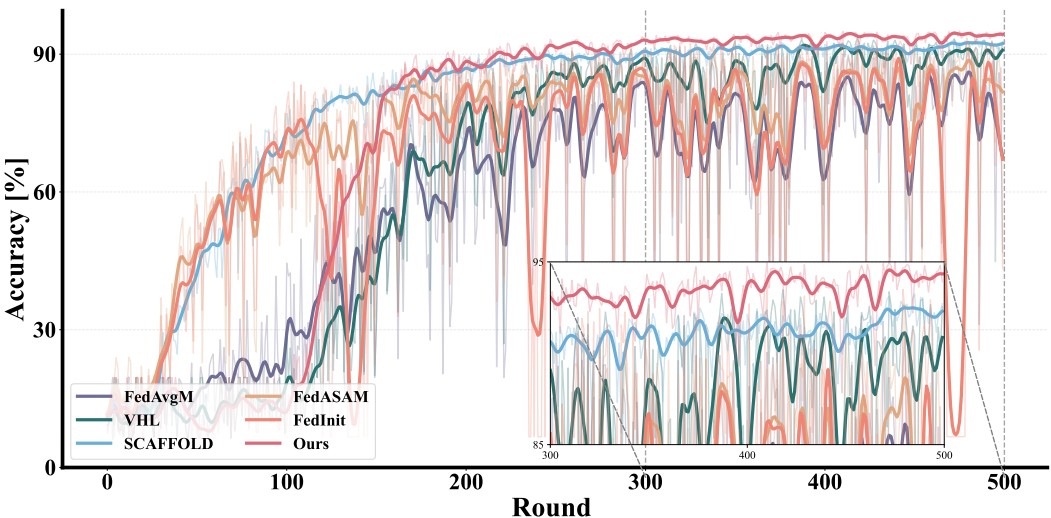

Figure 13: The training process visualization of top-5 baselines and our method FedGPS on SVHN, heterogeneity degree $\alpha = 0.1$, local epochs $E = 1$ and total client number $K = 100$ under heterogeneous scenario 1.

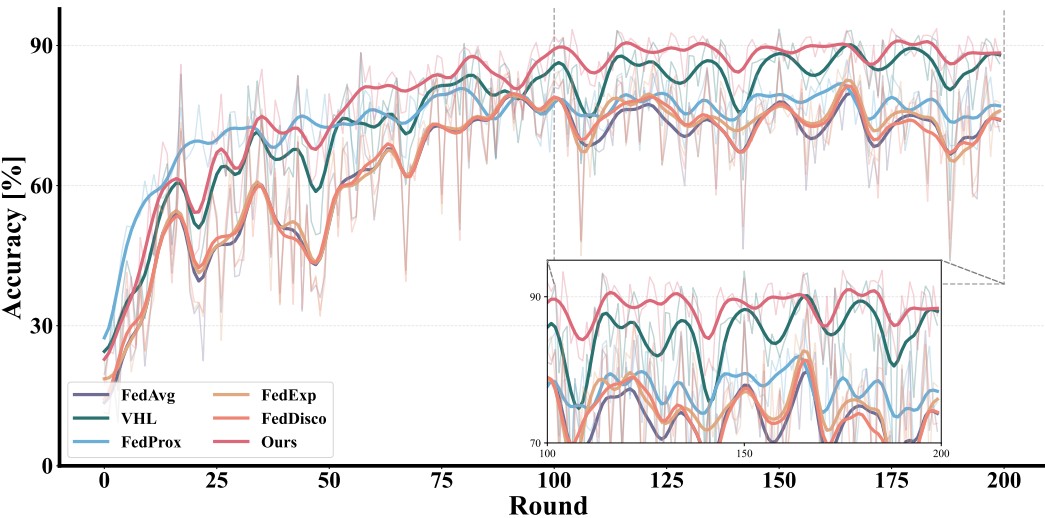

Figure 14: The training process visualization of top-5 baselines and our method FedGPS on SVHN, heterogeneity degree $\alpha = 0.1$, local epochs $E = 5$ and total client number $K = 10$ under heterogeneous scenario 1.

heterogeneous data distributions but also in handling variability from model initialization and random client selection process.

Table 20: Top-1 accuracy of baselines and our method FedGPS with 5 different heterogeneous scenarios on CIFAR-10, heterogeneity degree $\alpha = 0.1$, local epochs $E = 1$ and total client number $K = 10$ under 3 different training random seeds.

Dataset: CIFAR-10 Heterogeneity Level:$\alpha = 0.1$ Client Number:$K = 10$, Client Sampling Rate: $50\%$ Total Communication Round:$R = 500$ Local Epochs:$E = 1$ with 3 different random seeds for training

| | | FedAvg | FedAvgM | FedProx | SCAFFOLD | CCVR | VHL | FedSAM | FedExp | FedDecorr | FedDisco | FedInit | FedLESAM | NUCFL | FedGPS(Ours) |
|---|---|---|---|---|---|---|---|---|---|---|---|---|---|---|---|
| Heterogeneous scenario 1 | seed 0 | 84.21 | 85.74 | 86.13 | 82.39 | 84.30 | 89.07 | 86.49 | 84.00 | 85.76 | 85.69 | 86.84 | 88.80 | 83.76 | **90.31** |
| | seed 1 | 83.67 | 84.81 | 86.61 | 81.97 | 83.84 | 88.65 | 84.98 | 84.53 | 85.31 | 83.30 | 86.62 | 86.49 | 82.34 | **89.57** |
| | seed 2 | 85.00 | 84.75 | 86.45 | 81.12 | 83.77 | 90.02 | 85.80 | 83.19 | 85.60 | 85.75 | 88.08 | 86.67 | 81.98 | **90.29** |
| | Mean± std | 84.29 ± 0.67 | 85.10 ± 0.56 | 86.40 ± 0.24 | 81.83 ± 0.65 | 83.97 ± 0.29 | 89.25 ± 0.70 | 85.76 ± 0.76 | 83.91 ± 0.67 | 85.56 ± 0.23 | 84.91 ± 1.40 | 87.18 ± 0.79 | 87.00 ± 1.28 | 82.69 ± 0.94 | **90.06 ± 0.42** |
| | | | | | | | Target Acc=84% | | | | | | | | |
| Heterogeneous scenario 2 | seed 0 | 79.13 | 81.78 | 83.12 | 80.78 | 83.28 | 87.20 | 81.99 | 79.25 | 84.07 | 81.84 | 83.49 | 84.88 | 79.45 | **88.45** |
| | seed 1 | 83.34 | 85.17 | 84.35 | 78.60 | 81.64 | 88.22 | 84.25 | 79.72 | 82.96 | 81.93 | 86.72 | 85.62 | 77.46 | **88.89** |
| | seed 2 | 84.40 | 83.89 | 82.67 | 80.75 | 81.22 | 87.69 | 83.06 | 82.69 | 82.15 | 83.87 | 82.27 | 85.69 | 80.65 | **88.00** |
| | Mean± std | 82.29 ± 2.79 | 83.61 ± 1.71 | 83.38 ± 0.87 | 80.04 ± 1.25 | 82.05 ± 1.09 | 87.70 ± 0.51 | 83.10 ± 1.13 | 80.55 ± 1.87 | 83.06 ± 0.96 | 82.55 ± 1.15 | 84.16 ± 2.30 | 85.40 ± 0.45 | 79.19 ± 1.61 | **88.45 ± 0.45** |
| | | | | | | | Target Acc=79% | | | | | | | | |
| Heterogeneous scenario 3 | seed 0 | 80.63 | 81.35 | 82.37 | 79.08 | 83.20 | 86.83 | 80.45 | 79.60 | 81.38 | 80.42 | 80.48 | 84.78 | 79.76 | **87.78** |
| | seed 1 | 80.18 | 82.61 | 83.52 | 79.59 | 82.23 | 86.34 | 81.11 | 79.51 | 81.25 | 80.56 | 81.00 | 83.49 | 80.32 | **86.43** |
| | seed 2 | 79.51 | 80.36 | 82.91 | 81.56 | 81.87 | 87.00 | 80.34 | 80.29 | 81.86 | 79.94 | 79.29 | 83.80 | 82.01 | **87.44** |
| | Mean± std | 80.11 ± 0.56 | 81.44 ± 1.13 | 82.93 ± 0.58 | 80.08 ± 1.31 | 82.43 ± 0.69 | 86.72 ± 0.34 | 80.63 ± 0.42 | 79.80 ± 0.43 | 81.50 ± 0.32 | 80.31 ± 0.33 | 80.26 ± 0.88 | 84.02 ± 0.67 | 80.70 ± 1.17 | **87.22 ± 0.70** |
| | | | | | | | Target Acc=80% | | | | | | | | |
| Heterogeneous scenario 4 | seed 0 | 68.62 | 70.15 | 76.62 | 71.83 | 76.57 | 84.30 | 73.11 | 71.55 | 73.14 | 70.37 | 69.44 | 78.99 | 68.78 | **85.06** |
| | seed 1 | 68.50 | 69.59 | 77.87 | 69.23 | 76.31 | 83.67 | 71.22 | 66.78 | 72.98 | 67.90 | 68.87 | 76.59 | 68.96 | **85.08** |
| | seed 2 | 68.88 | 70.02 | 77.59 | 72.93 | 76.52 | 82.66 | 71.89 | 66.27 | 75.54 | 70.58 | 68.33 | 76.43 | 69.73 | **85.59** |
| | Mean± std | 68.67 ± 0.19 | 69.92 ± 0.29 | 77.36 ± 0.66 | 71.33 ± 1.90 | 76.47 ± 0.14 | 83.41 ± 1.04 | 72.07 ± 0.96 | 68.20 ± 2.91 | 73.89 ± 1.43 | 69.62 ± 1.49 | 68.88 ± 0.56 | 77.34 ± 1.43 | 69.16 ± 0.50 | **85.24 ± 0.30** |
| | | | | | | | Target Acc=68% | | | | | | | | |
| Heterogeneous scenario 5 | seed 0 | 65.86 | 67.51 | 68.81 | 68.43 | 74.72 | 81.05 | 66.68 | 66.66 | 73.77 | 69.94 | 68.04 | 74.15 | 65.78 | **82.04** |
| | seed 1 | 68.85 | 71.45 | 69.48 | 65.50 | 73.49 | 80.06 | 69.12 | 68.32 | 70.91 | 67.91 | 70.05 | 68.83 | 69.45 | **80.71** |
| | seed 2 | 72.56 | 66.21 | 67.52 | 66.15 | 73.11 | 78.76 | 73.37 | 68.16 | 68.84 | 68.51 | 67.46 | 70.76 | 73.87 | **79.65** |
| | Mean± std | 69.09 ± 3.36 | 68.39 ± 2.73 | 68.60 ± 1.00 | 66.69 ± 1.54 | 73.77 ± 0.84 | 79.96 ± 1.15 | 69.72 ± 3.39 | 67.71 ± 0.92 | 71.17 ± 2.48 | 68.79 ± 1.04 | 68.52 ± 1.36 | 71.25 ± 2.69 | 69.70 ± 4.05 | **80.80 ± 1.20** |
| | | | | | | | Target Acc=65% | | | | | | | | |
| Total Mean Acc ± std | | 76.89 ± 7.11 | 77.69 ± 7.45 | 79.73 ± 6.53 | 75.99 ± 6.24 | 79.74 ± 4.10 | 85.41 ± 3.51 | 78.26 ± 6.64 | 76.03 ± 7.11 | 79.03 ± 5.84 | 77.23 ± 7.03 | 77.80 ± 8.10 | 81.06 ± 6.29 | 76.29 ± 6.17 | **86.35 ± 3.35** |

Table 21: Top-1 accuracy of baselines and our method FedGPS with 5 different heterogeneous scenarios on CIFAR-10, heterogeneity degree $\alpha = 0.1$, local epochs $E = 5$ and total client number $K = 10$ under 3 different training random seeds.

Dataset: CIFAR-10 Heterogeneity Level:$\alpha = 0.1$ Client Number:$K = 10$, Client Sampling Rate: **50%** Total Communication Round:$R = 200$ Local Epochs:$E = 5$ with 3 different random seeds for training

| | | FedAvg | FedAvgM | FedProx | SCAFFOLD | CCVR | VHL | FedSAM | FedExp | FedDecorr | FedDisco | Fedini | FedESAM | NUCFL | FedGPS(Ours) |
|---|---|---|---|---|---|---|---|---|---|---|---|---|---|---|---|
| | | | | | | | Target Acc=85% | | | | | | | | |
| Heterogeneous scenario 1 | seed 0 | 85.89 | 85.60 | 85.52 | 83.75 | 83.95 | 88.10 | 85.79 | 85.05 | 84.53 | 85.60 | 79.23 | 86.36 | 82.80 | **88.47** |
| | seed 1 | 86.40 | 86.28 | 86.13 | 82.90 | 84.20 | 88.41 | 86.14 | 86.31 | 87.01 | 85.54 | 83.56 | 85.83 | 82.08 | **89.69** |
| | seed 2 | 85.67 | 85.75 | 85.95 | 83.67 | 84.63 | 88.67 | 86.20 | 85.82 | 85.29 | 85.87 | 83.05 | 85.34 | 81.87 | **88.89** |
| | Mean± std | 85.99 ± 0.37 | 85.88 ± 0.36 | 85.87 ± 0.31 | 83.44 ± 0.47 | 84.26 ± 0.34 | 88.39 ± 0.29 | 86.04 ± 0.22 | 85.73 ± 0.64 | 85.61 ± 1.27 | 85.67 ± 0.18 | 81.95 ± 2.37 | 85.84 ± 0.51 | 82.25 ± 0.49 | **89.02 ± 0.62** |
| | | | | | | | Target Acc=85% | | | | | | | | |
| Heterogeneous scenario 2 | seed 0 | 85.36 | 86.84 | 84.31 | 80.10 | 83.87 | 86.40 | 86.12 | 85.64 | 84.90 | 85.59 | 74.43 | 80.94 | 78.48 | **87.96** |
| | seed 1 | 85.85 | 85.93 | 83.97 | 81.57 | 83.69 | 87.49 | 86.60 | 85.29 | 85.16 | 86.39 | 74.25 | 83.33 | 80.67 | **87.96** |
| | seed 2 | 83.08 | 84.38 | 84.15 | 81.59 | 83.86 | 87.22 | 84.85 | 83.12 | 83.58 | 82.26 | 73.64 | 82.41 | 81.63 | **88.63** |
| | Mean± std | 84.76 ± 1.48 | 85.72 ± 1.24 | 84.14 ± 0.17 | 81.09 ± 0.85 | 83.81 ± 0.10 | 87.04 ± 0.57 | 85.86 ± 0.90 | 84.68 ± 1.37 | 84.55 ± 0.85 | 84.75 ± 2.19 | 74.11 ± 0.41 | 82.23 ± 1.21 | 80.26 ± 1.61 | **88.18 ± 0.39** |
| | | | | | | | Target Acc=84% | | | | | | | | |
| Heterogeneous scenario 3 | seed 0 | 84.21 | 84.06 | 82.87 | 82.14 | 83.32 | 84.50 | 81.38 | 82.49 | 83.36 | 84.66 | 75.76 | 81.33 | 76.51 | **85.79** |
| | seed 1 | 82.47 | 84.61 | 84.03 | 81.49 | 83.12 | 84.96 | 82.49 | 81.52 | 84.95 | 81.96 | 74.58 | 82.97 | 75.45 | **86.47** |
| | seed 2 | 82.02 | 82.94 | 84.46 | 80.66 | 83.48 | 85.00 | 83.02 | 83.05 | 83.60 | 82.70 | 75.22 | 82.22 | 77.49 | **85.41** |
| | Mean± std | 82.90 ± 1.16 | 83.87 ± 0.85 | 83.79 ± 0.82 | 81.43 ± 0.74 | 83.31 ± 0.18 | 84.82 ± 0.28 | 82.30 ± 0.84 | 82.35 ± 0.77 | 83.97 ± 0.86 | 83.11 ± 1.40 | 75.19 ± 0.59 | 82.17 ± 0.82 | 76.48 ± 1.02 | **85.89 ± 0.54** |
| | | | | | | | Target Acc=72% | | | | | | | | |
| Heterogeneous scenario 4 | seed 0 | 72.71 | 75.56 | 76.16 | 73.32 | 78.54 | 80.91 | 74.91 | 74.15 | 74.75 | 70.14 | 61.05 | 65.53 | 66.80 | **84.69** |
| | seed 1 | 73.22 | 74.37 | 76.53 | 72.6 | 77.17 | 81.86 | 75.63 | 72.62 | 75.73 | 71.16 | 62.87 | 67.82 | 68.36 | **85.10** |
| | seed 2 | 69.14 | 75.54 | 77.75 | 70.46 | 78.51 | 84.86 | 75.51 | 71.65 | 74.58 | 71.72 | 61.93 | 68.99 | 70.01 | **85.30** |
| | Mean± std | 71.69 ± 2.22 | 75.16 ± 0.68 | 76.81 ± 0.83 | 72.13 ± 1.49 | 78.07 ± 0.78 | 82.54 ± 2.06 | 75.35 ± 0.39 | 72.81 ± 1.26 | 75.02 ± 0.62 | 71.01 ± 0.80 | 61.95 ± 0.91 | 67.45 ± 1.76 | 68.39 ± 1.61 | **85.03 ± 0.31** |
| | | | | | | | Target Acc=65% | | | | | | | | |
| Heterogeneous scenario 5 | seed 0 | 65.51 | 70.35 | 74.60 | 74.14 | 75.36 | 76.88 | 68.01 | 71.74 | 74.75 | 66.79 | 62.82 | 64.99 | 64.57 | **77.70** |
| | seed 1 | 66.41 | 70.51 | 72.67 | 69.23 | 74.70 | 73.00 | 69.08 | 67.16 | 71.57 | 70.17 | 65.75 | 65.82 | 66.89 | **73.29** |
| | seed 2 | 69.67 | 68.07 | 72.0 | 69.56 | 74.50 | 74.98 | 70.01 | 68.08 | 70.43 | 69.08 | 61.60 | 65.31 | 65.76 | **75.14** |
| | Mean± std | 67.20 ± 2.19 | 69.64 ± 1.36 | 73.09 ± 1.35 | 70.98 ± 2.74 | 74.85 ± 0.45 | 74.95 ± 1.94 | 69.03 ± 1.00 | 69.53 ± 3.35 | 71.25 ± 0.71 | 68.68 ± 1.73 | 63.39 ± 2.13 | 65.37 ± 0.42 | 65.74 ± 1.16 | **75.38 ± 2.21** |
| Total Mean Acc ± std | | 78.51 ± 7.99 | 80.05 ± 6.80 | 80.74 ± 5.14 | 77.81 ± 5.52 | 80.86 ± 3.89 | 83.55 ± 5.02 | 79.72 ± 6.86 | 79.02 ± 6.98 | 80.08 ± 6.07 | 78.64 ± 7.62 | 71.32 ± 7.94 | 76.61 ± 8.80 | 74.62 ± 6.81 | **84.70 ± 5.14** |

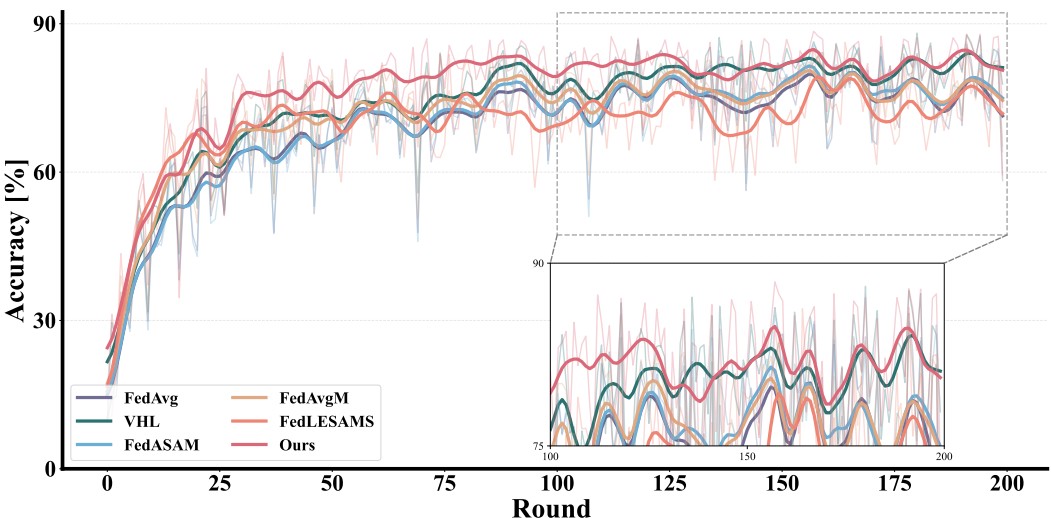

Figure 15: The training process visualization of top-5 baselines and our method FedGPS on CIFAR-10, heterogeneity degree $\alpha = 0.1$, local epochs $E = 5$ and total client number $K = 10$ under heterogeneous scenario 1.

