# OpenReview forum: "FedGPS: Statistical Rectification Against Data Heterogeneity in Federated Learning"
_NeurIPS.cc/2025/Conference — NeurIPS 2025 poster_

### Official Review · Reviewer_uV3Z · 2025-06-24

**Clarity:** 2
**Significance:** 2
**Originality:** 2
**Rating:** 4
**Confidence:** 5

**Summary:**

This paper presents a federated learning framework, named FedGPS, to address the data heterogeneous problems in FL, which 1) uses surrogate information to guide the local models to learn data distribution aligned with the global distribution and 2) refines the update direction at each step based on other client information at the gradient view.

**Questions:**

Please see the weakness.

**Ethical Concerns:**

["NO or VERY MINOR ethics concerns only"]

**Final Justification:**

The author addresses the main concerns. I decicde to raise my rating

**Limitations:**

Yes

**Quality:**

2

**Strengths And Weaknesses:**

### Strengths:

* S1: The problem solved in this paper, data heterogeneity in federated learning, is an important and widely concerned issue.

* S2: The paper designs the method from the perspective of loss-landscape, which has good interpretability.

### Weakness

* W1. The paper lacks novelty and is incremental. The method proposed in the paper seems to be a combination of the method proposed in [14,23] and the method proposed in [24].

* W2. Related work is incomplete and lacks some key SOTA works. For example. lack of comparison with methods based on heuristic search [R1][R2]. The motivation of these methods is similar to SAM, aiming to improve the generalization of FL by guiding training towards flat areas. Since these methods use completely different technical routes from SAM, it is incomplete for the author to only mention SAM in related work, and these methods need to be discussed and compared. In addition, discussing more the SOTA SAM-based works [R3][R4] can improve the quality of the paper.

* W3. From the training curves (Figures 4, 7, 10, and 11) provided by the author, the performance improvement seems limited. In addition, the author uses the highest accuracy as the evaluation standard of accuracy in the table, but it is difficult to accurately stop the training at the round with the highest performance in the real scene. I suggest that the author use the mean and standard deviation to show the experimental results.

* W4. The authors seem to have only evaluated extreme heterogeneous scenarios, not normal heterogeneous scenarios, such as \alpha=0.3, 0.6, and IID, which are commonly used settings.

* W5. The author's experiments on cross-device scenarios are insufficient, lacking experiments on more clients (e.g., K=500, 1000) and different participation ratios (the author only evaluates 10% of devices participating in local training).

* W6. It is unreasonable for the author to set E=1 for the main experiments, which will cause the FL communication overhead to increase sharply. In addition, according to the experimental results in the appendix, when E=5, the advantage of the method decreases significantly, and the author needs to give an explanation.

[R1] Is aggregation the only choice? Federated learning via layer-wise model recombination. KDD 2024

[R2] FedCross: Towards accurate federated learning via multi-model cross-aggregation. ICDE 2024

[R3] Beyond Local Sharpness: Communication-Efficient Global Sharpness-aware Minimization for Federated Learning. CVPR 2025

[R4] Flexible Sharpness-Aware Personalized Federated Learning. AAAI 2025

---

> ### Author Rebuttal · Authors · 2025-07-31
>
> We are grateful to the reviewers for their diligent efforts and for acknowledging both the **significance of our research questions and the interpretability** of our methodology.
> >**_Q1:_** Lacks novelty and is incremental.
>
> _**A1:**_ Thank the reviewers for the questions they raised, which gave us the opportunity to further clarify the contributions of FedGPS. Extensive experiments have shown that sharing statistical information has great potential to alleviate data heterogeneity, which also inspires FedGPS. Furthermore, FedGPS **takes into account statistical information from both the distribution and gradient perspectives** to enhance a global view.  Specifically, the **shift at the distribution level** is corrected through two-stage alignment based on the surrogate dataset. At the gradient level, **the gradient information of other clients** is considered before local training from a more global perspective. FedGPS has covered many heterogeneous scenarios across various datasets, further demonstrating its robustness.
>
> >**_Q2:_** Related work is incomplete and lacks some key SOTA works.
>
> _**A2:**_ We thank the reviewers for pointing out the shortcomings of related work, and we will add a discussion of the latest paper in the next version to improve our related work[R1-R4]. Additionally, we add FedMR[R1] as our new baselines; the results are listed in Tables 1-6. We will also add the results of this heuristic search method in the next version.
>
> **Table 1.The results in FedGPS under Heterogeneous scenario 1 with CIFAR-10, the best accuracy is reported.**
> ||K=10,$\lambda_s$=50%,R=500, E=1|K=10,$\lambda_s$=50%,R=200, E=5| K=100,$\lambda_s$=10%,R=500, E=1|
> |:-:|:-:|:-:|:-:|
> |FedMR [R1]|84.28|86.83|-|
> |**FedGPS(Ours)**|**90.31**|**88.47**|**78.32**|
>
> **Table 2. The results in FedGPS under Heterogeneous scenario 1 with CIFAR-10, average, and std accuracy are reported.**
> ||K=10,$\lambda_s$=50%,R=500, E=1|K=10,$\lambda_s$=50%,R=200, E=5| K=100,$\lambda_s$=10%,R=500, E=1|
> |:-:|:-:|:-:|:-:|
> |FedMR|78.63$\pm$3.20|80.33$\pm$3.76|-|
> |**FedGPS(Ours)**|**83.93$\pm$4.41**|**82.08$\pm$4.96**|**68.09$\pm$6.51**|
>
> **Table 3. The results in FedGPS under Heterogeneous scenario 1 with SVHN, the best accuracy is reported.**
> ||K=10,$\lambda_s$=50%,R=500, E=1|K=10,$\lambda_s$=50%,R=200, E=5| K=100,$\lambda_s$=10%,R=500, E=1|
> |:-:|:-:|:-:|:-:|
> |FedMR|91.32|86.64|-|
> |**FedGPS(Ours)**|**94.20**|**93.61**|**95.03**|
>
> **Table 4. The results in FedGPS under Heterogeneous scenario 1 with SVHN, average, and std accuracy are reported.**
> ||K=10,$\lambda_s$=50%,R=500, E=1|K=10,$\lambda_s$=50%,R=200, E=5| K=100,$\lambda_s$=10%,R=500, E=1|
> |:-:|:-:|:-:|:-:|
> |FedMR|80.01$\pm$6.08|75.21$\pm$5.75|-|
> |**FedGPS(Ours)**|**86.22$\pm$6.36**|**88.91$\pm$3.42**|**93.97$\pm$1.02**|
>
> **Table 5.The results in FedGPS under Heterogeneous scenario 1 with CIFAR-100, the best accuracy is reported.**
> ||K=10,$\lambda_s$=50%,R=500, E=1|K=10,$\lambda_s$=50%,R=200, E=5| K=100,$\lambda_s$=10%,R=500, E=1|
> |:-:|:-:|:-:|:-:|
> |FedMR|69.45|67.45|-|
> |**FedGPS(Ours)**|**71.14**|**68.90**|**58.77**|
>
> **Table 6. The results in FedGPS under Heterogeneous scenario 1 with CIFAR-100, average, and std accuracy are reported.**
> ||K=10,$\lambda_s$=50%,R=500, E=1|K=10,$\lambda_s$=50%,R=200, E=5| K=100,$\lambda_s$=10%,R=500, E=1|
> |:-:|:-:|:-:|:-:|
> |FedMR|67.35$\pm$0.86|66.76$\pm$1.79|-|
> |**FedGPS(Ours)**|**69.87$\pm$0.57**|**67.36$\pm$1.56**|**54.33$\pm$1.28**|
>
> >**_Q3:_** The performance improvement seems limited. I suggest that the author use the mean and standard deviation to show the experimental results.
>
> _**A3:**_ Thanks to the reviewers for their suggestions, we show part of the mean and standard deviation results in Table 7. Based on the mean and std metrics, it can be shown that FedGPS can **achieve higher performance gain on mean** compared to the highest accuracy. More results will be updated in the next version.
> **Table 7. The mean and standard deviation results under CIFAR-10 of the original results in our paper .**
> || K=100,$\lambda_s$=10%,R=500, E=1|K=10,$\lambda_s$=50%,R=500, E=1|
> |:-:|:-:|:-:|
> |FedAvg|34.26$\pm$10.27|72.56$\pm$8.35|
> |FedAvgM|39.72$\pm$11.45|72.44$\pm$8.81|
> |FedProx|35.52$\pm$9.96|79.24$\pm$4.17|
> |SCAFFOLD|55.26$\pm$2.57|73.59$\pm$6.39|
> |CCVR|61.77$\pm$1.44|81.78$\pm$1.34|
> |VHL|56.36$\pm$11.03|81.61$\pm$5.03|
> |FedASAM|30.63$\pm$9.55|73.48$\pm$6.86|
> |FedExp|25.59$\pm$6.83|71.67$\pm$6.45|
> |FedDecorr|41.84$\pm$12.42|73.03$\pm$6.47|
> |FedDisco|-|-|
> |FedInit|52.26$\pm$10.5|72.32$\pm$8.01|
> |FedLESAM|61.48$\pm$6.18|78.45$\pm$5.25|
> |NUCFL|39.29$\pm$8.44|72.54$\pm$6.78|
> |**FedGPS(Ours)**|**68.09$\pm$6.51**|**83.93$\pm$4.41**|
>
>
> >**_Q4:_** The authors seem to have only evaluated extreme heterogeneous scenarios, not normal heterogeneous scenarios, such as $\alpha$=0.3, 0.6, and IID, which are commonly used settings.
>
> _**A4:**_ We sincerely appreciate the reviewers' constructive feedback, which has significantly advanced FedGPS's applicability in federated scenarios, allowing it to **cover settings from homogeneous to extremely heterogeneous**. To demonstrate this, we present the experimental results for FedGPS on the CIFAR-10 dataset, with K=100 and E=1, $\lambda_s=10%$ in Table 8.
>
> **Table 8. The results of different federated scenarios under CIFAR-10.**
> || $\alpha$=0.3 K=100,$\lambda_s$=10%,R=500, E=1| $\alpha$=0.6 K=100,$\lambda_s$=10%,R=500, E=1| I.I.D K=100,$\lambda_s$=10%,R=500, E=1|
> |:-:|:-:|:-:|:-:|
> |FedAvg|82.67|88.07|90.67|
> |FedAvgM|84.74|88.22|91.08
> |FedProx|82.07|86.40|90.75
> |SCAFFOLD|77.41|81.94|85.81
> |CCVR|83.87|85.78|90.31
> |VHL|85.89|88.19|90.74
> |FedASAM|83.59|86.49|91.03
> |FedExp|82.71|85.47|89.33
> |FedDecorr|86.33|88.39|90.45
> |FedDisco|-|-|-|
> |FedInit|86.61|89.30|91.01
> |FedLESAM|86.44|88.46|90.43
> |NUCFL|77.56|81.65|84.94
> |**FedGPS(Ours)**|**87.60**|**89.43**|**91.22**
>
> >**_Q5:_** Lacking experiments on more clients (e.g., K=500, 1000) and different participation ratios (the author only evaluates 10% of devices participating in local training).
>
> _**A5:**_ Thanks to the reviewers for motivating a more comprehensive evaluation of FedGPS, we conduct more clients (K=500) in Table 9 to demonstrate that FedGPS still surpasses other baselines.
>
> **Table 9. More number of clients experimental results**
> || K=500,$\lambda_s$=10%,R=500, E=1|
> |:-:|:-:|
> |FedAvg|73.19|
> |FedAvgM|75.43|
> |FedProx|76.72|
> |SCAFFOLD|64.18|
> |CCVR|64.43|
> |VHL|80.58|
> |FedASAM|75.06|
> |FedExp|74.48|
> |FedDecorr|74.24|
> |FedDisco|-|
> |FedInit|73.05|
> |FedLESAM|80.67|
> |NUCFL|73.98|
> |**FedGPS(Ours)**|**82.18**|
>
> Furthermore, inspired by reviewers, we also perform experiments with different sampling rates; the results are listed in Table 10.
>
> **Table 10. Different sampling rates for K=100.**
> || $\alpha$=0.1 K=100,$\lambda_s$=5%,R=500, E=1| $\alpha$=0.1 K=100,$\lambda_s$=10%,R=500, E=1|  $\alpha$=0.1 K=100,$\lambda_s$=20%,R=500, E=1| $\alpha$=0.1 K=100,$\lambda_s$=50%,R=500, E=1|
> |:-:|:-:|:-:|:-:|:-:|
> |FedAvg|34.59|48.22|67.38|72.45
> |FedAvgM|35.32|58.80|68.34|73.89
> |FedProx|32.89|52.84|65.25|74.24
> |SCAFFOLD|36.73|60.17|67.59|69.34
> |CCVR|57.93|64.06|74.25|78.78
> |VHL|57.14|72.70|76.58|80.78
> |FedASAM|34.78|46.35|61.84|66.48
> |FedExp|34.38|38.26|58.35|64.52
> |FedDecorr|-|63.69|77.31|80.03
> |FedDisco|-|-|-|-|
> |FedInit|30.56|71.01|76.89|77.59
> |FedLESAM|68.86|72.64|76.35|76.90
> |NUCFL|38.85|53.72|68.31|71.91
> |**FedGPS(Ours)**|**70.89**|**78.32**|**79.72**|**81.97**
>
> Due to limited space, in addition to the highest acc reported, **we will also include the results of mean and standard deviation in the revised version**, and thank the reviewers for their suggestions.
>
> >**_Q6:_**  It is unreasonable for the author to set E=1 for the main experiments, which will cause the FL communication overhead to increase sharply. In addition, according to the experimental results in the appendix, when E=5, the advantage of the method decreases significantly, and the author needs to give an explanation.
>
> _**A6:**_ Thanks to the reviewers for their suggestions on local epochs. In fact, our setting follows many papers [R4-R7] which set local epochs to 1. In addition, we also experimented with E=5, and because the number of local epochs increased, the overfitting of the client to the local data and the deviation from the global makes the larger local epochs more difficult to improve and optimize, but our method still achieves the SOTA effect in all scenarios, which demonstrates the robust performance of FedGPS.
>
> References:\
> [R1] Hu M, et al. Is aggregation the only choice? federated learning via layer-wise model recombination. In KDD, 2024.\
> [R2] Xing X, et al. Flexible Sharpness-Aware Personalized Federated Learning. In AAAI, 2025.\
> [R3] Hu M, et al. FedCross: Towards accurate federated learning via multi-model cross-aggregation. In ICDE, 2024.\
> [R4] Caldarola D, et al. Beyond Local Sharpness: Communication-Efficient Global Sharpness-aware Minimization for Federated Learning. In CVPR, 2025.\
> [R5] Tang Z, et al. Virtual homogeneity learning: Defending against data heterogeneity in federated learning. In ICML, 2022.\
> [R6] Yang Z, et al. Fedfed: Feature distillation against data heterogeneity in federated learning. In NeurIPS, 2023.\
> [R7] Kim, Junhyung Lyle, et al. Adaptive Federated Learning with Auto-Tuned Clients. In ICLR, 2024.

---

> > ### Comment · Reviewer_uV3Z · 2025-08-04
> >
> > Thanks for your response. The rebuttal addresses the main concerns. Therefore, I have decided to raise my rating.

---

> > > ### Author Response · Authors · 2025-08-04
> > >
> > > We are delighted to have addressed the reviewers' concerns. We sincerely appreciate the reviewers for the feedback and raised scores. We are also deeply grateful for the valuable contributions which enhancing the quality of our paper.

---

### Official Review · Reviewer_V82G · 2025-07-01

**Clarity:** 3
**Significance:** 3
**Originality:** 3
**Rating:** 4
**Confidence:** 3

**Summary:**

This paper first identified a key issue that many existing FL algorithms lack robustness across diverse data heterogeneity scenarios. To address this problem, Federated Goal-Path Synergy (FedGPS) is proposed, aiming to enhance the robustness by leveraging both statistical and gradient information from other clients. Specifically, the author proposed a static goal-oriented objective function, including two stages: (1) aligning local distributions with a local surrogate distribution, and (2) aligning the local surrogate distribution with the global surrogate distribution. Another technique, dynamic path-oriented gradient rectification, is proposed to integrate non-self gradient information into local updates to correct the local optimization directions.

**Questions:**

1. Could the authors provide additional technical details regarding the generation process of surrogate distributions? Moreover, it would be beneficial to provide some justifications for why Gaussian distribution was chosen.
2. How do the authors plan to address potential privacy concerns associated with the proposed two techniques (i.e., surrogate distributions and share of non-self gradient information)?
3. Have the authors considered evaluating other non-iid scenarios beyond the Dirichlet partitioning method, such as assigning each client data from exactly C distinct classes (C=2 or C=3), ensuring that each client only sees a specific subset of all labels?

**Ethical Concerns:**

["NO or VERY MINOR ethics concerns only"]

**Final Justification:**

I think the authors have clarified most of my questions, particularly those related to the scope of the experimental evaluation and privacy considerations. Accordingly, I have decided to raise my score to 4.

**Limitations:**

Yes

**Quality:**

2

**Strengths And Weaknesses:**

Strengths:
- This paper is well-motivated and clearly structured.
- The idea of modeling the global data distribution using a two-stage surrogate distribution alignment is novel and intuitive.
- The authors conducted extensive and thorough experiments under a wide range of non-IID conditions to demonstrate the robustness of their proposed FedGPS compared to comprehensive baseline approaches.

Weaknesses:
- I am missing some details on how exactly the surrogate distribution is generated. The paper only provides a high-level description: “sampled from distinct Gaussian distributions”.
- The paper lacks sufficient discussion on any potential privacy risks from their proposed two techniques. The authors claim that the surrogate distribution is a privacy-preserving approach since no raw statistical information is shared. However, the local surrogate distributions might still be vulnerable to reconstruction or membership inference attacks.  Additionally, the dynamic path-oriented gradient rectification explicitly utilizes gradient information derived from other clients. More comprehensive analysis and clarification regarding the potential privacy risk associated with this gradient-sharing technique are needed.
- The motivation provided in Section 4.3, specifically the statement, "This insight builds on the ingenious concept that a malicious client in FL …, offering extra advantages," seems problematic and may require further clarification.

---

> ### Author Rebuttal · Authors · 2025-07-31
>
> We sincerely thank the reviewer for the feedback, noting our paper is '**well-motivated and clearly structured**', as well as '**novel and intuitive**'. We deeply appreciate the reviewer's valuable efforts in improving the quality of our paper, and we address the concerns as follows:
> >**_Q1:_**  How exactly the surrogate distribution is generated? It would be beneficial to provide some justifications for why Gaussian distribution was chosen.
>
> _**A1:**_ We thank the reviewers for their constructive suggestions, which have significantly enhanced FedGPS's readability. Our method **assigns a distinct Gaussian distribution to each class** in the original dataset. The surrogate data is then generated by sampling from these class-specific Gaussian distributions, with each client sampling from the same distribution for a given class (Note that, unlike heterogeneous, where clients only have partial classes, for the surrogate dataset, each client has surrogate data for all classes). The **underlying insight is that any real data distribution can be effectively transformed from a Gaussian distribution** [R1]. This approach also benefits from the ease of sampling Gaussian noise, making it well-suited for resource-constrained environments like FL. Furthermore, we evaluate other data distributions, such as the Laplace distribution, and as shown in Table 1, they yielded comparable performance to the Gaussian distribution. Ultimately, we select Gaussian noise for FedGPS.
>
> **Table 1. Performance comparison of Gaussian and Laplace distribution sampling for surrogate dataset generation.**
> ||K=10,$\lambda_s$=50%,R=500, E=1|K=10,$\lambda_s$=50%,R=200, E=5| K=100,$\lambda_s$=10%,R=500, E=1|
> |:-:|:-:|:-:|:-:|
> |Laplace distrbuiton for FedGPS|90.21|87.87|78.12|
> |Gaussian distrbuiton for FedGPS|**90.31**|**88.47**|**78.32**|
>
> >**_Q2:_**  Lacks sufficient discussion on any potential privacy risks from their proposed two techniques. The authors claim that the surrogate distribution is a privacy-preserving approach since no raw statistical information is shared. However, the local surrogate distributions might still be vulnerable to reconstruction or membership inference attacks. Additionally, the dynamic path-oriented gradient rectification explicitly utilizes gradient information derived from other clients. More comprehensive analysis and clarification regarding the potential privacy risk associated with this gradient-sharing technique are needed.
>
> _**A2:**_ Thank the reviewers for their comments on the paper, which have helped us improve the quality of the paper. In summary, **the two technologies of FedGPS do not transmit any additional client's information to any other client compared to traditional methods**. On the contrary, **FedGPS replaces raw data prototypes with surrogate prototypes instead, thereby further protecting privacy compared to previous works [R2-R5]**. We list the privacy clarification among surrogate distribution and gradient information in the following:
> - **Regarding the surrogate distribution privacy issue**, because the surrogate is sampled from different Gaussian distributions, **it does not contain any information related to the local data**. Further, we transmit the aggregated class-wise prototypes of surrogate data. After **aggregating all the high-dimensional embeddings of each class**, the surrogate data information is further protected. Many papers[R2-R5] also transmit using the original data prototypes, which offer a weaker level of information protection than what we have. (Detailed information can be referred to the response to **_Reviewer 3XPZ Q1_**).
> - **Regarding the gradient information privacy issue**, FedGPS **doesn't transmit the gradient information of a certain client to any other client.** Every client **only uploads its own information to the server and downloads the aggregated information from the server**. This process is the same as most other federated methods in uploading and downloading the aggregated information. (Detailed information can be referred to the response to **_Reviewer 2KuF Q1_**).
>
> We also add these discussions to the next version. Thanks for your suggestions.
>
>
> >**_Q3:_** "This insight builds on the ingenious concept that a malicious client in FL …, offering extra advantages," seems problematic and may require further clarification.
>
> _**A3:**_ Thanks to the reviewers for suggesting where misunderstandings may have occurred. Our motivation for incorporating the non-self gradient indeed **draws a high-level concept from the model replacement strategy in the backdoor to attack federated models.** In this scenario, the malicious client leverages a profound understanding of the aggregation process and the collective behavior of benign clients. By accurately anticipating **how other clients' updates contribute to the global model**, the attacker strategically crafts and scales their own malicious update. This sophisticated manipulation allows them to effectively "replace" the global model with their backdoored version, despite the presence of numerous benign participants. The core insight here is that **knowledge of the aggregate influence of other clients provides a significant advantage in shaping the global model**. We thank the reviewer again for pointing out this confusing part. We will clarify these insights in the next version.
>
> >_**Q4:**_ Have the authors considered evaluating other non-iid scenarios beyond the Dirichlet partitioning method, such as assigning each client data from exactly C distinct classes (C=2 or C=3), ensuring that each client only sees a specific subset of all labels?
>
> _**A4:**_ Thanks to the reviewers for their suggestions, we consider more heterogeneous partitioning methods to better show that FedGPS is applicable to other partitions besides Dirichlet. We conduct experiments where each client's data only has C classes. We test both (C=2 and C=3), the results are listed in Table 2.
>
> **Table 2. CIFAR-10 results with $\alpha=0.1, K=100, \lambda_s=10\%, R=500, E=1$.**
> || C=2|C=3|
> |:-:|:-:|:-:|
> |FedAvg|51.75|69.97|
> |FedAvgM|50.61|73.21|
> |FedProx|49.64|68.96|
> |SCAFFOLD|56.08|74.34|
> |CCVR|55.83|73.69|
> |VHL|73.53|84.16|
> |FedASAM|55.33|69.37|
> |FedExp|47.53|67.29|
> |FedDecorr|69.36|79.60|
> |FedDisco|-|-|
> |FedInit|54.80|71.12|
> |FedLESAM|66.62|79.42|
> |NUCFL|50.03|72.89|
> |**FedGPS(Ours)**|**78.17**|**85.71**|
>
> References:\
> [R1] Rezende D, Mohamed S. Variational inference with normalizing flows. In ICML, 2015.\
> [R2] Huang, Wenke, et al. Rethinking Federated Learning with Domain Shift: A Prototype View. In CVPR, 2023.\
> [R3] Tan, Yue, et al. Fedproto: Federated prototype learning across heterogeneous clients. In AAAI, 2021.\
> [R4] Dai, Yutong, et al. Tackling data heterogeneity in federated learning with class prototypes. In AAAI, 2023\
> [R5] Wang, Lei, et al. Taming cross-domain representation variance in federated prototype learning with heterogeneous data domains. In NeurIPS, 2024.

---

> > ### Comment · Reviewer_V82G · 2025-08-02
> >
> > Thank you for your response. I think the authors' clarification has addressed most of my questions, especially regarding the scope of the experimental evaluation.

---

> > > ### Author Response · Authors · 2025-08-02
> > >
> > > We are sincerely grateful to the reviewers for the valuable comments,  which have led to a significant improvement in our paper's quality. We also appreciate the acknowledgment of our response.  If you have any further questions, we look forward to addressing your concerns to further improve our work. Thank you for your dedication to FedGPS.

---

> > > > ### Comment · Reviewer_V82G · 2025-08-08
> > > >
> > > > I don't have any further questions, and I have raised my score to borderline accept.

---

> > > > > ### Author Response · Authors · 2025-08-09
> > > > >
> > > > > Thank you for your reply, we are very glad that you raised score to borderline accept. Your recognition is a great help to FedGPS. Thank you for your efforts and time on our paper.

---

### Official Review · Reviewer_3XPZ · 2025-07-02

**Clarity:** 2
**Significance:** 3
**Originality:** 2
**Rating:** 4
**Confidence:** 4

**Summary:**

The paper proposed FedGPS, a framework that integrates statistical distribution and gradient information from other clients in federated learning, aiming to mitigate data heterogeneity issues and improve the performance of trained models. The framework consists of a static goal-oriented objective that incorporates feature alignment between local real and surrogate distributions, as well as alignment between local and global surrogate distributions, and dynamic path-oriented rectification that rectifies local update direction using gradients from other clients. Experiments showed that the proposed method achieved improved accuracies and convergence speed compared to previous work.

**Questions:**

1. Why is it necessary to introduce the surrogate distributions? Based on the assumption that the global surrogate distribution is computed from the global real distribution (which is not defined in the paper), if equation (6) is the objective function for local update, then why not simply use the real global and local statistics, i.e., use the distance between the local feature and global feature (sent from the server) in the loss function? In addition, what is purpose of aligning the local real distribution and the local surrogate distribution if the latter stems from the former one?

2. What is the communication overhead introduced by the extra statistical and gradient information transmission?

**Ethical Concerns:**

["NO or VERY MINOR ethics concerns only"]

**Final Justification:**

The rebuttal has addressed most of my concerns.

**Limitations:**

yes

**Quality:**

3

**Strengths And Weaknesses:**

Strengths:

1. The paper proposed an effective method to tackle data heterogeneity issues. Statistical information when properly used can help alleviate the divergence between local and global models while preserving user privacy. The proposed method is intuitive and could be applied in a wide range of cases.

2. The paper conducted extensive experiments to show the performance of the proposed method, which improves over previous work in most cases.

Weaknesses:

1. The paper did not formally define the local surrogate distribution and global surrogate distribution, which are crucial components of the proposed method. In addition, more information needs to be transmitted between the server and clients such as the global features and gradients of other clients. It should be made clear what information is transmitted in each round of communication.

2. The notations are inconsistent in Sections 3 and 4. Below are some examples:

- In line 111 the set of clients is K but in line 112 it is S.
- In line 163, sum of distributions.
- In lines 175-176, Stage 1 and Stage 2 are not further defined or referenced.
- In line 180, \delta in bound.

---

> ### Author Rebuttal · Authors · 2025-07-31
>
> We deeply appreciate the reviewer's efforts, particularly their recognition that the proposed method is **intuitive and could be applied in a wide range of cases**, as well as their **affirmation of our experimental results**. We also address the questions raised in the following.
> >**_Q1:_** Formally define the local surrogate distribution and global surrogate distribution.
>
> _**A1:**_ Thank the reviewers for helping to make our paper clearer and easier to understand. In summary,
> - We **give the formal definition** of local and global surrogate distribution.
> - The detailed process of **how to use local and global surrogate distribution of FedGPS in practice**.
>
> We first recap some notations in Table 1(we will revise some notations in the next version). Then we give the formal define of local surrogate distribution and global surrogate distribution in the following:
>
> **Table 1. Some notations.**
> |Notation|Meaning|
> |:-:|:-:|
> |$\mathcal{D}_s$| Surrogate Dataset(No raw information, **all clients have the same dataset**)|
> |$\mathcal{D}_k$| Local raw dataset, each client hold its own|
> |$\psi_k$|Feature extractor of client k|
> |$\xi_s$| Data sample from surrogate dataset|
> |$\xi_k$|Data sample from local dataset of client $k$ |
> |$c$| $c$-th class in dataset|
> |$\mathcal{E}^c$|Embedding representation of $c$-th calss (512-dimension in our paper)|
>
> **Definition 1.(Local Surrogate Distribution).** For a client $k$ in a federated learning system, the **Local Surrogate Distribution ($\mathcal{P}_{k}^{s}$)** is conceptually defined as the set of feature embeddings obtained by passing each data point from surrogate dataset through the $k$-th local model's feature extractor($\psi_k$).
>
> **Definition 2.(Global Surrogate Distribution).** At the server side, the **Global Surrogate Distribution ($\mathcal{P}^{s}$)** is defined as the set of feature embeddings obtained by passing each data point from surrogate dataset through the global extractor($\psi$).
>
> In the implementation, to ensure privacy and reduce communication overhead, what is transmitted to the server is a compressed, privacy-preserving statistical representation of surrogate distribution. Typically these embeddings are then aggregated to form a set of class-wise prototype vectors (e.g., 512-dimensional), with each prototype representing a specific class. This distribution serves as a compact proxy for local surrogate distribution:
> $$\mathcal{E}_{k}^{c}=\frac{1}{|\mathcal{D}_s^c|}\sum\psi_k(\xi_s^c),\tag{1}$$
>
> where $ \mathcal{D}^c_s $ denotes the subset of the surrogate dataset belonging to class $c$. Follow the Eq(1) we can get the $C$ prototypes for each class in the surrogate dataset to present the local surrogate distribution in a privacy-preserving and communication-friendly way. The global surrogate distribution is also replaced with aggregated local surrogate prototypes (only selected clients').
> $$\mathcal{E}^{c}=\sum_{k\in\mathcal{S}_t}\mathcal{E}_k^c\tag{2} $$
> This prototype-based method also been applied in many FL works e.g.,[R1-R4]. **Moreover, FedGPS only transmit the the aggregated surrogate data's prototype, which is safer than the local data's prototype.**
>
>
> >**_Q2:_** More information needs to be transmitted between the server and clients such as the global features and gradients of other clients. It should be made clear what information is transmitted in each round of communication.
>
> _**A2:**_ We sincerely appreciate your kind suggestion. Due to the limited rebuttal space, the detailed communication overheads are listed in Table 1 of the response to **_Reviwer 2KuF_**. In summary,
> - Our method **only introduces additional $C*512$** communication overheads during the upload process, which is much smaller than the original model.
> - During the download process, in addition to the global model, **an additional aggregated gradient information from the previous round of participating clients is required.** This is the same size as the original model, and there is also global aggregated global surrogate prototypes (this can still be ignored). Therefore, **the additional communication overhead of FedGPS is limited.**
> - Inspired by the comments, we **implement a communication-friendly version of FedGPS**, called FedGPS-CF. Compared to FedAvg, **only additional class-wise prototypes that can be ignored are required** still show comparable peroformance with FedGPS.
>
> >**_Q3:_** Notation problems:
>
> _**A3:**_ We appreciate your careful review and highlighting these notation inconsistencies in Sections 3 and 4. We will revise the text accordingly in the next version.
> - In line 111 the set of clients is K but in line 112 it is S.
>     - We will fix this problem in the next version to replace $\mathcal{S}$ with $\mathcal{K}$ in Line 111.
> - In line 163, sum of distributions.
>     - We will replace these notations according to the response to **Q1**.
> - In lines 175-176, Stage 1 and Stage 2 are not further defined or referenced.
>     - Stage 1 denotes the alignment at local period in the Fig.3(a) and Stage 2 is the alignment at local global stage in Fig.3(a). We will fix these problems.
> - In line 180, \delta in bound.
>     - We will fix this problem in the next version to replace $\delta$ with $\kappa$ in the Theorem 4.1.
>
> >**_Q4:_** Why is it necessary to introduce the surrogate distributions? Based on the assumption that the global surrogate distribution is computed from the global real distribution (which is not defined in the paper), if equation (6) is the objective function for local update, then why not simply use the real global and local statistics, i.e., use the distance between the local feature and global feature (sent from the server) in the loss function? In addition, what is purpose of aligning the local real distribution and the local surrogate distribution if the latter stems from the former one?
>
> _**A4:**_ The surrogate dataset is sampled from different Gaussian distributions representing distinct classes. Thus, the surrogate dataset is different from local datasets (**without any information about raw data**). **Secondly,** extensive experiments also demonstrate the potential of sharing statistical information for alleviating data heterogeneity. Therefore, from a distribution perspective, we use a two-stage alignment method to align the locally biased statistical information with the global data while protecting privacy. **Lastly,** FedGPS employs a two-stage approach. On one hand, it **prevents the directly exposure of the statistical information of the original data**. On the other hand, all clients use the same surrogate distribution, which reduces the deviation of both global and local statistical information. (The gap can be bridged by aligning the distribution of different models on the same pure noise dataset)
>
> References:\
> [R1] Huang, Wenke, et al. Rethinking Federated Learning with Domain Shift: A Prototype View. In CVPR, 2023.\
> [R2] Tan, Yue, et al. Fedproto: Federated prototype learning across heterogeneous clients. In AAAI, 2021.\
> [R3] Dai, Yutong, et al. Tackling data heterogeneity in federated learning with class prototypes. In AAAI, 2023\
> [R4] Wang, Lei, et al. Taming cross-domain representation variance in federated prototype learning with heterogeneous data domains. In NeurIPS, 2024.

---

> > ### Comment · Reviewer_3XPZ · 2025-08-06
> >
> > I appreciate the authors' response. My concerns on the distribution formulation and communication overhead are addressed. A suggestion is to formally introduce the surrogate dataset, and then the surrogate distributions.

---

> ### Author Response · Authors · 2025-08-06
>
> Thanks for your valuable comments, which improve the readability of our paper. We are delighted to address your concerns.
>
> With your constructive comments, we will introduce the surrogate dataset first. Our method **assigns a distinct Gaussian distribution to each class** in the original dataset. The surrogate data is then generated by sampling from these class-specific Gaussian distributions, with each client sampling from the same distribution for a given class (Note that, unlike heterogeneous, where clients only have partial classes, for the surrogate dataset, each client has surrogate data for all classes). **This introduction will be added in our revision.**
>
> We are always welcome to solve any further questions, and your help will be invaluable to FedGPS.

---

### Official Review · Reviewer_LAwd · 2025-07-02

**Clarity:** 3
**Significance:** 2
**Originality:** 2
**Rating:** 4
**Confidence:** 2

**Summary:**

Federated Learning (FL) struggles with data heterogeneity, limiting model performance. Existing methods lack robustness across diverse scenarios. This work shows that sharing statistical information helps mitigate heterogeneity by providing a global perspective. Inspired by this, the authors propose ​​FedGPS (Federated Goal-Path Synergy)​​, a novel framework that combines statistical distribution and gradient information from other clients. FedGPS statically adjusts learning objectives to model global data distribution and dynamically refines local updates using external gradient information. Experiments demonstrate that FedGPS outperforms existing methods in various heterogeneity scenarios, proving its effectiveness and robustness across datasets.

**Questions:**

Please refer to the weakness part.

**Ethical Concerns:**

["NO or VERY MINOR ethics concerns only"]

**Final Justification:**

The comments have addressed most of my concerns. I decide to raise my score.

**Limitations:**

Please refer to the weakness part.

**Quality:**

3

**Strengths And Weaknesses:**

Strengths:
1. This paper proposes a robust approach (FedGPS) that effectively integrates statistical and gradient information to address data heterogeneity.
2. This paper conducts experiments across diverse datasets and heterogeneity scenarios.
3. This paper is easy to read and follow.


Weaknesses:
1. For Figure 1, could the authors provide more explanation about the Nemenyi post-hoc test? This method appears less commonly used in evaluations.
2. In line 79, the authors describe refining the update direction using other clients' information. Would it be possible to support this with theoretical justification?
3. For Equation 7, could the authors clarify:
- How is λ determined?
- Why does incorporating non-self gradients improve the update direction?
4. Have the authors considered evaluating their method on Vision Transformers (ViTs) or NLP tasks?

---

> ### Author Rebuttal · Authors · 2025-07-31
>
> We sincerely thank the reviewers for their invaluable assistance and insightful contributions to our paper. We are particularly pleased to hear that our paper was found to be **easy to follow** and that our **extensive experimental work was acknowledged**. We now proceed to address the concerns raised by the reviewers
> >**_Q1:_** Provide more explanation about the Nemenyi post-hoc test. This method appears to be less commonly used in evaluations.
>
> _**A1:**_ Thanks for pointing out this potentially confusing setting. Accordingly, we have added the following explanations in the revision.
>
> The Nemenyi post-hoc test in Figure 1 is a non-parametric statistical method used for pairwise comparisons of multiple groups [R1] (e.g., algorithms or models) after a significant result from a Friedman test (a non-parametric analog to repeated-measures ANOVA). It ranks the performance of each method across multiple independent runs or datasets and computes a "critical distance" (CD) threshold. If the average rank difference between two methods exceeds the CD, their performances are considered statistically significantly different at a given significance level (typically α=0.05). The test is conservative and accounts for multiple comparisons to control the family-wise error rate, making it robust for scenarios like ours, where we evaluate algorithm robustness across diverse heterogeneity partitions (e.g., different random seeds for Dirichlet distributions). \
> In Figure 1, the Nemenyi post-hoc test assesses the robustness of baseline methods across different heterogeneity scenarios. The results show overlapping CD intervals for most baselines, indicating no statistically significant performance differences among them. This finding highlights the need for our proposed approach, as existing methods exhibit limited adaptability to varied data distributions. Furthermore, the **Nemenyi test has been widely adopted in holistic evaluations [R2-R5] and federated learning scenarios[R6-R7]**.
>
> >**_Q2:_** In line 79, the authors describe refining the update direction using other clients' information. Would it be possible to support this with theoretical justification?
>
> _**A2:**_  Thanks to the reviewers' valuable comments to improve the quality of our paper. The **insight behind incorporating information from other clients in FedGPS stems from works like SCAFFOLD** [R8] and other related research [R9], which also use other client information as a control variate to adjust update direction. Beyond this intuition, we provide a theoretical justification using a Taylor expansion to demonstrate that integrating other clients' information can indeed **further decrease the deviation between local and global update directions**. Here's a detailed proof:
>
> We denote local loss function as $f_k(\cdot):\mathbb{R}^d\rightarrow\mathbb{R}$ and global loss function $F(\cdot)$. First of all: The original local update uses the vanilla gradient descent on the local model $\theta_k$ is denoted as $g_{\text{old}}=\nabla f_k(\theta_k).$ For FedGPS, we incorporate non-self gradient $\delta_{\theta_k}$ to local model , we denote $\frac{\delta_{\theta_k}}{||\delta_{\theta_k}||}$ as $g_k'$:
> $$\theta'_k=\theta+\lambda_g g_k'.$$
> Then we get a new update direction computed based on the new model parameters $\theta_k'$:
>
> $$g_{\text{new}}=\nabla f_k(\theta_k').$$
> For a continuously twice-differentiable function $f_k(\theta)$, the gradient function$\nabla f_k(\theta)$ expands around point $\theta$ along direction $g_k'$:
> $$\nabla f_k (\theta_k+\lambda_g g_k')\approx \nabla f_k(\theta_k) +\nabla^2 f_k(\theta_k)(\lambda_g g_k')+R_3,$$
> where $R_3$ represents higher-order terms that can be neglected, and $\nabla^2 f_k$ is the Hessian at $\theta_k$. Thus, we can get:
> $$\nabla f_k (\theta_k')\approx \nabla f_k(\theta_k) +\lambda_g \nabla^2 f_k(\theta_k)g_k' .$$
> Here we assume that the loss function is convex. So the Hessian $\bar{H}=\nabla^2 f_k$  is positive semi-definite. Ideally, we assume non-self gradients contain all the gradients from other clients; we can denote $\delta_{\theta_k} = \nabla F(\theta) - \nabla f_k(\theta_k)$. Substitute into the expansion:
> $$\nabla f_k(\theta_k') \approx \nabla f_k(\theta_k) + \lambda_g \bar{H} (\nabla F(\theta) - \nabla f_i(\theta))=(I - \lambda_g \bar{H}) \nabla f_k(\theta_k) + \lambda_g \bar{H} \nabla F(\theta),$$
> where $I$ is the identity matrix. As a result:
> - The original bias between local and global gradient: $d_0 = ||g_{\text{old}} - \nabla F(\theta)||$
> - Refined update direction bias between new model parameters $\theta_k'$ and global gradient: $d'= \|g_{\text{new}} - \nabla F(\theta)\| \approx \|(I - \lambda_g \bar{H}) (\nabla f_k(\theta) - \nabla F(\theta))\| \leq \|I - \lambda_g \bar{H}\| \cdot d_0$. If $\lambda_g$ is tuned to make $||I - \lambda_g \bar{H} < 1||$, then $d'<d_0$, which reduces the shift.
>
> In practice, direct access to all client gradient information is often limited due to privacy and communication overhead. Nevertheless, through careful hyperparameter tuning, FedGPS consistently achieves state-of-the-art (SOTA) performance across various heterogeneous scenarios.
>
>
> >**_Q3:_** For Equation 7, could the authors clarify:(1)How is $\lambda$ determined?(2)Why does incorporating non-self gradients improve the update direction?
>
> _**A3:**_ Thanks to the reviewers for their valuable questions.
> - Our approach is **not very sensitive to different $\lambda_g$**, we **conduct the grid search** for $\lambda_g\in${$0.1,0.2,0.3,0.4,0.5,0.6,0.7,0.8,0.9,1.0$}. The results are listed in the following Table 1. Therefore, in order to save the time of hyperparameter tuning, we choose 0.5 as a fixed parameter. The following results in our paper also prove that this parameter can still make FedGPS achieve the effect of SOTA. This also shows that **FedGPS does not need to spend too much time on parameter tuning.**
>
> **Table 1. Different hyperparameters of $\lambda_g$.**
> ||0.1|0.2|0.3|0.4|0.5|0.6|0.7|0.8|0.9|1.0|
> |:-:|:-:|:-:|:-:|:-:|:-:|:-:|:-:|:-:|:-:|:-:|
> |FedGPS|77.55|77.68|77.99|77.37|**78.32**|77.88|78.08|77.93|77.79|77.87|
>
> - Regarding why incorporating non-self gradients improves the update direction, our **intuition stems from works like SCAFFOLD[R8], which show that integrating information from other clients can refine one's own update direction**. Furthermore, we also provide a theoretical justification that incorporating the information from other clients can further decrease the deviation between local and global update directions. The detailed proof is listed in the response to **_Q2_**.
>
> We are grateful for the reviewer's constructive comments, which have enabled us to offer a more thorough explanation of FedGPS's SOTA performance.
>
> >**_Q4:_** Have the authors considered evaluating their method on Vision Transformers (ViTs) or NLP tasks?
>
> _**A4:**_ We sincerely appreciate your constructive question. Accordingly, we explore the applicability of our method beyond the current evaluations. The following results and discussions have been added to the revision.
> To verify the effectiveness of our method, **we further incorporate the ViT model in [R10] for FL**. The results, which demonstrate consistent improvements in accuracy compared to baselines, have been reported in Table 2.
>
> **Table 2.ViT-based model under CIFAR-10 dataset with $\alpha=0.1$.**
> || K=100,$\lambda_s$=10%,R=500, E=1|K=10,$\lambda_s$=50%,R=500, E=1|
> |:-:|:-:|:-:|
> |FedAvg|30.45|60.85|
> |FedAvgM|33.56|60.97|
> |FedProx|48.46|59.32|
> |SCAFFOLD|35.38|63.56|
> |CCVR|-|61.92|
> |VHL|47.36|70.09|
> |FedASAM|35.43|58.55|
> |FedExp|28.35|-|
> |FedDecorr|39.63|61.18|
> |FedDisco|-|59.19|
> |FedInit|32.34|39.98|
> |FedLESAM|45.83|63.69|
> |NUCFL|38.89|62.33|
> |**FedGPS(Ours)**|**56.71**|**73.56**|
>
> Due to limited time, we will incorporate more experimental results of ViT in various scenarios in the upcoming version.
>
> Regarding NLP tasks, our method may require additional adaptations to effectively handle sequential data in NLP, especially how to design surrogate information for statistical information alignment. While we have not yet conducted these experiments due to the need for further methodological refinements, this insightful question has inspired us to prioritize NLP evaluations in future extensions of our work. We appreciate the reviewer's insightful comments in broadening the scope of our research.
>
> References:\
> [R1] Nemenyi, Peter Bjorn. Distribution-free multiple comparisons. Princeton University, 1963.\
> [R2] Demšar, Janez. Statistical comparisons of classifiers over multiple data sets. In JMLR 2006.\
> [R3] Chen, Xiangzhi, et al. Disentangling cognitive diagnosis with limited exercise labels. In NeurIPS, 2023.\
> [R4] Jansen, Christoph, et al. Statistical multicriteria benchmarking via the GSD-front. In NeurIPS, 2024.\
> [R5] Guidotti, Riccardo, et al. Generative model for decision trees. In AAAI, 2024.\
> [R6] Yang, Yuwen, et al. Federated multi-task learning on non-iid data silos: An experimental study. In ICMR, 2024.\
> [R7] Song, Yukun, et al. Causal multi-label feature selection in federated setting. In arXiv:2403.06419, 2024.\
> [R8] Karimireddy S P, et al. Scaffold: Stochastic controlled averaging for federated learning. In ICML, 2020.\
> [R9] Wenjing Yan, et al. Problem-Parameter-Free Federated Learning. In ICLR 2025.\
> [R10] vision-transformers-cifar10: Training Vision Transformers (ViT) and related models on CIFAR-10.

---

> > ### Author Response · Authors · 2025-08-06
> >
> > **Dear Reviewer #LAwd:**
> >
> > Thanks again for your efforts in reviewing this paper, providing informative and constructive thoughts. We’d appreciate it if you could confirm whether we will have further discussion. We understand that the workload of the reviewer, and in order to save time, we have made a brief summary of the above response:
> > - About the Nemenyi post-hoc, we give the **detailed explanation** of the Nemenyi post-hoc test.
> > - Regarding why incorporating non-self gradients improves the update direction. Based on your insightful comments, we **give the theoretical justification** using Taylor expansion, which demonstrates **integrating other clients' gradient information can further decrease the deviation between local and global update directions.**
> > - For the selection of $\lambda_g$. We give the hyperparameter experiment that shows **FedGPS is not very sensitive to different $\lambda_g$**.
> > - For the problem of using ViT with FedGPS. We conduct **experiments with ViT-based models**. The results still show SOTA performance of FedGPS.
> >
> > We appreciate the contribution of reviewers to our paper, and we look forward to answering your questions, which will be very important to our paper.
> >
> > Best regards,\
> > Authors #3545.

---

> > ### Comment · Reviewer_LAwd · 2025-08-06
> >
> > Thank you for your response. The comments have addressed most of my concerns. I decide to raise my score.

---

> > > ### Author Response · Authors · 2025-08-06
> > >
> > > We are very grateful to the reviewers for their recognition of our reply. Your questions helped us clarify the advantages of FedGPS more clearly, both theoretically and empirically. We thank you for your acknowledgment and improved score. Thank you again for your efforts on our paper.

---

### Official Review · Reviewer_2KuF · 2025-07-02

**Clarity:** 4
**Significance:** 3
**Originality:** 2
**Rating:** 4
**Confidence:** 3

**Summary:**

This work conducts comprehensive evaluations across different heterogeneous scenarios, and asserts most existing methods exhibit limited robustness. Motivated by the observation, this work proposes FedGPS, which statically modifies each client's learning objective to resemble the global data distribution, while adjusting local update directions with additional gradient information.

**Questions:**

- Doesn’t incorporating gradient information from other clients incur additional computational cost? In Def 4.2, it seems that to compute $\delta_{\theta_i}^t$, each participating client has to communicate $\Delta_{\theta_k}^{t-1}$. Is this incorporated as "communication round" in the experiments?

- By communicating local gradient information with other clients, can it hurt the improved privacy aspect of federated learning?

- How come comparison to adaptive methods [1, 2, 3] are not included? These also try to address heterogeneity issue in federated learning.

[1] S. Reddi et al., Adaptive Federated Optimization

[2] J.L. Kim et al., Adaptive Federated Learning with Auto-Tuned Clients

[3] S. Mukherjee et al., Locally Adaptive Federated Learning

**Ethical Concerns:**

["NO or VERY MINOR ethics concerns only"]

**Final Justification:**

The authors provided very detailed answer to my concerns. In particular, they clarified that the clients do not have to share the local gradient with other clients. They further provided additional experimental results, including a variant of their proposed method with the same communication complexity with FedAvg, and still exhibit competitive performance. I am thus raising my score.

**Limitations:**

Yes

**Paper Formatting Concerns:**

No major formatting issue

**Quality:**

3

**Strengths And Weaknesses:**

### Strength:
- The paper is well-written and easy to follow.
- Reported empirical performance is strong, although the reported metrics seem a bit unfair, if the communication cost of non-self gradient computation is not taken into account.

### Weakness:
- It seems that the proposed method requires many more communications than existing methods. In particular, I'm confused about the non-self gradient for each client. It seems that each client has to communicate high-dimensional vector to every other client. This is even more expensive than one "communication round" of federated learning in the traditional sense (every participating client periodically communicates with the server). Is this true?

---

> ### Author Rebuttal · Authors · 2025-07-31
>
> We are truly grateful for the reviewers' dedicated efforts, which greatly improve the quality of FedGPS. Their **well-written and easy to follow** comments were invaluable, and we especially appreciate their **strong affirmation of our experimental results**. Additionally, we provide the following answers to the reviewers' concerns:
> >**_Q1:_** FedGPS requires many more communications than existing methods. It seems that each client has to communicate a high-dimensional vector to every other client. Extra computational cost. Incorporating gradient information from other clients incurs additional computational cost? By communicating local gradient information with other clients, can it hurt the improved privacy aspect of federated learning?
>
> _**A1:**_ Thank you for your valuable feedback and for raising this important point regarding the communication and computational costs of FedGPS. In summary,
> - During the download stage, aligning with your valuable feedback, there is one additional model (containing the aggregated gradient) of the same size as the global model.
> - During the upload stage, **there is no additional communication overhead**.
> - During training, **no communication is required between clients**, namely, only communication is needed between the client and the server.
> - We **further implement a variant with no extra communication overhead** called FedGPS-CF, showing comparable performance with FedGPS.
>
> Thus, FedGPS protects privacy as previous works and brings about 1.5 times communication overhead than vanilla methods, e.g., FedAvg.
>
> Inspired by your constructive comments, we have provided a detailed table description of the two consecutive upload and download rounds in the revised version, along with the associated local computational requirements. Additionally, we include a comparison with FedAvg to demonstrate that FedGPS does not impose significant additional overhead. We denote the whole model size as $M$ and the total classes of the dataset as $C$, e.g., $C=10$ for CIFAR-10.
>
> **Table 1. The whole process comparison between FedAvg and FedGPS (Ours)**.
> |Process|FedAvg|FedGPS|
> |:-:|:-:|:-:|
> |Global aggregation at round t-1|Update global model$\theta^{t}=\sum\theta_{k,E}^{t-1}$|1.Update global model $\theta^{t}=\theta^{t-1}+\eta_g\sum_{k\in\mathcal{S}_{t-1}}\Delta_k^{t-1}$ 2.Update global surrogate protoypes $\mathcal{E}^{c}=\sum\mathcal{E}^c_k$|
> |Explanation||Apart from the direct aggregation parameters, e.g, FedAvg-like, the $\Delta$ of client parameters has also been widely used in many studies[R1-R4].|
> |Server|Select subset $\mathcal{S}_t$ to participate Round t|Select subset $\mathcal{S}_t$ to participate Round t|
> |Round t selected $\mathcal{S}_t$ **download** from server|Global model $\theta^{t}$ **(#Comm $M$)**|1.Global model $\theta^{t}$;2.Global model update infomation $\Delta \theta^t=\theta^{t}-\theta^{t-1}=\eta_g\sum_{k\in\mathcal{S}_{t-1}}\Delta_k^{t-1}$ **(#Comm $2M+C*512$)**|
> |Explanation||$\Delta\theta^t$ contains the gradient aggregation information updated by the selected client in the previous round $t-1$, The global surrogate distribution is represented by a prototype for each class. The prototype for each class is a 512-dimensional vector.|
> |Local opereation|Update local model $\theta^{t}_{k,0}=\theta^{t}$ ($0$ means the model without local epochs training)|1.Update local model $\theta^{t}_{k,0}=\theta^{t}$;2.Compute Non-Self Gradient based on $\Delta\theta^{t}$. |
> |Local extra opereation explanation|| **A:** $\delta_{\theta}^{t}=\Delta \theta^{t}-\Delta_{k}^{t-1}$ If this client is selected last round which means we should distract its local update $\Delta_{k}^{t-1}$ of last round (this is kept locally) **B:** $\delta_{\theta}^t=\Delta \theta^{t}$ If this client is not selected last round which means $\Delta \theta^{t}$ contains all other client's gradient information. |
> |Local training|Traditional SGD use loss function and local data|SGD use Eq(6) in FedGPS with local data|
> |Explanation||Incorporating gradient information from other clients only occurs by adding the parameters together before each gradient descent. This additional computation overhead is almost negligible and can be disregarded. There are a total of iteration times of parameter summation operations. (**Negligible additional computation expenses**)|
> |Round t selected $\mathcal{S}_t$ **upload** to server|New local model parameters $\theta^{t}_{k,E}$ **(#Comm $M$)**|1.Local updated parameters $\Delta_{k}^{t}=\theta_{k,E}^{t}-\theta_{k,0}^{t}$ 2.Compute local surrogate prototypes $\mathcal{E}_{k}^{c}=\frac{1}{\|\mathcal{D}_s^c\|}\sum\psi_k(\xi_s^c)$ **(#Comm $M+C*512$)** |
> |Global aggregation at round t|$\theta^{t+1}=\sum\theta_{k,E}^{t}$|$\theta^{t+1}=\theta^{t}+\eta_g\sum_{k\in\mathcal{S}_{t}}\Delta_k^{t}$|
>
> The extra $C*512$ (where $C * 512 \ll M$，e.g., 0.05% in CIFAR-10) is the local uploaded local surrogate distribution prototypes and downloads the global surrogate distribution prototype. （The detailed explanation of local and global surrogate information can be referred to the reply to **_Reviewer 3XPZ Q1_**）.
> To further clarify, we emphasize that in each round of FedGPS, **we do not require all clients to send their own gradients to other clients**. Instead, based on the information from the selected clients in the previous round, we aggregate their gradients for dual purposes: updating the global model and providing aggregated gradient information for client selection in the next round. As a result, the upload cost remains identical to that of vanilla FedAvg. During the download from the server, clients receive not only the latest model but also the aggregated gradient information from the server. This aggregation process inherently protects individual client gradients from being accessed by others (**addressing privacy concerns**). Consequently, throughout the entire FedGPS process, there is no direct information exchange between any clients, and all information obtained from the server is solely aggregated data.
>
> Inspired by your valuable comments, **we further design a communication-friendly variant of FedGPS, called FedGPS-CF**. Specifically, when **uploading** the model, FedGPS-CF, like FedGPS, **only has a communication cost of $M+C*512$**. When **downloading** from the server, it **still only downloads the global model and global surrogate prototypes, meaning the communication cost remains $M+C*512$**. Here, we use the difference between the global model downloaded in two rounds from the server to represent the gradient aggregation information of other clients. Similarly, if a client is selected in both adjacent rounds, its own update information should be removed. Finally, FedGPS-CF achieves comparable performance to FedGPS, the results are listed in Table 2. However, it reduces the download overhead of $M$, making FedGPS-CF only have an additional communication cost of $C*512$ compared to FedAvg, which is negligible. Once again, we thank you for your help and for inspiring our paper.
>
> **Table 2. The results between FedGPS-CF and FedGPS under Heterogeneous scenario 1 with CIFAR-10**.
> ||K=10,$\lambda_s$=50%,R=500, E=1|K=10,$\lambda_s$=50%,R=200, E=5| K=100,$\lambda_s$=10%,R=500, E=1|
> |:-:|:-:|:-:|:-:|
> |FedGPS-CF|90.01|88.13|78.07|
> |FedGPS|90.31|88.47|78.32|
>
> >**_Q2:_** How come comparison to adaptive methods are not included? These also try to address heterogeneity issue in federated learning.
> >
> _**A2:**_ We agree that comparing a new method with adaptive methods is necessary, since these methods are also outstanding approaches to address data heterogeneity in FL. Following your valuable comments, we have **selected two representative adaptive methods** $\Delta$-SGD and FedAdam, from references [R5, R6] and **incorporated them as additional baselines in our experiments**. Along with the other methods already evaluated, these comparisons further demonstrate the superiority of FedGPS. The results are listed in Tables 3, 4, and 5. We have discussed these papers [R5-R7] and listed these results in our revised version.
>
> **Table 3. The results in FedGPS under Heterogeneous scenario 1 with CIFAR-10.**
> ||K=10,$\lambda_s$=50%,R=500, E=1|K=10,$\lambda_s$=50%,R=200, E=5| K=100,$\lambda_s$=10%,R=500, E=1|
> |:-:|:-:|:-:|:-:|
> |FedAdam [R5]|85.34|85.05|67.43|
> |$\Delta-SGD$ [R6]|86.96|86.66|69.49|
> |**FedGPS(Ours)**|**90.31**|**88.47**|**78.32**|
>
> **Table 4. The results in FedGPS under Heterogeneous scenario 1 with SVHN.**
> ||K=10,$\lambda_s$=50%,R=500, E=1|K=10,$\lambda_s$=50%,R=200, E=5| K=100,$\lambda_s$=10%,R=500, E=1|
> |:-:|:-:|:-:|:-:|
> |FedAdam|89.80|91.01|93.27|
> |$\Delta-SGD$|89.78|90.21|94.08|
> |**FedGPS(Ours)**|**94.20**|**93.61**|**95.03**|
>
> **Table 5. The results in FedGPS under Heterogeneous scenario 1 with CIFAR-100.**
> ||K=10,$\lambda_s$=50%,R=500, E=1|K=10,$\lambda_s$=50%,R=200, E=5| K=100,$\lambda_s$=10%,R=500, E=1|
> |:-:|:-:|:-:|:-:|
> |FedAdam|69.89|65.33|55.43|
> |$\Delta-SGD$|70.07|67.78|57.48|
> |**FedGPS(Ours)**|**71.14**|**68.90**|**58.77**|
>
> References:\
> [R1] Durmus Alp Emre Acar, et al. Federated learning based on dynamic regularization In ICLR, 2021.\
> [R2] Taehwan Lee, et al. Rethinking the Flat Minima Searching in Federated Learning. In ICML, 2024.\
> [R3] Ziqing Fan, et al. Locally Estimated Global Perturbations are Better than Local Perturbations for Federated Sharpness-aware Minimization. In ICML, 2024.\
> [R4] Wenjing Yan, et al. Problem-Parameter-Free Federated Learning. In ICLR 2025.\
> [R5] Reddi, Sashank J., et al. Adaptive Federated Optimization. In ICLR, 2021.\
> [R6] Kim, Junhyung Lyle, et al. Adaptive Federated Learning with Auto-Tuned Clients. In ICLR, 2024.\
> [R7] Mukherjee S, et al. Locally adaptive federated learning. In TMLR, 2023.\
> [R8] Karimireddy S P, et al. Scaffold: Stochastic controlled averaging for federated learning. In ICML, 2020.

---

> > ### Author Response · Authors · 2025-08-06
> >
> > **Dear Reviewer #2KuF:**
> >
> > Thanks for your valuable comments, which enhance the quality of our paper. It also makes our paper more accurately convey the key idea of FedGPS. We understand that the workload of rthe eviewer, and in order to save time, we have made a brief summary of the above response:
> > - For the communication overheads, we give a detailed table to explain the whole process of FedGPS. Based on the detailed explanation, it can be seen that **FedGPS doesn't require communication between clients,** only an additional aggregated model is needed when the clients download information from the server. The communication overhead incurred by other surrogate distribution prototypes is negligible compared to the model size.
> > - Inspired by your constructive comments, we also **propose a communication-friendly version called FedGPS-CF, which has nearly no communication overheads.**
> > - Regarding the privacy problem, our method and vanilla FL have comparable privacy protection ability since the clients do not communicate gradients with each other.
> > - Follow your suggestions to compare with adaptive methods. We implement two adaptive methods: FedAdam and $\Delta$-SGD.
> >
> > We'd be glad to answer any questions and look forward to any further discussion. Thanks for your great help with our paper again.
> >
> > Sincerely,\
> > Authors #3545.

---

> > > ### Author Response · Authors · 2025-08-06
> > > **Follow up the Rebuttal**
> > >
> > > **Dear Reviewer #2KuF:**
> > >
> > > Thank you again for your valuable time in reviewing our paper and your constructive comments. The time is constrained for more discussions. We’d be grateful if you could give more suggestions and look forward to further discussions to enhance this paper.
> > >
> > > Best regards,\
> > > Authors of #3545.

---

> > > > ### Comment · Reviewer_2KuF · 2025-08-06
> > > >
> > > > Dear authors,
> > > >
> > > > Thank you very much for the detailed and meticulous reply, and I apologize for the late reply.
> > > >
> > > > I appreciate the thorough explanation and the additional experimental results. All my previous concerns are resolved, and the fact that the proposed method is competitive even against adaptive methods is makes the proposed method a valuable addition to the community. I have updated my score accordingly.

---

> > > > > ### Author Response · Authors · 2025-08-06
> > > > >
> > > > > Thanks to the reviewers for their responses. We are happy to have resolved all your questions and updated your scores. We also appreciate your recognition of FedGPS as a **valuable addition to the community**. If you have further suggestions, please don't hesitate to let us know. Thank you for your dedication to FedGPS.

---

### Note · Authors · 2025-08-13

Dear PC, SAC, AC, and Reviewers:\
We sincerely thank you for dedicating your time and expertise to reviewing our paper. Your insightful and valuable feedback has significantly enhanced the quality of our work. We are pleased to highlight the following positive aspects noted by the reviewers:
- Well-written and easy to follow (Reviewer 2KuF, V82G, LAwd).
- Important problem (Reviewer uV3Z), well-motivated, idea is novel and intuitive (Reviewer V82G), proposed method is intuitive and could be applied in a wide range of cases (Reviewer 3XPZ), good interpretability (Reviewer uV3Z).
- Strong empirical performance, experiments across diverse datasets and scenarios, extensive experiments (Reviewer 2KuF, LAwd, 3XPZ, V82G).

During the rebuttal, we mainly addressed the following questions raised by the reviewers, which really enhanced the quality of our paper:
- **Overheads of FedGPS**:We provided a detailed explanation of the communication and local computation processes, clarifying that FedGPS does not require communication between clients.
- **Privacy concerns**: We demonstrated that FedGPS offers privacy protection comparable to vanilla federated learning.
- Concern about **local and global surrogate distribution**: We included a formal definition of local and global surrogate distributions and elaborated on their practical application in FedGPS.
- To further validate FedGPS, we **conducted new experiments**, including:
    - 1. Additional baseline;
    - 2. More Heterogeneous Scenarios;
    - 3. More clients;
    - 4. More heterogeneity partition method;
    - 5. Different sampling rate;
    - 6. ViT-based model.

These revisions will be incorporated into the updated version of the paper. We are grateful for the reviewers' thorough engagement with our work and for their willingness to reconsider their ratings in light of our rebuttal. We are delighted that our responses addressed all reviewer concerns.

Once again, we extend our heartfelt thanks to the PC, SAC, AC, and Reviewers for their invaluable contributions to improving our paper.

Best regards,\
Authors #3545.

---

### Decision · Program_Chairs · 2025-09-17

**Decision:**

Accept (poster)

**Comment:**

(a) Summary of claims and findings:

The paper proposes FedGPS, a federated learning framework designed to improve robustness under data heterogeneity. It combines two ideas: (1) a goal-oriented objective that aligns local models using surrogate statistical information in two stages (local real -> local surrogate; local surrogate -> global surrogate), and (2) a path-oriented rectification that adjusts local update directions with aggregated non-self gradient information provided by the server. The authors claim improved robustness across many heterogeneity settings and datasets. In rebuttal, they clarified that clients never communicate directly; only aggregated information is shared via the server. They also introduced a communication-friendly variant, FedGPS-CF, and added extensive experiments, including adaptive baselines, ViT models, wider heterogeneity regimes (including IID and milder non-IID), more clients (up to K=500), and participation-rate sweeps. They provided definitions for surrogate distributions and a Taylor-expansion-based justification for gradient rectification.

(b) Strengths:

- Addresses an important FL challenge with an intuitive combination of distributional alignment and gradient rectification.
- Clear writing and organization per several reviewers.
- Strong and now broader empirical evaluation after rebuttal: many datasets, heterogeneity regimes, client counts, and baselines (including adaptive methods).
- Practical communication clarification and a lighter FedGPS-CF variant.
- Formalization of surrogate distributions and a simple theoretical rationale for gradient rectification.
- Reported means and standard deviations added; ViT experiments included.

(c) Weaknesses / missing pieces:

- Novelty is partly incremental; components relate to prototypes and control-variate/gradient-adjustment ideas (e.g., SCAFFOLD-like intuitions). The main novelty is in the specific synergy and two-stage surrogate alignment.
- Privacy is discussed but not rigorously analyzed. While only aggregated info is shared, a formal threat model or empirical privacy risk assessment (e.g., attack simulations) is missing.
- Some notational inconsistencies in the original submission; authors committed to fix.
- Communication overhead, while clarified, still adds roughly 1.5x download vs FedAvg for the main variant; CF reduces this but details belong in the main text with clear accounting.
- Limited theory: the Taylor argument is heuristic; no convergence or robustness guarantees.
- NLP tasks are deferred; ViT coverage improved but remains limited.

(d) Reasons for decision:

On balance, the paper is a solid empirical contribution with a clear problem framing and a practical method that appears robust across many conditions. The rebuttal addressed the key concerns: communication misunderstanding, scope of evaluation (including adaptive baselines, larger K, different α/C-label splits, participation rates), and definitions/notation. Multiple reviewers raised their scores to borderline accept after rebuttal, and no major blocking flaw remains. The novelty is moderate and the theory is light, so I recommend poster rather than a higher track. Still, the breadth of experiments and practical relevance justify acceptance.

(e) Rebuttal impact and discussion:

- R2KuF (communication/privacy/adaptive baselines): Authors clarified no client-to-client comms; only server-shared aggregated gradients and prototypes. Provided detailed protocol and FedGPS-CF with negligible extra overhead; added adaptive baselines (FedAdam, µ-SGD) showing competitive or superior results. Reviewer’s concerns were resolved; score raised. I give substantial weight to this resolution.
LAwd (stats test, theory for rectification, lambda choice, ViT/NLP): Authors explained Nemenyi test usage, gave a Taylor-based justification, provided a λ grid showing low sensitivity, added ViT results, and discussed NLP as future work. Reviewer raised score. These points are satisfactorily addressed for an empirical paper.
- 3XPZ (definitions, comm clarity, notation): Authors added formal definitions of local/global surrogate distributions via class-wise prototypes and clarified round-by-round communication; committed to fix notation. Reviewer’s concerns addressed; they raised to borderline accept. This alleviates clarity concerns.
- V82G (novelty, baselines breadth, metrics, heterogeneity breadth, K/participation, E setting): Authors added SOTA/heuristic baselines (e.g., FedMR), reported mean±std, added normal heterogeneity and IID settings, K=500, participation sweeps, and discussed E=1 vs E=5 behavior. Reviewer acknowledged and raised score. While novelty remains moderate, empirical breadth is now convincing.
- uV3Z (privacy risks, surrogate generation, motivation phrasing, additional partitions): Authors detailed Gaussian surrogate generation (also tried Laplace), clarified privacy stance (aggregated prototypes and no direct gradient sharing), refined the wording around the “malicious client” analogy, and added C-class label partition results. Reviewer raised score. I consider privacy discussion improved but still largely qualitative; acceptable for poster.

I read the author response and the ensuing discussion. The rebuttal addressed the major objections; no external information beyond the reviews and rebuttal was used.